# Effect of debris-flow sediment grain-size distribution on fan morphology

Haruka Tsunetaka[1], Norifumi Hotta[2], Yuichi Sakai[3], Thad Wasklewicz[4]

[1]Forestry and Forest Products Research Institute, Ibaraki, Japan
[2]Graduate School of Agricultural and Life Sciences, The University of Tokyo, Tokyo, Japan
[3]School of Agriculture, Utsunomiya University, Tochigi, Japan
[4]Stantec Inc., Environmental Services, Geohazards and Geomorphology Group, Fort Collins, CO, USA

*Correspondence to*: Haruka Tsunetaka (tsunetakaharuka@ffpri.affrc.go.jp)

**Abstract.** Knowledge of how debris flows result in the fan-shaped morphology around a channel outlet is crucial for
mitigation of debris-flow-related disasters and investigation of previous sediment transport from the upper channel.
Therefore, using a flume connected to a deposition area (inundation plane), this study conducted fan-morphology
experiments to assess the effects of differences in grain-size distribution within debris flows on changes in fan morphology.
Two types of debris-flow material, i.e., monogranular particles comprising monodispersed sediment particles and
multigranular particles comprising polydispersed sediment particles, were used to generate monogranular and multigranular
experimental debris flows, respectively. By adjusting the average grain size coincident between the monogranular and
multigranular flows, we generated two types of debris flow with similar debris mixture hydrographs but different grain-size
distributions in the flume. Although the flow depths were mostly similar between the monogranular and multigranular flows
before the start of the debris-flow runout at the deposition area, the runout distances of the front of the multigranular flows
were shorter than those of the monogranular flows. The difference in runout distance was responsible for the variations in the
extent to and location in which the debris flows changed their direction of descent, resulting in the different shapes and
morphologies of the fans in response to grain-size distribution. Although the direction of descent of the flows changed
repeatedly, the extent of morphological symmetry of the debris-flow fans increased at a similar time during fan formation
irrespective of the grain-size distribution. In contrast to this similarity in the rate of change in fan symmetry, the shift of the
multigranular flow directions eventually increased the extent of asymmetry in fan morphology and expanded the scale of
deviations in fan morphology between experimental test runs. Therefore, wide-ranging grain-size distributions within debris
flows likely result in complex fan morphology with a high degree of asymmetry.

## 1 Introduction

Debris flows often cause damage to downstream communities and infrastructure through their runout and associated
sediment deposition (Dowling and Santi, 2014). Understanding how debris flows manifest around the channel outlet is
important for mitigation of their impact on downstream areas and for prevention of related hazards. Debris flows often occur

with various recurrence intervals and different magnitude in the same watershed (Jakob et al., 2005; Brayshaw and Hassan, 2009; Frank et al., 2019). Sediment deposition attributable to such debris flows leads to the formation of the fan-shaped morphology around the channel outlet, i.e., the so-called debris-flow fan, which is recognized as a geomorphological record of sedimentary sequences driven by past climatic and environmental conditions (Dühnforth et al., 2007; De Haas et al., 2015a, 2019; Schürch et al., 2016; D'Arcy et al., 2017; Kiefer et al., 2021). In this sense, studies on the morphology and the stratigraphy of debris-flow fans are fundamental to interpretation of previous sediment dynamics and their drivers. Assessing how debris flows result in fan morphology around a channel outlet is therefore crucial both for investigation of sediment transport episodes and for mitigation of debris-flow-related disasters.

The morphology of a debris-flow fan is governed mainly by three processes that are driven by the runout and deposition of debris flows. Debris-flow surges are stacked stepwise around the outlet of the channel while backfilling the existing channel on the fan (De Haas et al., 2016, 2018a). The backfilling process reduces the flow capacity of the existing channel by decreasing the surface gradient, which consequently results in avulsion that shifts the flow direction of subsequent debris-flow surges (De Haas et al., 2016, 2018a). The avulsion of a debris-flow surge involves erosion of the sediment of the fan that leads to channelization on the fan (De Haas et al., 2016, 2018a). The newly formed channel will then be backfilled when further debris-flow surges occur, thereby repeating the fan-forming cycle of the backfilling, avulsion, and channelization processes (De Haas et al., 2016, 2018a). Monitoring in situ debris-flow runout around a channel outlet is difficult because of the low frequency of occurrence of debris flows (e.g., De Haas et al., 2018a; Imaizumi et al., 2019). However, earlier experiments using a reduced-scale flume demonstrated that the composition (grain-size distribution) and sequence of debris-flow surges govern the formation of fan morphology and the tempo of the fan-forming cycle (De Haas et al., 2016, 2018b; Adams et al., 2019; Tsunetaka et al., 2019). Moreover, relationships derived between the sequence of natural debris-flow lobes and fan morphology indicate that the fan-forming cycle is driven by the backfilling, avulsion, and channelization processes, similar to the results obtained from flume tests (De Haas et al., 2018a, 2019).

The critical role of the grain-size distribution within debris flows on fan morphology is attributable to its influence on the characteristics of debris-flow runout. In a channel, a descending debris flow has a wide-ranging grain-size distribution because it erodes sediment particles from the channel bed and entrains them within the flow (Egashira et al., 2001; De Haas and Van Woerkom, 2016). Entrained small particles that can behave like fluid are likely to decrease frictional resistance (Kaitna et al., 2016; Sakai et al., 2019), whereas large particles that behave as solids can increase frictional resistance in the debris flow (von Boetticher et al., 2016; Pudasaini and Mergili, 2019; Nishiguchi and Uchida, 2022). Thus, differences in the grain-size distribution of debris flows can change the runout distance around the channel outlet (De Haas et al., 2015b; Hürlimann et al., 2015). Moreover, the velocity that erodes channel deposits is susceptible to the influence of both grain-size distribution and slope of the channel bed (Egashira et al., 2001; Takahashi, 2007), and this erosion velocity differs fundamentally from the velocity that entrains the eroded sediment (Pudasaini and Krautblatter, 2021). The discord between the velocities of erosion and entrainment potentially leads to variation in debris-flow mobility via change of inertia, which is responsible for the variation in runout distance of debris flows (Pudasaini and Krautblatter, 2021). Debris-flow surges result

in the fan morphology via stacking and deposition processes that likely differ in accordance with the variation of runout distance (De Haas et al., 2016, 2018a).

The effects of differences in grain-size distribution on debris-flow mechanics might arise even during debris-flow runout. Debris flows consist of solid (i.e., sediment particles) and fluid (i.e., interstitial water and colloidal sediment particles) phases (Pudasaini and Mergili, 2019; Nishiguchi and Uchida, 2022). When debris flows leave the channel outlet, the relative

difference in velocity between the solid and fluid phases increases and leads to phase separation (Pudasaini and Fischer, 2020; Baselt et al., 2022). Around the channel outlet, the solid phase eventually translates into sediment deposition, but the fluid phase continuously descends with the progress of phase separation. The extent of the phase separation might vary in response to the grain-size distribution within a debris flow (Major and Iverson, 1999; Pudasaini and Fischer, 2020), potentially resulting in further difference in runout distance of the solid phase, in addition to the effects attributable to

sediment erosion and entrainment processes in the channel. In practice, runout distance is also affected by the sediment volume of the debris flow (D'Agostino et al., 2010), which is also governed by the grain-size distribution. In other words, the grain-size distribution can influence the characteristics of both the debris-flow development in the channel and the runout distance after debris flows leave the channel outlet. Therefore, it is difficult to unravel how variation of the grain-size distribution within the debris flow might constrain fan morphology during runout and inundation, while discerning these

differences of the debris flow in the channel. This complexity in the effects of the grain-size distribution within debris flows on flow properties hampers comprehensive interpretation of fan morphology based on the known mechanics of debris-flow runout and inundation.

The primary objective of this study was to assess how the grain-size distribution within a debris flow influences fan morphology, especially during debris-flow runout and inundation. We conducted reduced-scale flume tests to compare fan

morphologies that resulted from single debris-flow surges with different grain-size distributions but with similar sediment mixture hydrographs. To investigate whether differences in the runout properties of both solid and fluid phases cause different sediment deposition patterns, we intended to avoid the effects of morphology resulting from previous debris-flow surges on the debris flow that runs out at the flume outlet. To achieve this, we focused on how a single successive debris-flow surge forms the fan-like morphology around the flume outlet without geomorphological effects arising from previous

debris flows. Thus, in this study, debris-flow fans are defined as the sediment deposition formed by a single successive debris-flow surge rather than the accumulation of multiple debris-flow surges. Using photogrammetry and video-image analysis, we investigated how differences in grain-size distribution within debris flows influence variations in runout characteristics and fan morphologies. The intention underlying this comparison was to interpret the differences in fan morphology in terms of known debris-flow mechanics. The final objective was to elucidate whether differences in grain-size

distribution within debris flows could change fan morphology solely by influencing the runout process without variation of the dynamic properties of the debris flow in the channel.

## 2 Methods

This research consisted of two parts: (1) analysis of debris-flow runout and (2) analysis of debris-flow fan morphology. Analysis of debris-flow runout was performed to assess how differences in the grain-size distribution of the debris flow could affect the distance and velocity of debris-flow runout. Analysis of debris-flow fan morphology was performed to investigate whether fan morphology changes in accordance with differences in the grain-size distribution. On the basis of the results, we discuss the mechanism of changes in fan morphology facilitated by differences in the grain-size distribution.

### 2.1 Flume test

### 2.1.1 Experiment setup and operation

We used a straight flume (8 m long and 0.1 m wide, with a uniform 15° bed slope, Fig. 1a) that imitates a channel to generate the experimental debris-flow surges. A deposition area was connected to the lower end of the flume to experimentally model the formation of a debris-flow fan by the runout and inundation associated with the generated debris-flow surge. In this study, debris-flow runout is defined as the descent of the debris flow downstream from the flume outlet. A 5 m long section at the lower end of the flume was filled with 0.08 m$^3$ of sediment particles that were horizontally flattened to the flume bed to achieve analogous erodible bed conditions for all the experimental test runs (Fig. 1a). The erodible bed was mostly constant at 0.2 m deep but it ranged from 0–0.2 m at the upper and lower ends of the erodible bed (Fig. 1a). Sediment particles (~1 mm in diameter) were glued onto the surface of the deposition area to represent roughness, and we drew square grid lines (0.2 × 0.2 m) to aid measurement of the runout distance and arrival time of the flow front in the deposition area (Fig. 1b).

By suddenly supplying water from the upper end of the flume, we generated a granular–water mixture flow that imitated a debris flow, similar to Lanzoni et al. (2017). We could not control the erodible bed saturation completely because the bed materials included voids. Fully saturated bed conditions were approximated by carefully supplying clear water across the entire erodible bed using watering cans just before we initiated the water supply from the upper end of the flume. Following this operation, a steady flow of clear water (fluid density: ~1000 kg m$^{-3}$) was supplied at a rate of 0.003 m$^3$ s$^{-1}$ for 60 s from the upper end of the flume. The supplied water plunged over the erodible bed and flowed downstream, generating a runoff front over the bed sediment particles. The runoff front scoured the sediment particles of the erodible bed and entrained the eroded material, dispersing the entrained particles throughout the flow depth, and eventually transforming the flow into a granular–water mixture that imitated a single debris-flow surge (Lanzoni et al., 2017). The generated granular–water mixture flow descended to the deposition area, causing runout and inundation, which formed the fan morphology. The slope of the deposition area decreased from 12° to 3° at a rate of 3° per meter (Fig. 1a, b).

### 2.1.2 Sediment material

In this study, we generated granular–water mixture flows with similar sediment mixture hydrographs but different grain-size distributions to compare the effects of debris-flow grain-size distribution on fan morphology during the debris-flow runout and inundation processes. To accomplish this, two types of sediment particles were used to generate two types of granular–

water mixture flows: monogranular particles comprising quasi-monodispersed sediment particles with size of 2.02–3.24 mm (average grain size, $D_{50}$: 2.6 mm) and multigranular particles comprising polydispersed sediment particles with size of 0.6–7.5 mm (Fig. 1c, Table 1). The density and the internal friction angle of both particles were 2640 kg m$^{-3}$ and 34.0°, respectively (Hotta et al., 2021). Hereafter, the granular–water mixture flows generated by the monogranular particles and by the multigranular particles are referred to as monogranular flow and multigranular flow, respectively. We conducted four

separate experimental test runs for both the monogranular flow and the multigranular flow.

Importantly, the sediment volume entrained by the debris flow decreases when the grain size of the sediment particles of the channel bed is sufficiently larger than that within the debris flow (Egashira et al., 2001). This phenomenon has been explained by Pudasaini and Krautblatter (2021) with their mechanical erosion model for debris flows that involves the state of excess energy during the erosion process. Small sediment particles such as cohesive materials might behave like pore

fluid in a debris flow, leading to changes in flow resistance (Kaitna et al., 2016; Sakai et al., 2019; Nishiguchi and Uchida, 2022). These effects attributable to the relatively large and small particles are responsible for changes in debris-flow properties (e.g., flow depth and velocity) during the descent of a debris flow in an experimental flume (Sakai et al., 2019; Hotta et al., 2021), likely leading to changes in fan morphology. In other words, if the experimental debris flows were not constrained with similar flow properties in the flume test runs, it would have been even more difficult to interpret how

debris-flow grain-size distribution might influence fan morphology via flow runout and inundation.

When the multigranular particles comprise cohesionless material without extremely large particles, the generated multigranular flows might exhibit flow depth and velocity analogous to those of monogranular flows with similar $D_{50}$ (Takahashi, 2007). Thus, to avoid occurrences of changes in flow depth and velocity by the small and large particles of the multigranular flows, while using cohesionless particles (>0.6 mm) and removing large particles with size >7.5 mm, we

adjusted the mixing ratio of the multigranular particles to maintain the same $D_{50}$ between the monogranular and multigranular flows (Fig. 1c, Table 1). In doing so, we assumed that the generated multigranular flows were complete two-phase flows consisting of a coarse–solid fraction (sediment particles) phase and a fluid (clear water) phase. The objective was to generate multigranular flows that did not involve the fine–solid fraction phase behaving like pore fluid and to entrain the eroded sediment at a rate similar to that of the monogranular flows.

### 2.1.3 Measurements

We measured the flow depths of the generated debris flows in the flume and investigated the properties of runout and fan morphology at the deposition area. By comparing the changes in the flow depths between the monogranular and

multigranular flows, we assessed whether the multigranular flows exhibited hydrograph and velocity characteristics analogous to those of the monogranular flows. Note that we could not directly measure the flow depths of the generated debris flows because the thickness of the erodible bed decreased sequentially in response to sediment erosion by the debris flow. The continuous sediment erosion in response to debris-flow descent made it impossible to define the boundary between the debris-flow bottom and the bed surface (e.g., Lanzoni et al., 2017), which hampered quantitative measurement of debris-flow depth. Instead of flow depth, we measured changes in the displacement of the flow surface using three ultrasonic displacement meters (Omron, E4PA) at a sampling rate of 50 Hz. The ultrasonic displacement meters were installed at three positions separated by a distance of 1 m above the sediment bed from upstream to downstream in the flume; hereafter, referred to as the upper, middle, and lower measurement positions, respectively (Fig. 1a). Because the initial thickness of the erodible bed was adjusted to 0.2 m, the flow depths of the debris flows were calculated by subtracting this initial thickness from the measured displacement. We compared the flow rate in the flume among the test runs on the basis of differences in the timing at which the debris-flow front reached the lower position.

The measurements of runout and fan morphology relied on image analyses. To observe the runout and fan-forming processes, four digital cameras were installed above the deposition area (Fig. 1a), similar to Tsunetaka et al. (2019). One of these cameras (PENTAX, K-3 II) recorded video of the fan-forming processes at a frame rate of 60 fps. The images extracted from the recorded video were used to analyze the debris-flow runout distance and velocity, and the changes in flow direction during inundation at the deposition area. The other three cameras (Nikon, D5100) were automatically synchronized using the external shutter (Canon, TC-80N3) and captured images at 1 s intervals. These sets of three synchronized images were processed to generate topographic data of the formed fan morphology using a photogrammetry method.

## 2.2 Image-based analysis

### 2.2.1 Runout and inundation analysis

We measured changes in the runout distance of the fronts of the generated debris flows with temporal resolution of 0.1 s using the captured video and the grid lines drawn in the deposition area. During the early stage of debris-flow runout, the solid phase (sediment particles) and fluid phase (clear water) descended synchronously as a complete granular–water mixture flow (Fig. S1a), but then they flowed separately with different velocities in accordance with the deceleration of the solid phase (Fig. S1b). Because the timing of the phase separation was clear in all experimental cases, we measured the fronts of both the solid and the fluid phases after separation to compare the extent of phase separation between the monogranular and multigranular flows. The runout distances of the solid and fluid phases were defined as the distance from the flume outlet to the front of the solid and fluid phases, respectively (Fig. 2).

To investigate the characteristics of debris-flow inundation in the deposition area, we performed particle-image-velocimetry (PIV) analysis. First, we prepared paired image sets consisting of two images extracted at a 1/60 s time resolution from the video. Using a PIV analysis plugin for image-based analysis software based on a cross-correlation algorithm (Tseng et al.,

2012), the paired image sets were processed to estimate the vectors of the flow velocity of either the solid or the water or both at the surface of the deposition area. Because the video was acquired from an almost-vertical direction above the area with a 9° slope, the camera was not strictly vertical to the entire deposition area. Moreover, the development of fan morphology was responsible for the spatial variation in the shooting depth of the video. The accuracy of the velocity projected by PIV analysis was thus not spatiotemporally constant. Given this, the measurements of the flow-velocity vectors

were considered to investigate the changes in flow direction that occurred during fan-morphology formation rather to measure flow velocity.

Generally, the avulsion on debris-flow fans is triggered by the plugging of existing channels by previous debris-flow surges (e.g., De Haas et al., 2018a). However, because only a single successive debris-flow surge was examined for each experimental run in this study, we could not observe clear plugging. Thus, when the flow vectors projected by the PIV

analysis indicated shifting of the flow direction, we considered that the debris flow caused the avulsion. Occurrence points of the avulsion were estimated using the PIV results and orthophotos acquired 20, 30, and 40 s after the start of debris-flow runout. To investigate the changes in the location of the avulsion, we calculated the ratio between the distance from the flume outlet to the occurrence point of the avulsion and the distance of the runout of the solid phase (hereafter, referred to as the normalized avulsion distance). We compared the differences in the normalized avulsion distances between the

monogranular and multigranular test runs.

### 2.2.2 Topographic analysis

We measured the process of fan-morphology formation in response to debris-flow runout and inundation using structure motion multi-view stereo (SfM-MVS) photogrammetry (Westoby et al., 2012; Fonstad et al., 2013). Using the SfM-MVS photogrammetry software (Agisoft, Metashape Professional version 1.5.1), we produced digital elevation models (DEMs)

and orthophotos with 1 mm resolution from respective sets of three synchronized images. Each DEM and orthophoto were georeferenced using the coordinates of the visible (exposed) intersections of the grid lines in the deposition area (i.e., at the intersections of the grid lines that were not concealed by deposited sediment). To assess the DEM accuracy, we used a ruler to directly measure the deposit thickness of the fan morphology at the intersections of the grid lines after each respective experimental run, and we compared the measurements with the deposit thickness extracted from the generated DEM. The

measured elevations corresponded to the DEM-extracted elevations, indicating that the DEMs represented reasonable approximations of the fan morphology (Fig. S2).

During debris-flow inundation in the deposition area, the SfM-MVS photogrammetry could not perform measurements for locations in which flows descended (i.e., moving zones), which resulted in holes in the DEMs due to missing topographic data. Conversely, the vectors of flow velocity projected by PIV analysis could only be observed in moving zones.

Consideration of both the DEMs and the vectors projected by PIV analysis allowed assessment of the relationships between changes in flow direction and fan development.

To assess the differences in the deposition slope of the fans, the surface slope of the final fan was analyzed. First, to avoid the influence of small surface undulations arising from surface roughness of grain-size scale, the resolution of the DEM of the final fan was reduced to 0.01 from 0.001 m. The slope was calculated from the processed DEMs using the 8-direction method of Jenson and Domingue (1998).

Additionally, to investigate the differences in the shape of fans derived from both the monogranular and the multigranular flows, we proposed an index that focuses on fan-shape symmetry. The proposed symmetric index (*SI*) is defined as follows:

$$SI = LL/LR \qquad (1)$$

where *LL* and *LR* represent the length of the fan from the centerline of the flume to the edge of the left-bank side and to the edge of the right-bank side of the fan shape, respectively (Fig. 2). When fan width is close to symmetry, the *SI* value is approximately one. We calculated the *SI* values for the width of the fan at cross sections at 0.2 m intervals from the outlet of the flume using orthophotos and DEMs acquired 10, 20, 30, 40, and 50 s after the start of debris-flow runout.

## 3 Results

### 3.1 Flow properties in the flume

Changes in surface height at the upper measurement position indicate that the erodible bed was gradually eroded after the arrival of the flow front irrespective of the grain-size distribution (Fig. 3a, b). After ~22–23 s, the surface heights of the multigranular test runs decreased to below 0.15 m (Fig. 3b), whereas those of the monogranular test runs were >0.15 m at the same time (Fig. 3a), indicating that bed material was eroded at a slightly faster rate by the multigranular flows than by the monogranular flows. In relation to this difference, the fronts of the multigranular flows reached the middle measurement position somewhat faster than those of the monogranular flows, although the differences in arrival time were <1 s between the monogranular and multigranular test runs (Fig. 3c, d). Focusing on the flow fronts, both the monogranular flows and the multigranular flows descended the flume from the upper to the lower measurement positions in ~6–7 s (Fig. 3e, f). Given an initial erodible bed thickness of approximately 0.2 m, the peaks of the monogranular and multigranular flows developed from ~0.03 to 0.07 m before reaching the flume outlet.

Because the ultrasonic sensors measured the flow or initial bed surface at a point scale, the measurements were susceptible to local undulations in the flow surface and erodible bed. Indeed, some spikes of the flow surface ranging from approximately 0.03–0.05 m were measured, especially at the middle and lower measurement positions, but these impacts were ephemeral (Fig. 3c–f). Before the arrival of the flow, measurements of the surface height were not completely uniform at 0.2 m, especially at the middle measurement position (Fig. 3c, d), reflecting that the initial bed surface was somewhat disturbed and undulated probably because of the need to supply water to meet the saturated bed conditions. These inevitable limitations arising from our operation and measurement settings possibly affected the variation in the peak of flow depth. Importantly, differences in the surface height of the main body of the flows were minimized between the test runs after the descent of the

flow front (Fig 3), indicating that local undulations in the initial bed surface scarcely impacted the flow properties excluding the flow front.

Following the descent of the flow front, the rate of decrease in surface height was found increasingly similar between the test runs irrespective of the grain-size distribution within the debris flows (Fig. 3). It indicates that the thickness of the erodible bed decreased monotonically with time in accordance with the erosion and entrainment of sediment by the flow body and tail. Overall, the results derived from the flume experiments revealed that differences in grain-size distribution did not lead to substantial changes in the hydrograph and arrival time of the generated granular flows in the flume, with the exception of the 260 peak of flow depth.

## 3.2 Runout of debris-flow front

Characteristics of debris-flow runout were clearly different between the monogranular and multigranular flows. Before the runout distance exceeded 1 m, flow velocities (i.e., the slope of the graphs) differed somewhat between the test runs irrespective of the grain-size distribution within the debris flows (Fig. 4), which was likely attributable to variation in the 265 peak of flow depth (Fig. 3e, f). At this stage, the solid and fluid phases of both types of flow descended together as a single complete mixture flow, and their velocities were synchronized with each other.

After the runout distance of the flow fronts exceeded ~1.0 m, the velocities of the monogranular flows decreased gradually with increase in runout distance, but the velocities of the solid and fluid phases remained analogous (Fig. 4a). However, the trend of the multigranular flows differed. The velocity of the solid phase of the multigranular flows decreased rapidly, which 270 increased the relative difference in the velocities of the flow fronts between the solid and fluid phases of the multigranular flows (Fig. 4b). The separation between the solid and fluid phases of the multigranular flows thus occurred at an earlier stage of the runout process in comparison with that of the monogranular flows (Fig. 4).

Following the start of phase separation of the multigranular flows, the solid phase continued its runout with further increase in the relative difference in the velocities between both phases, especially after the runout distance of the fluid phase 275 exceeded 2 m (Fig. 4a). Before the runout distance exceeded 2 m, the velocities of the monogranular flows were similar to those of the fluid phase of the multigranular flows (Fig. 4). Subsequently, the monogranular flows decelerated, whereas the fluid phase of the multigranular flows maintained its velocity and descended at ~0.5 m s$^{-1}$. Consequently, the fluid phase of the multigranular flows traveled slightly faster and progressed further downstream. Phase separation of the monogranular flows occurred after the runout distance of the flow fronts exceeded ~2.7–2.8 m. Therefore, the runout distance and velocity 280 differed not only between the monogranular and multigranular flows but also between the respective solid and fluid phases of these flows.

In this context, the locations at which the front of the solid phase stopped (i.e., deposited sediment particles) differed between the monogranular and multigranular flows. Thus, in the early stage of formation of fan morphology, in contrast to the monogranular flows, lobe-like morphologies were formed on the upstream side by the multigranular flows (Fig. S3). The 285 eventual runout distance of the fronts of the solid phase of the monogranular flows was ~2.7–2.8 m (Fig. 5), which

corresponded to the runout distance at the start of phase separation of the monogranular flows (Fig. 4a). The eventual runout distance of the fronts of the solid phase of the multigranular flows was shorter (~2.2 m) in comparison with that of the monogranular flows (Fig. 5). However, considering the measurements of the changes in runout distance, the solid phase of the multigranular flows descended over 1 m after the start of phase separation (Figs. 4b and 5). Phase separation of the monogranular flows was triggered by termination of the fronts of the solid phase, whereas this relationship was not clear for the multigranular flows.

The grain sizes of the deposits of multigranular test runs 2 and 3 were observed (Figs. 6 and 7). At all observation points, relatively large particles were deposited above the base of the deposition area (i.e., zero on the ruler) to thickness of ~1–2 cm (Figs. 6b–f and 7b–f). More small particles were deposited above the relatively large particles at observation points b–e (Figs. 6b–e and 7b–e), indicating that transported sediment particles were stacked above the lobe-like morphology following the halting of the front of the solid phase. Around the front of the solid phase (i.e., the downstream edge of the fans), only relatively large particles were observed (Figs. 6f and 7f). The sediment particles were thus segregated by grain size, and consequently relatively large particles accumulated at the fronts of the multigranular flows.

### 3.3 Formed fan morphology

The extent of the changes in flow direction and deposition range of sediment particles differed between the monogranular and multigranular flows. In the first 10 s of flow runout, both types of granular flow descended in an approximately straight direction (Fig. S3). After 20 s from the start of flow runout, the monogranular flows descended in a straight line through the zone with a 9°–12° slope (i.e., from the flume outlet to 2 m downstream), but the flow direction shifted somewhat toward the left-bank side owing to avulsion in the zone with a 6° slope (i.e., over 2 m downstream from the flume outlet) (Fig. 8). The multigranular flows, after 20 s from the start of flow runout, changed their flow direction further in the upper zone (i.e., at approximately 1.8 m downstream from the outlet of the flume) in comparison with the monogranular flows (Fig. 9). In multigranular test runs 1 and 4, the flow direction shifted to the left- and right-bank sides, respectively, whereas in multigranular test runs 2 and 3, the flow bifurcated (Fig. 9).

After 30 s from the start of flow runout, the monogranular flows descended continuously further toward the left-bank side, but in test run 4, the flow became slightly bifurcated (Fig. 10). At this stage, in multigranular test run 1, the flow descent direction shifted somewhat from toward the left-bank side to toward the right-bank side (Fig. 11a). In test runs 2–4, the flow direction was mostly maintained but the location at which the flow direction changed moved ~0.2 m upstream (i.e., to approximately 1.6 m downstream from the outlet of the flume) (Fig. 11b–d). After 40 s from the start of flow runout, at ~2 m lower from the outlet of the flume, the flow bifurcated in monogranular test run 1 (Fig. 12a), continuously descended toward the left-bank side in monogranular test runs 2 and 3 (Fig. 12b, c), and mainly descended toward the right-bank side in monogranular test run 4 (Fig. 12d). In the test runs of the multigranular flows, the point of drifting of flow direction occurred further upstream of the deposition area, i.e., ~1.4 m downstream of the outlet of the flume (Fig. 13). The descent direction of the multigranular flow inclined toward the right-bank side in test runs 1 and 4 (Fig. 13a, d), but inclined toward the left-bank

side in test runs 2 and 3 (Fig. 13b, c). Subsequently, there was no substantial change in descent direction of any of the flows (Figs. S4 and S5). The eventual range of sediment deposition differed in response to grain-size distribution (Figs. S6 and S7), and also varied substantially between the multigranular test runs (Fig. S7). The normalized avulsion distances of the monogranular test runs were almost constant at approximately 0.67 from after 20 s to 40 s from the start of flow runout (Fig. 14). Unlike this fixed position of the avulsion of the monogranular flows, the normalized avulsion distance of the multigranular test runs gradually decreased from ~0.78 to ~0.59 from after 20 s to 40 s from the start of flow runout (Fig. 14). This highlights the difference in the trend of the inundation processes between monogranular and multigranular flows.

Focusing on the final fan morphology, the monogranular flows exhibited similar longitudinal profiles among the various test runs (Fig. 15a). Similarly, the multigranular flows also exhibited similar longitudinal profiles irrespective of the direction of shifting of the flow (Fig. 15b). The deposit thickness of the monogranular flows was deeper than that of the multigranular flows in the area more than 2 m downstream from the flume outlet (Fig. 15c), in response to the differences in the runout distances of the flow fronts between the monogranular and multigranular flows (Fig. 5). At the cross section 1 m downstream from the flume outlet, the monogranular flows exhibited relatively symmetric cross-sectional profiles in all test runs (Fig. 15d), whereas the deposit thickness of the multigranular flows varied between test runs (Fig. 15e). The variation in the cross-sectional profiles of the multigranular flows was attributable to whether the peak of sediment deposition was located on the left- or right-bank side, depending on the direction of the shift in flow runout (Fig. 15f). This difference was emphasized further downstream. At the cross section 2.2 m downstream from the flume outlet, the fan width and thickness were similar between the monogranular test runs, and the peak of deposit thickness was located almost at the center of the fan (Fig. 15g). In contrast, the cross-sectional range and peak of sediment deposition differed between the multigranular test runs, indicating asymmetric cross-sectional profiles (Fig. 15h). Consequently, the deposit thickness of the multigranular fans varied by more than 0.03 m at some locations owing to differences in the shift in flow direction (Fig. 15i).

Corresponding to the difference in the runout distance of the solid phase (Fig. 5), the surface slopes along the center of the final fan morphology were different between the monogranular and multigranular flows (Fig. 16). The slopes of the monogranular test runs were similar at ~10° from the flume outlet to ~2 m downstream, but they increased to a maximum of ~15° and became somewhat varied further downstream between experimental runs (Fig. 16a). Similarly, the slopes of the multigranular test runs were similar at ~10° from the flume outlet to ~1.5 m downstream (Fig. 16b). However, the slopes of the multigranular test runs started to increase further upstream in comparison with the monogranular test runs; the slopes increased to a maximum of ~23° from ~1.5 to ~2.2 m downstream (Fig. 16b). These differences, reflected in the averaged slopes, indicate steeper surface slopes of the multigranular-fan morphology (Fig. 16c). Note that beyond 2.5 m downstream, the deposition thickness of the multigranular test runs was close to zero (Fig. 15b), indicating that the slope values do not represent the surface slopes of the final fan morphology. Indeed, in the section from 2.5 to 3.0 m downstream from the flume outlet, the slopes of the multigranular test runs were gentler (i.e., ~6°–8°) and closer to the surface slope of the deposition area (6°) in comparison with the monogranular test runs (Fig. 16c).

Changes in the *SI* values revealed the relevance of the shifting flow direction with regard to the formed fan morphology. After 10 s from the start of flow runout, the *SI* values differed between the test runs, especially at over 1.6 m downstream from the flume outlet, irrespective of the grain-size distribution (Fig. 17a, b). After 20 s from the start of runout of the

355 monogranular flows, *SI* values of >1 were observed in test run 3 (Fig. 17c), whereas such values were observed in another test run (run 4) after 10 s (Fig. 17a). This reflects the avulsion of the monogranular flows that somewhat shifted the flow direction in the zone with a 6° bed slope (Figs. 8 and 17c). After 20 s from the start of runout of the multigranular flows, the range of the *SI* values differed substantially between the various test runs (Fig. 17d). At this stage, depending on differences in the extent of avulsion between the multigranular test runs (Fig. 9), the *SI* values of the monogranular flows were close to

360 1.0 in test runs 2 and 4, but differed substantially from ~0.3 to 2.0 at 2.2 m downstream from the flume outlet between test runs 1 and 3 (Fig. 17d). Therefore, the cross-sectional asymmetry of the fans became increasingly conspicuous owing to the avulsion process of the multigranular flows.

Although the asymmetry of the fan shape became increased by avulsion in the early stage of the formation of fan morphology, after 30 s from the start of flow runout, the range of *SI* values became narrow and close to 1.0 irrespective of

365 the measurement location and grain-size distribution (Fig. 17e, f). With the exception of multigranular test run 3 (Fig. 17e), the *SI* values were in the range of ~0.8–1.3, indicating that both the monogranular and the multigranular flows produced symmetric fan shapes when the flows descended for 30 s (Fig. 17e, f). Because of the variation in flow direction after 40 s from the start of flow runout (Figs. 12 and 13), the range of *SI* values widened among both the monogranular and the multigranular test runs (Fig. 17g, h). The *SI* values of the monogranular flows were approximately 1.0 at all measurement

locations in test run 4, but were greater than 1.0 in test runs 1–3, especially at distal locations from the flume outlet (Fig. 17g). At this stage, the *SI* values of the multigranular flows differed substantially between test runs, ranging between ~0.5 and 1.4 at the maxima, indicating notable avulsion (Fig. 17h). Because of the absence of notable changes in flow direction during the period 40–50 s from the start of flow runout (Figs. S4 and S5), the *SI* values after 50 s were analogous to those after 40 s irrespective of measurement location and grain-size distribution (Fig. 17g–j).

The sequence of the variation in the symmetry of fan shape is reflected in the standard deviation (SD) of the *SI* values between test runs (Fig. 18). For almost all measurement locations and timings, the SD values for the multigranular flows were mostly greater than those of the monogranular flows (Fig. 18). The differences in the SD values between the monogranular and multigranular flows became notable with increase in the distance from the flume outlet, especially in the measurements after 20, 40, and 50 s from the start of flow runout (Fig. 18b, d, e). Importantly, after 30 s from the start of

flow runout, the SD values for both the monogranular and the multigranular flows were mostly <0.2 irrespective of the grain-size distribution (Fig. 18c). It indicates that irrespective of the wide-ranging variations in the direction of flow descent (Figs. 8 and 9) and in the symmetry of the fan shape (Fig. 17c, d), both the monogranular and the multigranular fan morphologies gained analogous fan shapes between test runs (Fig. 18c). Therefore, even if the rate of change in the symmetry of the fan shape was similar, the wide-ranging grain-size distribution within debris flows potentially leads to the

formation of complex fan morphology via increase in the extent of avulsion that shifts the flow direction.

## 4 Discussion

### 4.1 Effects of grain-size distribution on formation of fan morphology

Avulsion occurred in both the monogranular and the multigranular flows but its extent and occurrence location differed owing to differences in grain-size distribution (Figs. 8–13). The runout distance of the fronts of the multigranular flows was shorter than that of the monogranular flows (Fig. 4), which led to change in the occurrence point of the avulsion further toward upper locations during runout and inundation (Fig. 14). Consequently, the multigranular flows shifted their flow directions at locations closer to the outlet of the flume in comparison with the monogranular flows (Figs. 12 and 13). The differences in the extent and location of debris-flow avulsion resulted in different fan morphologies between the monogranular and multigranular flows (Figs. 15–17). Thus, changes in the runout distance attributable to differences in the grain-size distribution of the debris flows were responsible for the variation in fan morphology.

Relatively small and large particles within a debris flow can both influence changes in the runout distance of multigranular flow fronts (De Haas et al., 2015b; Hürlimann et al., 2015). In this study, the decrease in flow resistance due to small sediment particles was intentionally avoided by adjusting the composition of the multigranular flows. Indeed, the arrival time of the flow fronts in the flume was similar between the monogranular and multigranular flows (Fig. 3), suggesting that the effects of small particles on flow resistance were negligible. Unlike the unrelated small sediment particles, large sediment particles were accumulated in the multigranular flow fronts, at least during their runout (Figs. 6 and 7), and potentially caused the decrease in the runout distance (Fig. 5). Large sediment particles increase flow resistance and decrease flow velocity as bed slope decreases (e.g., Egashira et al., 1997; Takahashi, 2007). The velocity of the fronts of the solid phase of the multigranular flows decreased substantially when runout distance exceeded 1 m (i.e., when the front reached the point at which the bed slope decreased from 12° to 9°) in comparison with that of the monogranular flows (Fig. 4), suggesting that large particles caused a decrease in flow velocity. Corresponding to the short runout distance (Fig. 5) and the point of occurrence of avulsion close to the flume outlet (Fig. 14), the multigranular flows resulted in steeper surface slopes in comparison with the monogranular flows (Fig. 16), which supports the suggestion that the multigranular flows were affected by a stronger shear resistant force during runout and inundation. Thus, even when debris flows have hydrographs that are similar at the outlet of the channel, differences in the extent of accumulation of large particles in the flow front can lead to changes in runout distance and consequently form fans with different morphology.

Separation between the solid and fluid phases might be one of the principal mechanisms that alter runout distance. The fluid phase consisting of pore fluid in a multiphase-mixture flow generally acts to reduce flow resistance and drive flow descent (Takahashi, 2007; von Boetticher et al., 2016; Pudasaini and Mergili, 2019). The substantial decrease in the velocity of the front of the solid phase of the multigranular flows progressed phase separation during flow runout in the early stage (Fig. 4b), which increasingly can reduce the velocity of the solid phase owing to the absence of pore fluid. Numerical simulations based on the mechanical model that considered phase separation demonstrated that a strong front structure attributable to accumulation of solids in the flow front can lead to rapid phase separation (Pudasaini and Fischer, 2020). Thus, the large

sediment particles that accumulated at the flow front of the multigranular flows potentially advanced phase separation during

flow runout. Therefore, the increase in flow resistance of the multigranular flow fronts could have arisen owing to synergistic effects between the increase in the representative grain size of the solid phase and the decrease in the pore fluid by phase separation.

It is noteworthy that the fronts of the solid phase of the multigranular flows continued their runout after the start of phase separation (Fig. 4b), which is different from the coincidental start of phase separation and halting of the front of the solid

phase in the monogranular flows (Figs. 4a and 5). The solid-phase runout after the start of phase separation in the multigranular flows implies that the solid phase retained sufficient momentum to entrain and transport sediment particles. In this sense, deposition of the solid phase of the monogranular flow fronts was caused by the complete and sudden stop of the solid phase (i.e., sediment particles), which might be physically different from that of the multigranular flow fronts. Theoretical analysis of debris-flow mechanics that carefully divides the effects between the erosion velocity (i.e., the

velocity of sediment erosion from the bed by the flow) and the entrainment velocity (i.e., the velocity of the transportation of eroded sediment by the flow) demonstrated that the contribution to flow momentum is different between the erosion and the entrainment velocities; consequently, their differences are responsible for the state of the erosion-induced excess energy and the mobility of debris flows (Pudasaini and Krautblatter, 2021). Detailed analysis of the difference between the erosion (deposition) and entrainment velocities is difficult owing to limitations of the experimental setup. However, the different

trends in runout between the monogranular and multigranular flows highlight that further understanding of the erosion/deposition mechanisms is crucial for accurate estimation of debris-flow deposition range that can be achieved with the mechanical erosion and mobility model (Pudasaini and Krautblatter, 2021).

## 4.2 Variations in fan morphology

In comparison with the monogranular flows, the multigranular flows formed fans with reasonably asymmetric morphology

(Fig. 17), which resulted from the avulsion process that caused marked shifts in flow direction (Fig. 13). Despite differences in the extent of avulsion between the monogranular and multigranular flows, the extent of symmetry in fan morphology increased at the same timings (Figs. 17 and 18), suggesting that the pace at which avulsion occurred was similar irrespective of the grain-size distribution of the debris flow. The wide-ranging grain-size distribution within debris flows thus leads to marked shifts of flow direction by avulsion rather than to changes in the pace of avulsion, and likely expands the horizontal

deposition range of the sediment.

Importantly, in comparison with monogranular fans, the extent of asymmetry of the multigranular fans differed substantially between test runs (Figs. 17 and 18). The variations in the asymmetry of the multigranular fans suggest that debris flows with wide-ranging grain-size distribution can randomly shift their descent direction when the flows behave as unsteady flows that are freed from horizontal constraints owing to the channel-like topography. Some models assume that multigranular debris

flows can be approximated to monogranular debris flows with the same average grain size (e.g., Egashira et al., 1997; Takahashi, 2007). Despite this assumption, such models allow estimation of debris-flow properties such as flow velocity and

depth, especially under a steady flow state (Egashira et al., 1997; Takahashi, 2007). Indeed, in the flume, experimental results exhibited similar flow velocity and depth as debris flows with the same average grain size but with different grain-size distributions (Fig. 3). However, given that natural debris flows generally consist of wide-ranging grain-size sediment particles (e.g., Zanuttigh and Lamberti, 2007; Johnson et al., 2012), the use of debris-flow models that involve grain-size approximation could result in errors in the estimated runout distance of debris flows owing to unsteady behavior during flow runout. This likely leads to inevitable uncertainty in the estimation of fan morphology formed by debris-flow runout.

Even in the early stage of flow runout, i.e., after 20 s from the start of runout, the shape of the multigranular fans exhibited asymmetry in comparison with that of the monogranular fans (Fig. 17c, d), which was likely responsible for greater final asymmetry in multigranular fan morphology (Fig. 17i, j). It is likely that the short runout distance of the multigranular flows resulted in thick and steep sediment deposition close to the flume outlet, and the swift phase separation accelerated the inundation of the fluid phase to the distal downstream area. In this sense, phase separation facilitated the increase in the extent of unsaturation of the fan deposits. A bed consisting of unsaturated sediment particles potentially decreases the pore-fluid pressure at the bottom of a debris flow and increases the resistance of the flow body (Major and Iverson, 1999; Staley et al., 2011), resulting in complex patterns of flow direction and sediment deposition (Tsunetaka et al., 2019). Thus, the variations in the extent of the saturation of the fan sediment materials facilitated by phase separation might have triggered the differences in the fan morphology between the multigranular test runs.

In this context, the extent of phase separation broadly constrains fan morphology. The advance in the multiphase model describing a granular–fluid mixture flow allows us to reflect on the effects of separation between the granular (solid) and fluid phases in numerical simulations (Pudasaini and Mergili, 2019), and to progress the theoretical interpretation of debris-flow mechanics. Our results demonstrate that further investigation of the relationships between the grain-size distribution within debris flows and the extent of phase separation and related changes in runout distance could lead to accurate forecasting of the range of debris flow deposition and inundation.

## 4.3 Implications for natural debris-flow fans

It should be noted that the flume tests conducted in this study were operated under limited conditions that considered only two types of grain-size distribution. Therefore, the extent to which the obtained experimental results represent the properties of natural debris-flow fans should be assessed with caution. The observations of the grain-size profiles of the multigranular flows (Figs. 6 and 7) indicate that the grain-size segregation of the sediment particles was similar to that of natural debris flows (e.g., Iverson, 1997; Zanuttigh and Lamberti, 2007; Johnson et al., 2012). Additionally, the wide-ranging grain-size distribution of the debris flows caused horizontal widening of the deposition range (Fig. 15). This relationship between horizontal deposition characteristics and grain-size distribution is also observed in stratigraphic records of natural debris-flow fans (Pederson et al., 2015). Thus, in terms of qualitative observations, our flume tests can be considered representative to a certain extent of the properties of natural debris-flow fans.

In terms of a geometrical scaling relationship, we compared the relationships between the volumes of the debris flows ($V$) and the inundation areas of the sediment deposits ($A$) similar to De Haas et al. (2015b) and Baselt et al. (2022) (Fig. 19). The inundation areas of the monogranular and multigranular test runs were ~2.224 m$^2$ and ~2.159 m$^2$, respectively (Table S1), which highlights that when the hydrograph and velocity of debris flows are similar before the start of runout, the effects of grain-size distribution within the debris flows on fan formation are reflected in change in the horizontal shape of the sediment inundation range but without substantial variation in the gross area. Owing to this similarity in the inundation area regardless of the grain-size distribution, all experimental runs were plotted in almost identical locations on the log-log $V$-$A$ plane, and just below the best-fit regression curve for natural debris flows derived by Griswold and Iverson (2008) (Fig. 19). The $V$-$A$ scaling relationships indicate that our experimental results are geometrically within the range of natural debris flows, and that our flume tests were well-controlled experiments across all experimental runs, especially regarding the resultant inundation areas. Given this reproducibility of the inundation area, although we performed only four test runs for both the monogranular flows and the multigranular flows, we believe that the obtained observations adequately reflect the representative behavior of experimental debris flows under the operation and setup of our flume tests.

In addition to these qualitative and geometric similarities between the experimental and natural debris flows, the similar flow depths suggest that the experimental debris flows were well-controlled in terms of their hydrographs, at least in the flume (Fig. 3). However, some dynamic properties, such as flow resistance (Egashira et al., 2001), sediment erosion and entrainment rate (McCoy et al., 2012; De Haas et al., 2022), and flow friction (Pudasaini and Miller, 2013; Lucas et al., 2014), are strongly governed by the scales of grain size and flow volume. Thus, especially for the experimental debris flows after their runout, our flume tests might not have completely met the dynamic similarity law of debris flows, similar to many other reduced-scale flume tests (e.g., De Haas et al., 2015b; Iverson, 2015). In this regard, our flume tests focus on a limited aspect of the effects of the grain-size distribution within debris flows. Although the effects of fine sediment (e.g., silt and clay) were intentionally excluded in our experiments to control the hydrograph and the velocity of the debris flows in the flume, fine sediment might alter the resistance and stress structure of natural debris flows (Kaitna et al., 2016; Sakai et al., 2019; Nishiguchi and Uchida, 2022). Because these changes in the resistance and stress of debris flows might affect the rate of separation between the solid and fluid phases (Pudasaini and Fischer, 2020; Baselt et al., 2022), our flume tests could not identify the extent of phase separation on the scale of natural debris flows that comprise wide-ranging sediment particle size from silt to large boulders. In nature, various factors (e.g., phase separation) associated with particle size and grain-size distribution interact, and therefore the behavior of debris flows becomes increasingly complex (e.g., De Haas et al., 2018b). This is reflected in the wide-ranging variation in the $V$-$A$ relationship of natural debris flows (Fig. 19). Our flume tests demonstrate that differences in the grain-size distribution within debris flows can change fan morphology, and likely support interpretation of the formation processes of fan morphology resulting from a single successive debris-flow surge. However, comprehensive assessment of the extent of the respective effects in relation to grain-size distribution within natural debris flows will require further accumulated field data.

## 5 Conclusions

In this study, we conducted flume-based experiments to investigate how differences in the grain-size distribution within debris flows change the morphology of debris-flow fans. Two types of sediment particles were used to generate two types of granular–water mixture flows that imitated a single debris-flow surge: monogranular particles comprising quasi-monodispersed sediment particles and multigranular particles comprising polydispersed sediment particles. The granular–water mixture flows generated using the monogranular particles and the multigranular particles were referred to as monogranular flows and multigranular flows, respectively. The average grain size was adjusted to coincide between the monogranular and multigranular flows, which allowed us to compare the fan morphologies formed by debris flows that had similar debris mixture hydrographs but different grain-size distributions.

Despite similarities in the debris mixture hydrographs before the start of debris-flow runout, the runout distance of the fronts of the multigranular flows was less than that of the monogranular flows, which was likely attributable both to accumulation of relatively large sediment particles, and to the swift separation between the solid and fluid phases of the multigranular flows during runout. The short runout distances of the multigranular flows were responsible for changes in the location at which the avulsion occurred, which led to avulsion that markedly shifted the flow direction during fan formation. Consequently, in comparison with the monogranular fans, the fans of the multigranular flows formed with horizontally asymmetric shapes, highlighting that fan morphology can vary in response to grain-size distribution within a debris flow.

The extent of the symmetry of debris-flow fan morphology increased at a similar time during debris-flow runout irrespective of grain-size distribution and test runs. However, avulsion that shifted the flow direction increased the extent of asymmetry of fan morphology, and also increased the morphological deviations between test runs, especially for the multigranular flows. Therefore, wide-ranging grain-size distribution within a debris flow rather than change in the rate of fan formation likely results in complex fan morphology with high asymmetry. Our results suggest that further understanding of the relationships between differences in grain-size distribution and runout of debris flows could reduce uncertainty in the estimation and interpretation of debris-flow fan morphology.

## Data availability

The data used in this study are freely available from the corresponding author upon request.

## Author contribution

HT designed the study, carried out the flume tests and all analyses, and wrote the paper. NH and YS supported the flume tests, provided input for the interpretation of the results, and reviewed and edited the paper. TW was involved in conceptualizing the study, shaping the methodology and discussion, and writing the paper.

## Acknowledgement

The authors are grateful to the staff of CTI Engineering Co., Ltd. for their assistance with the flume tests. The authors would like to thank Ivo Baselt for sharing data concerning debris-flow volume and deposition area. The authors also would like to thank Shiva P. Pudasaini and an anonymous reviewer for their constructive comments and suggestions, which improved the paper. The authors thank James Buxton, MSc, from Edanz (https://jp.edanz.com/ac) for editing a draft of this manuscript.

## Conflicts of Interest

The authors declare that they have no conflict of interest.

## Funding

The research was supported by JSPS KAKENHI (grant numbers 18J01961 and 19KK0392).

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

**Table 1:** Details of sediment particles and mixing ratios.

| Range of grain size (mm) | Average grain size (mm) | Mixing ratio of monogranular particles (%) | Mixing ratio of multigranular particles (%) |
|---|---|---|---|
| 0.6–0.8 | 0.7 | 0 | 10 |
| 0.8–1.36 | 1.1 | 0 | 12.5 |
| 1.36–2.02 | 1.7 | 0 | 15 |
| 2.02–3.24 | 2.6 | 100 | 25 |
| 3.24–4.57 | 3.9 | 0 | 15 |
| 4.57–5.85 | 5.2 | 0 | 12.5 |
| 5.85–7.5 | 6.7 | 0 | 10 |



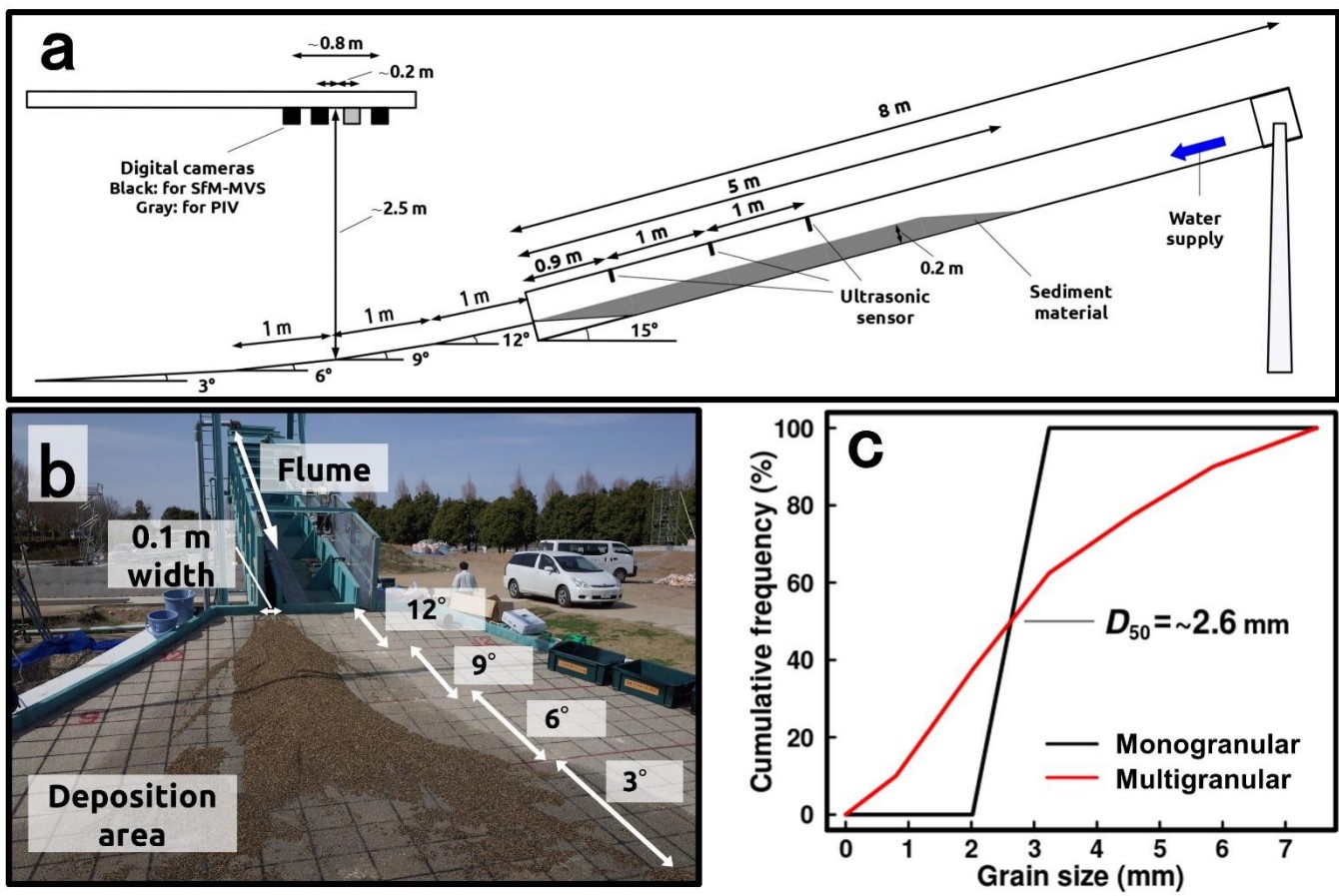


**Figure 1:** Test flume setup. (a) Dimensions of the test flume and equipment. (b) View of the flume and the deposition area. (c) Grain-size distribution of the sediment materials used in the experiments. Figure modified from Tsunetaka et al. (2019).

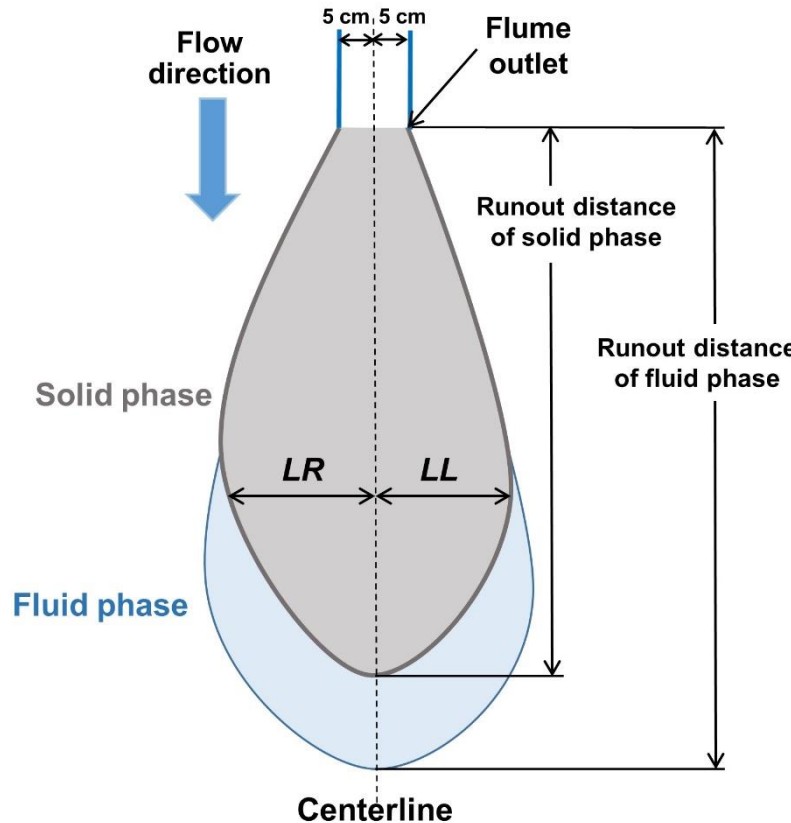


**Figure 2:** Sketch showing definitions of measurements associated with the flume experiments. The centerline is drawn as an extension of the central longitudinal axis of the flume.

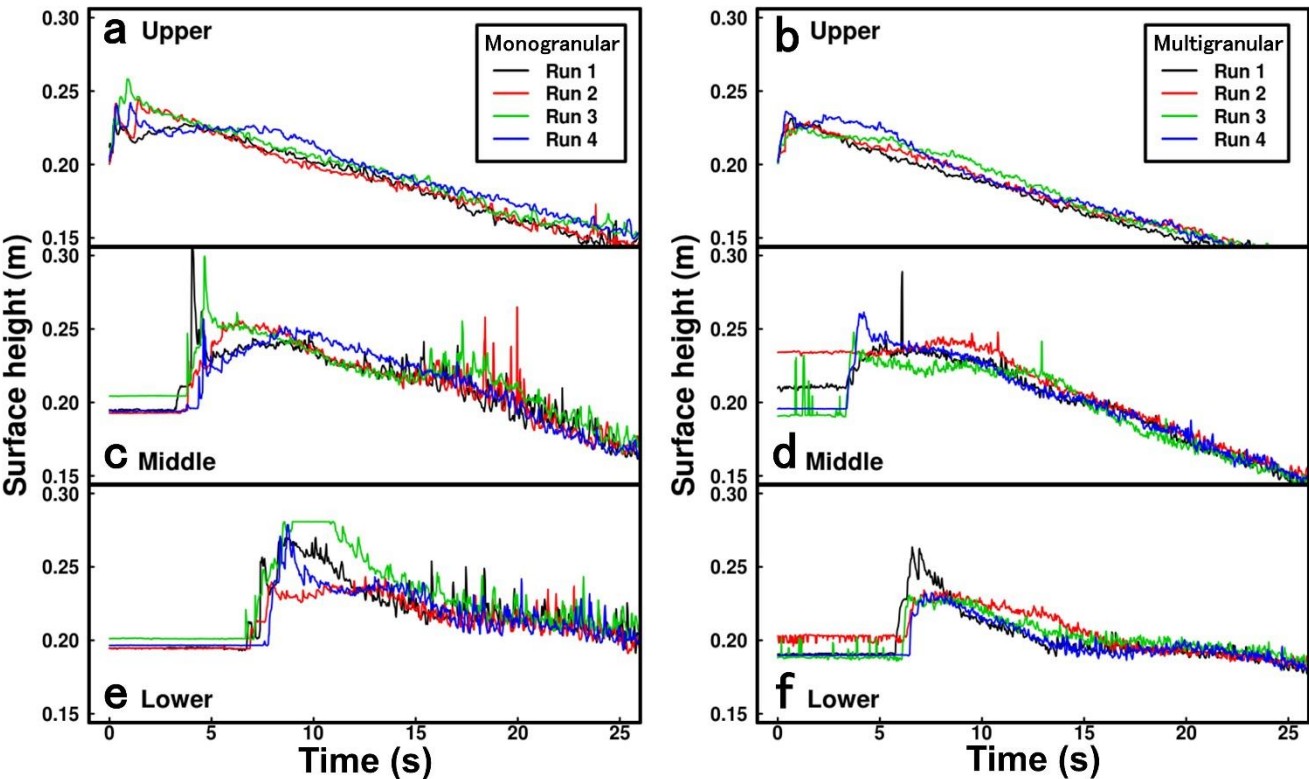

**Figure 3:** Changes in the debris-flow surface in the flume. The left and right panels show results of the monogranular and multigranular flows, respectively. The different colors of the lines correspond to respective experimental runs: (a)and (b) measurements at the upper measurement position, (c) and (d) measurements at the middle measurement position, and (e) and (f) measurements at the lower measurement position. The time (x-axis) was set to assume that the flow front arrived at the upper measurement position at time zero. Any change in the initial thickness of the bed surface (e.g., Fig. 3d) due to local undulation was probably canceled out by the debris-flow descent.

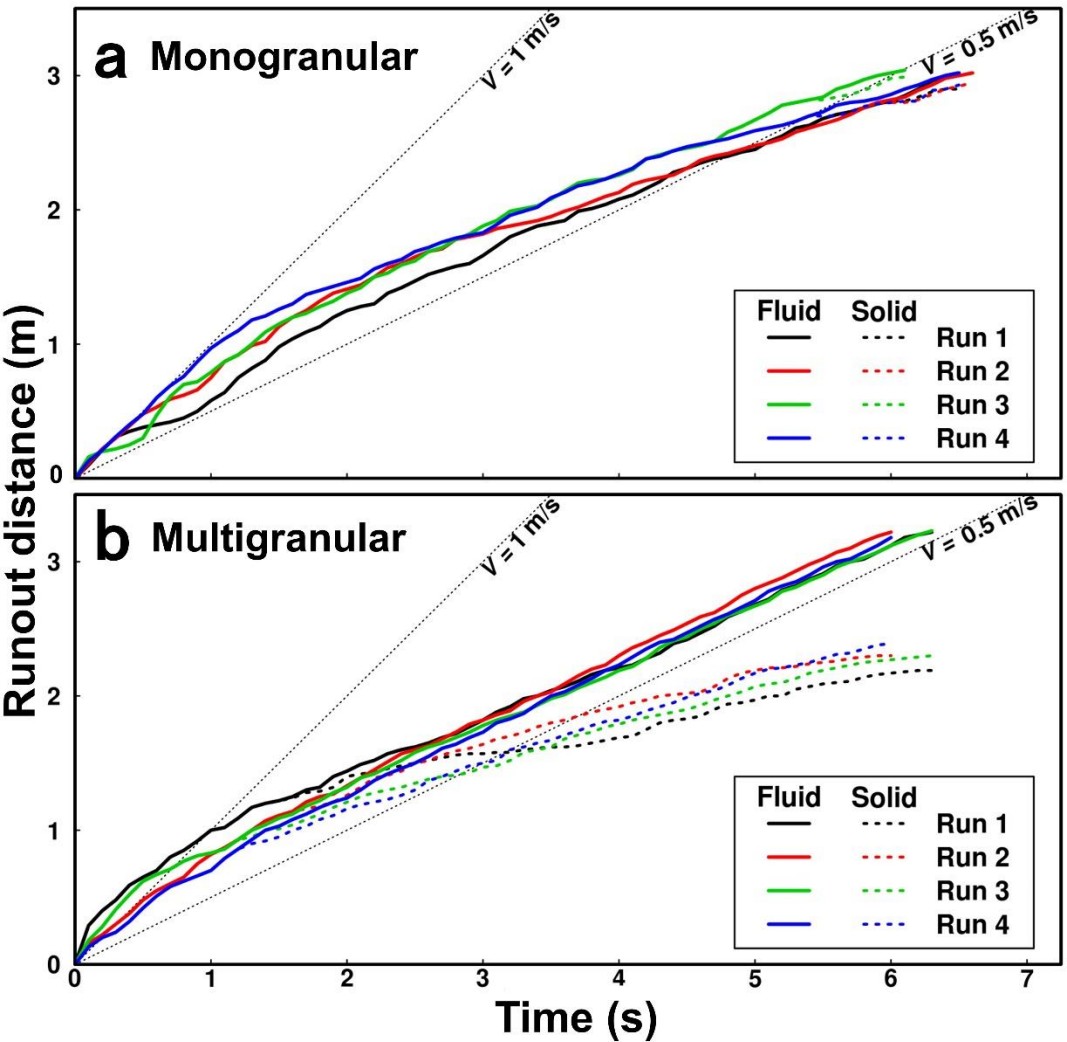

**Figure 4:** Change in runout distances of the flow fronts with time: (a) monogranular flows and (b) multigranular flows. Continuous and broken lines indicate runout distances for the fluid and solid phases, respectively. Black dotted lines are assumed graphs for velocities of 0.5 and 1 m s$^{-1}$.

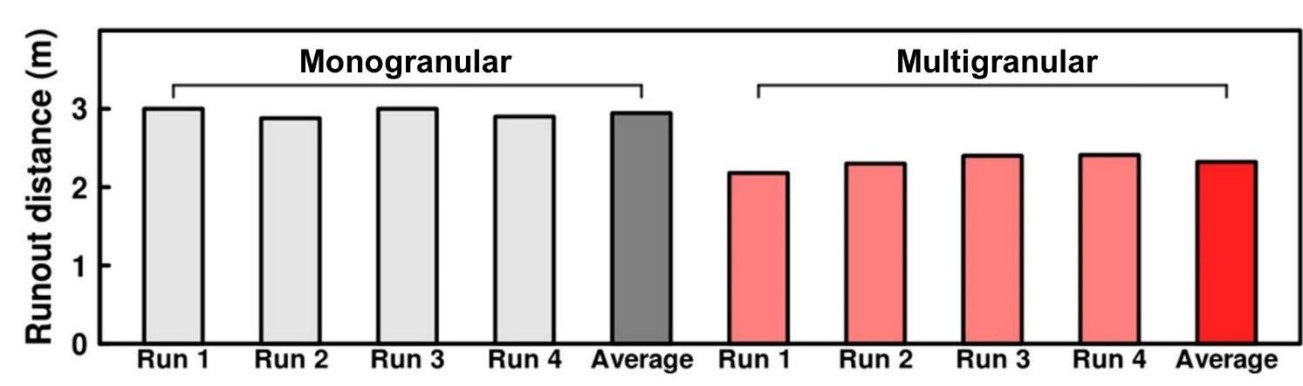

**Figure 5:** Comparison of the total runout distance of the fronts of the solid phase of monogranular and multigranular flows. The locations of the fronts of the solid phase are depicted by white lines in Fig. S3.

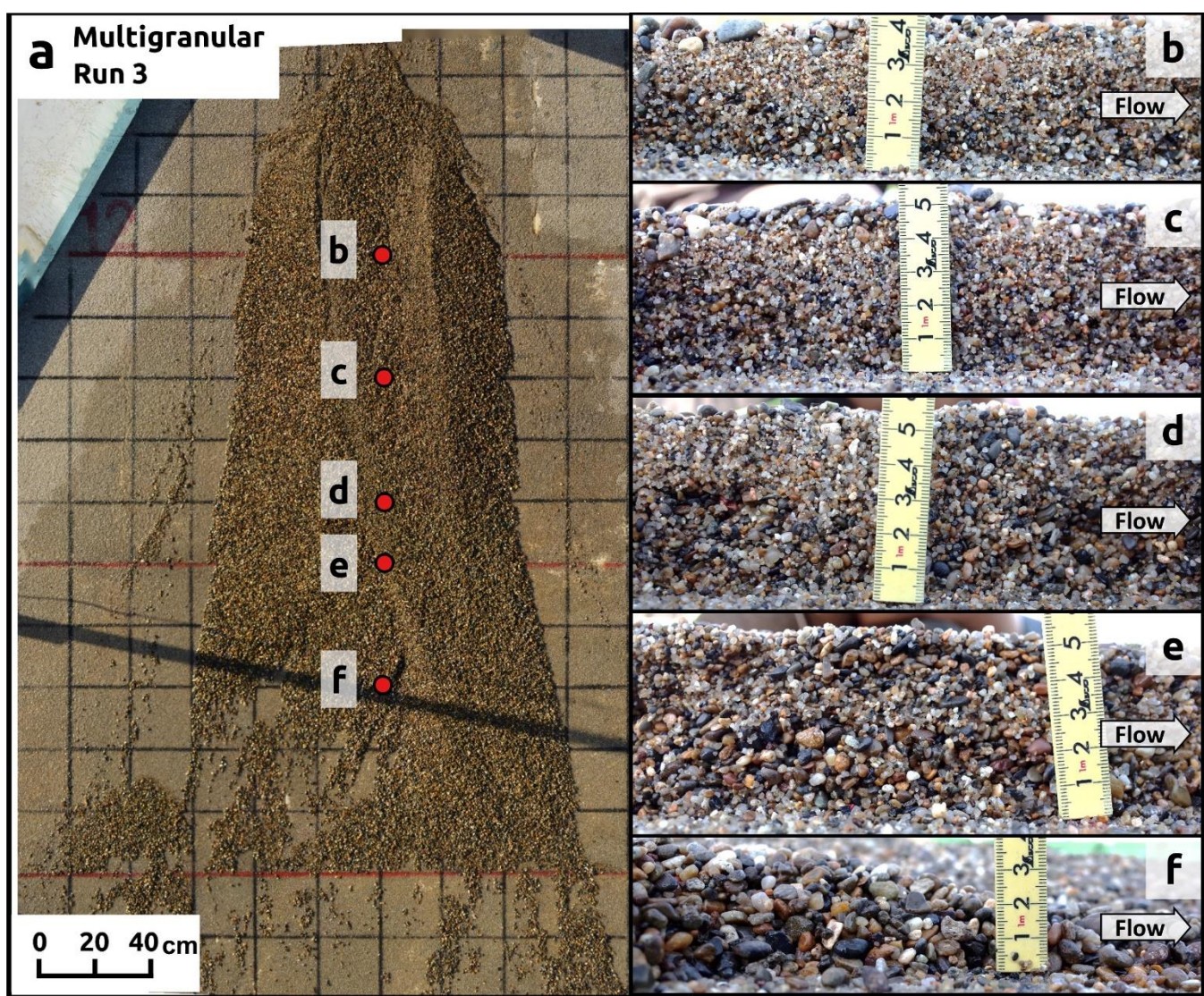

**Figure 6:** (a) Orthophoto of the debris-flow fan formed by the multigranular flow in test run 3. The red circles indicate the points at which the images were taken. Images of the longitudinal profile from the right-bank side view: (b) 1 m downstream from the flume outlet (slope change point from 12° to 9°), (c) 1.4 m downstream from the flume outlet, (d) 1.8 m downstream from the flume outlet, (e) 2 m downstream from the flume outlet (slope change point from 9° to 6°), and (f) 2.4 m downstream from the flume outlet.


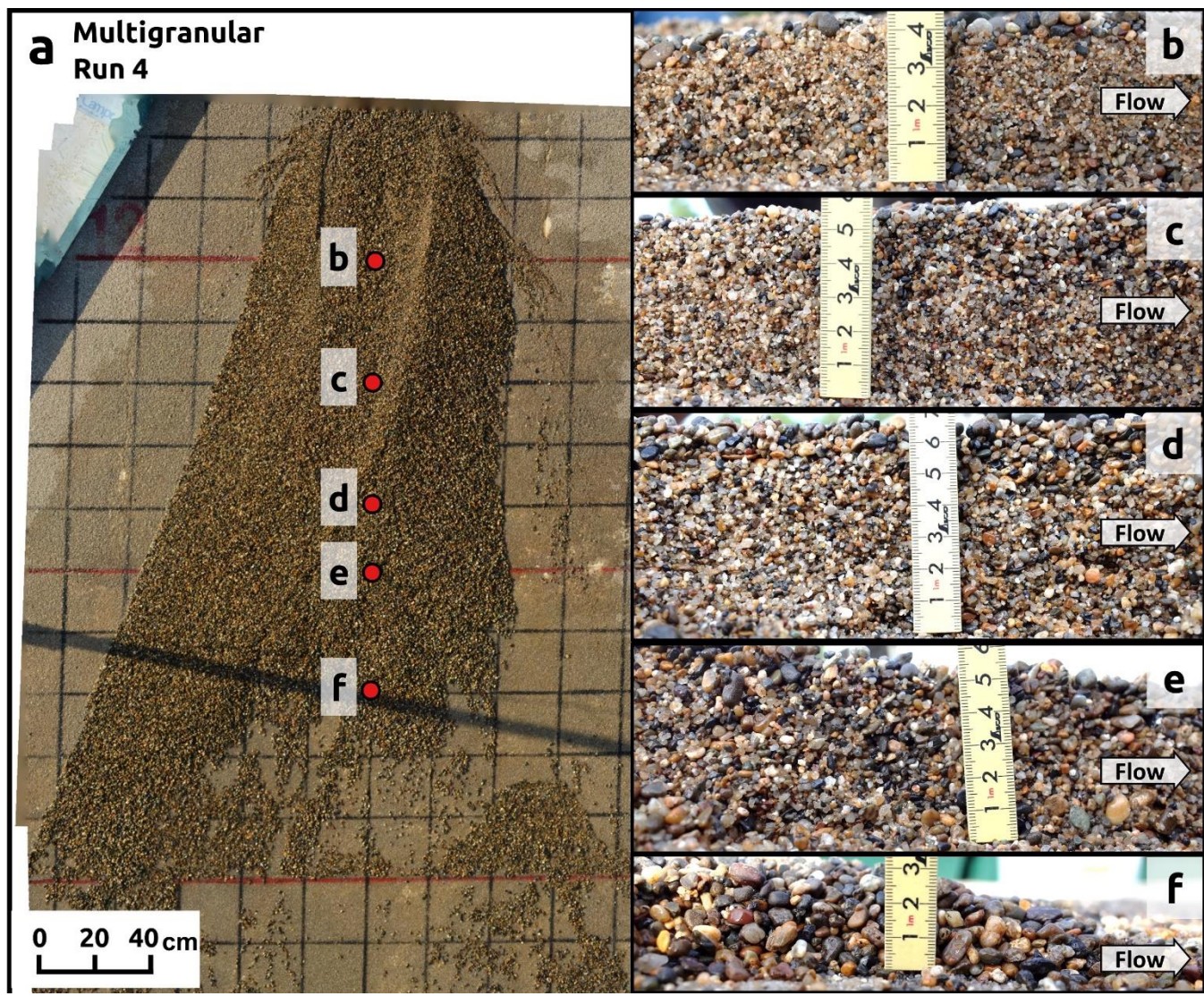

**Figure 7:** (a) Orthophoto of the debris-flow fan formed by the multigranular flow in test run 4. The red circles indicate the points at which the images were taken. Images of the longitudinal profile from the right-bank side view: (b) 1 m downstream from the flume outlet (slope change point from 12° to 9°), (c) 1.4 m downstream from the flume outlet, (d) 1.8 m downstream from the flume outlet, (e) 2 m downstream from the flume outlet (slope change point from 9° to 6°), and (f) 2.4 m downstream from the flume outlet.



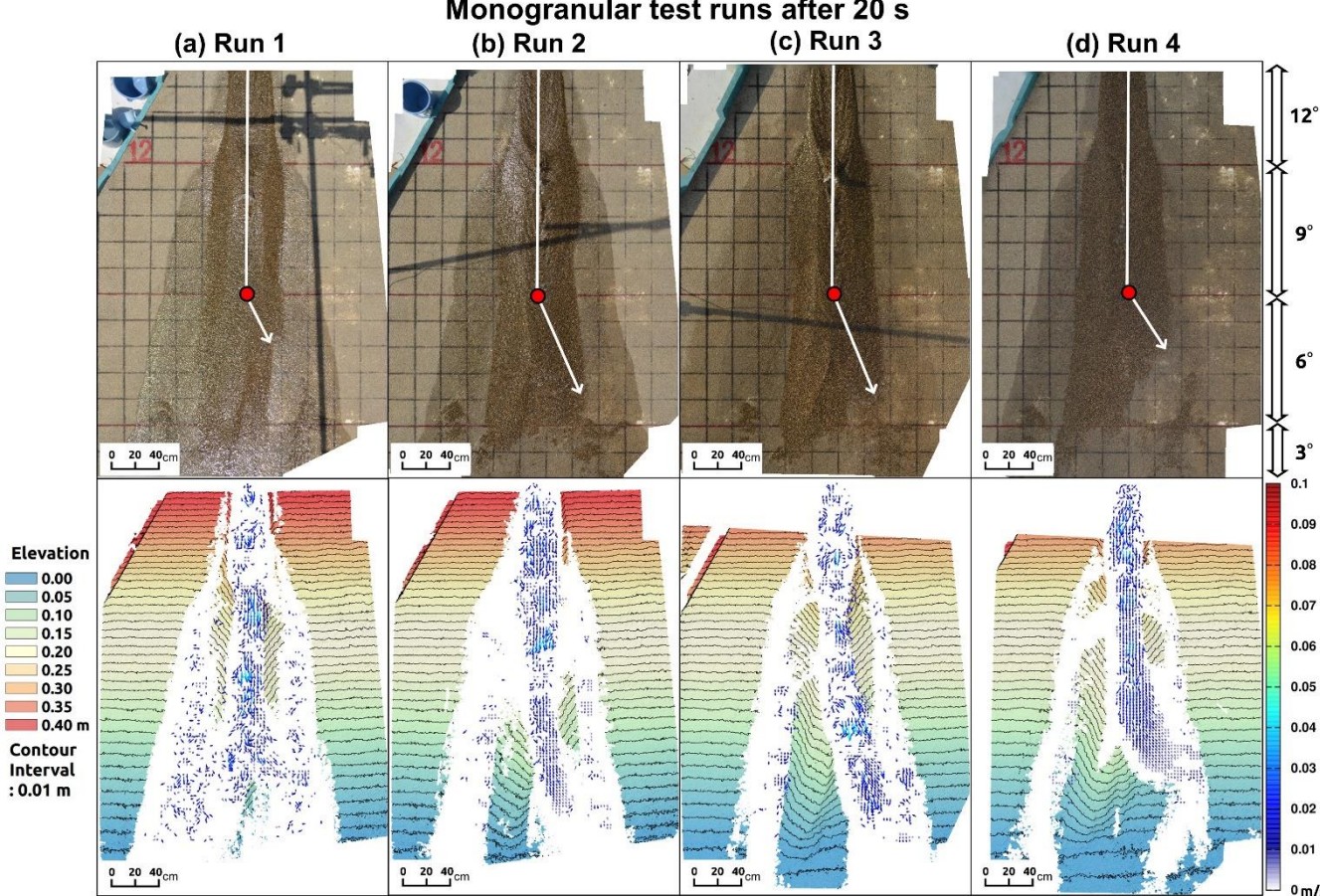

**Figure 8:** Fan morphology 20 s after the start of runout of the monogranular flows. The upper and lower panels show orthophotos and digital elevation models (DEMs) with flow vectors, respectively. Respective sets of the upper orthophoto and lower DEM represent corresponding results of each experimental test run. The white arrows on the orthophotos indicate the assumed principal direction of flow descent. The red points in the orthophotos indicate the assumed occurrence point of the avulsion. The elevation of the DEMs is depicted assuming that the area with a 3° slope (i.e., the area furthest downstream from where the slope angle changed from a 6° to 3° slope) has elevation of zero.


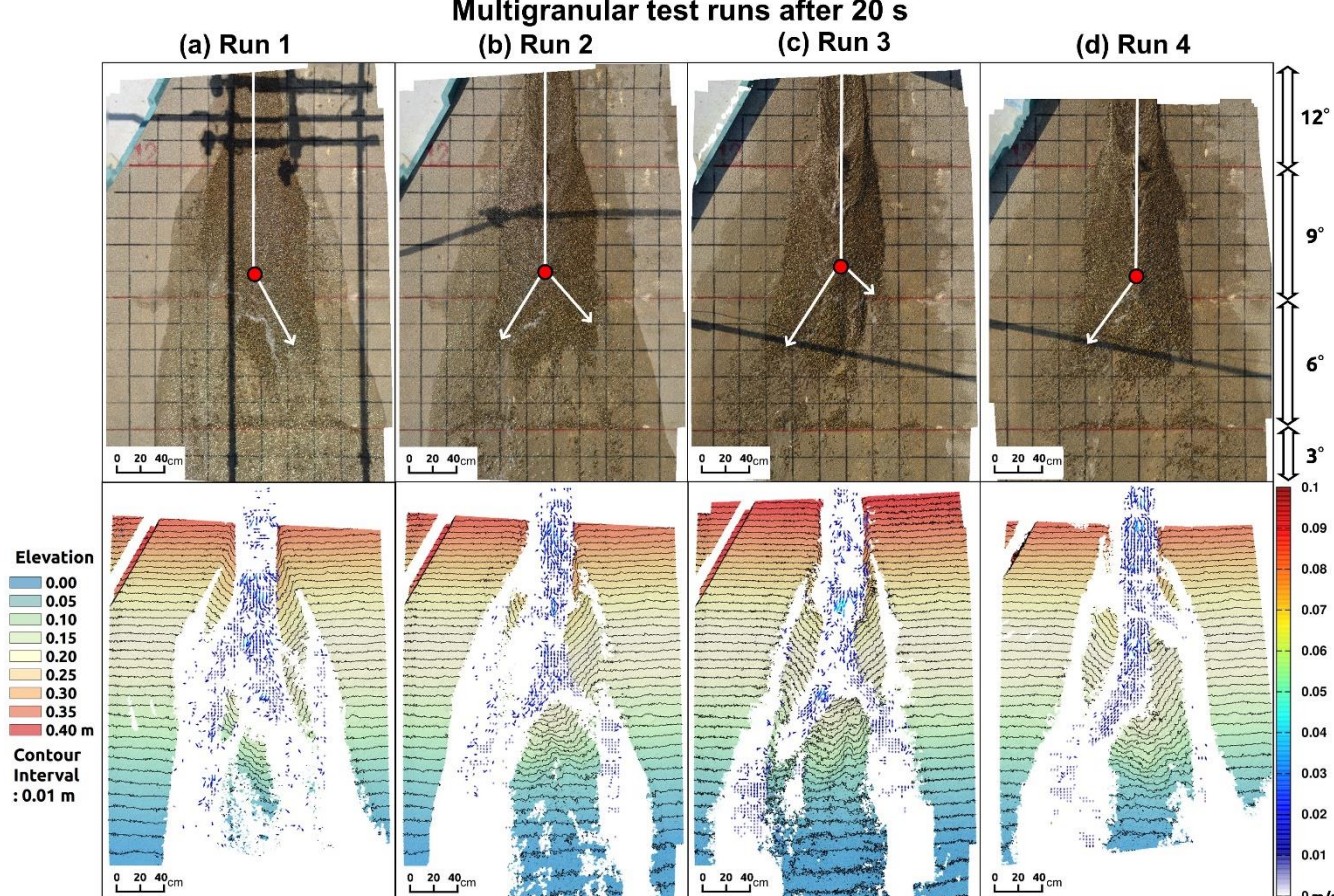

**Figure 9:** Fan morphology 20 s after the start of runout of the multigranular flows. The upper and lower panels show orthophotos and digital elevation models (DEMs) with flow vectors, respectively. Respective sets of the upper orthophoto and lower DEM represent corresponding results of each experimental test run. The white arrows on the orthophotos indicate the assumed principal direction of flow descent. The red points in the orthophotos indicate the assumed occurrence point of the avulsion. The elevation of the DEMs is depicted assuming that the area with a 3° slope (i.e., the area furthest downstream from where the slope angle changed from a 6° to 3° slope) has elevation of zero.


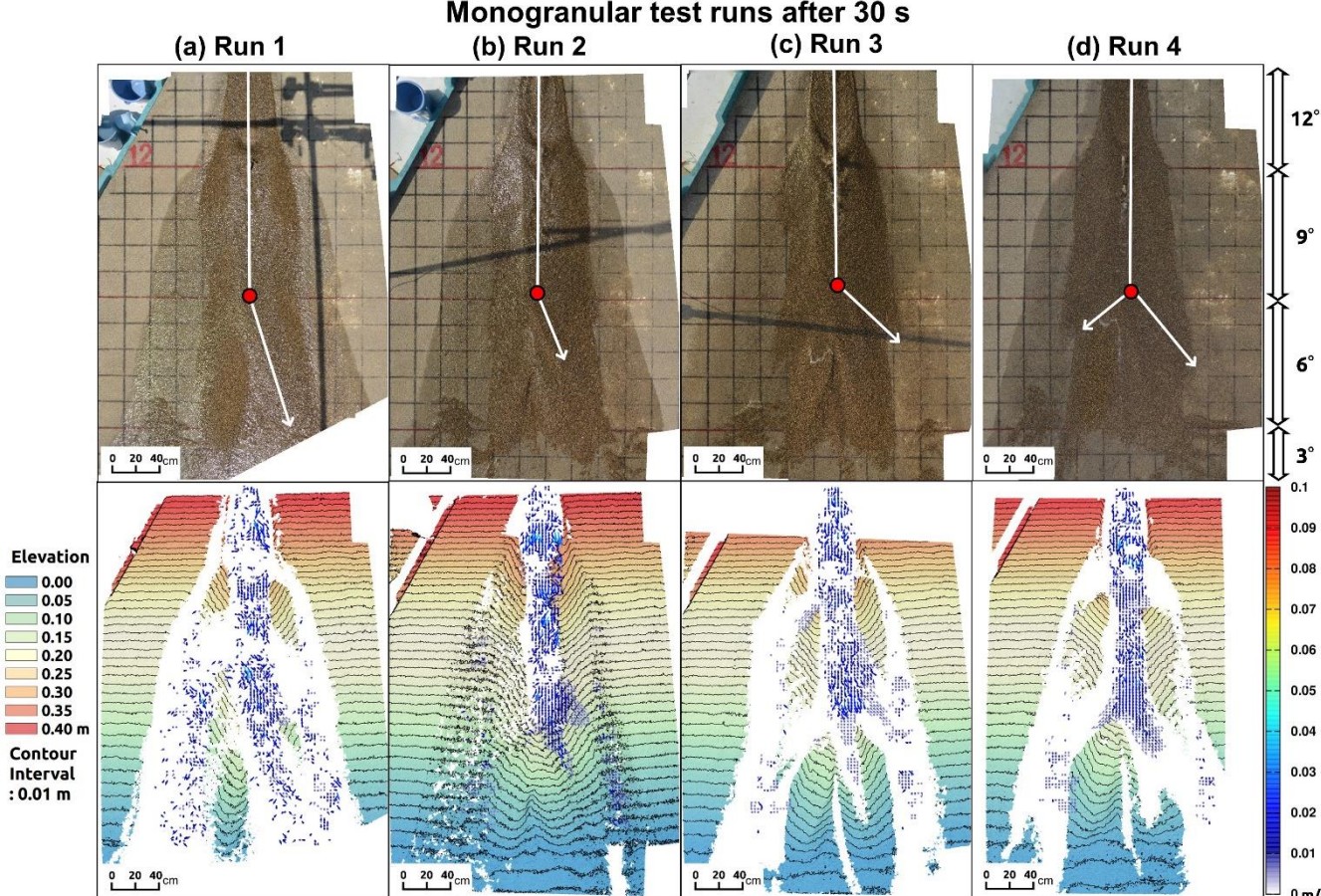

**Figure 10:** Fan morphology 30 s after the start of runout of the monogranular flows. The upper and lower panels show orthophotos and digital elevation models (DEMs) with flow vectors, respectively. Respective sets of the upper orthophoto and lower DEM represent corresponding results of each experimental test run. The white arrows on the orthophotos indicate the assumed principal direction of flow descent. The red points in the orthophotos indicate the assumed occurrence point of the avulsion. The elevation of the DEMs is depicted assuming that the area with a 3° slope (i.e., the area furthest downstream from where the slope angle changed from a 6° to 3° slope) has elevation of zero.

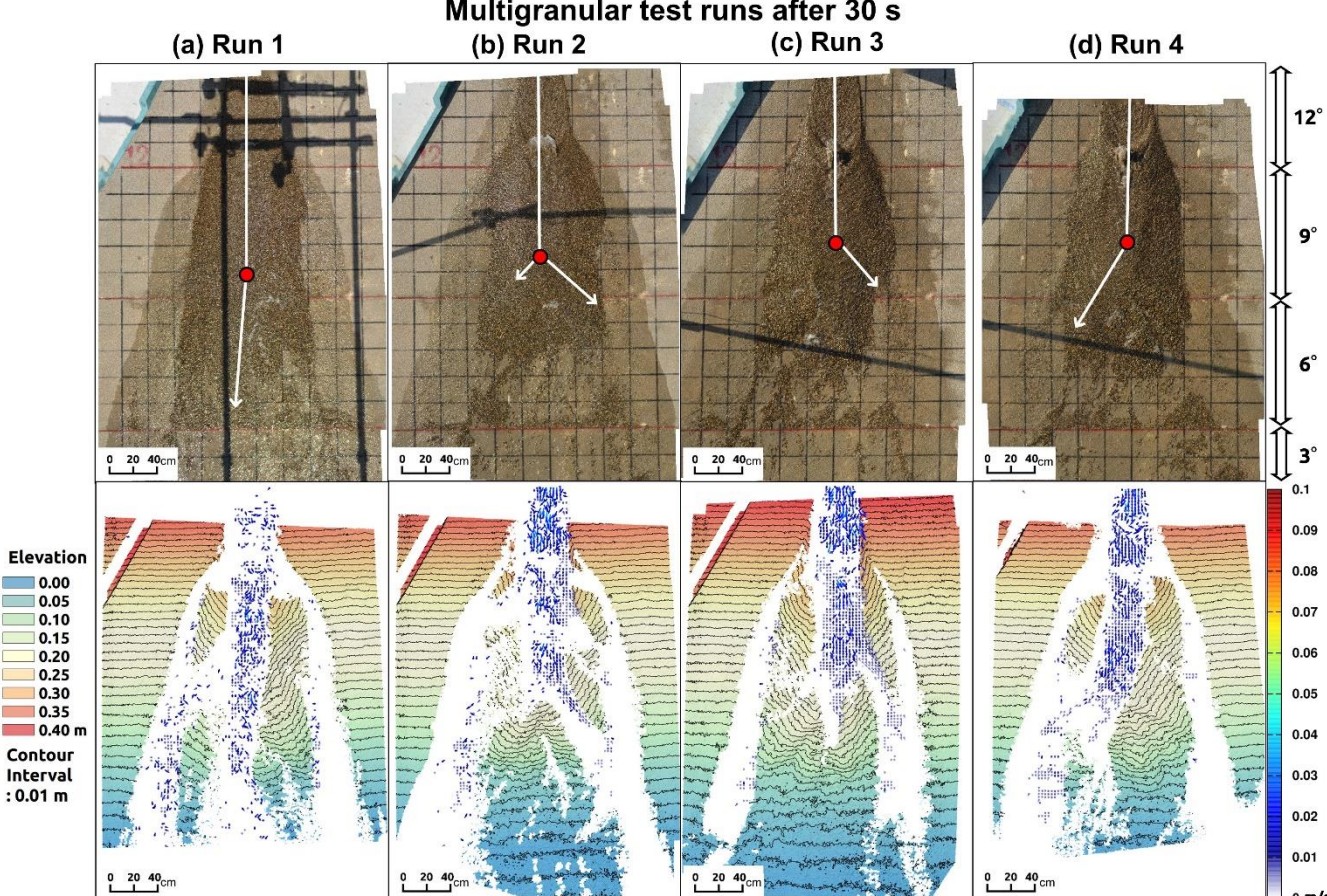

**Figure 11**: Fan morphology 30 s after the start of runout of the multigranular flows. The upper and lower panels show orthophotos and digital elevation models (DEMs) with flow vectors, respectively. Respective sets of the upper orthophoto and lower DEM represent corresponding results of each experimental test run. The white arrows on the orthophotos indicate the assumed principal direction of flow descent. The red points in the orthophotos indicate the assumed occurrence point of the avulsion. The elevation of the DEMs is depicted assuming that the area with a 3° slope (i.e., the area furthest downstream from where the slope angle changed from a 6° to 3° slope) has elevation of zero.



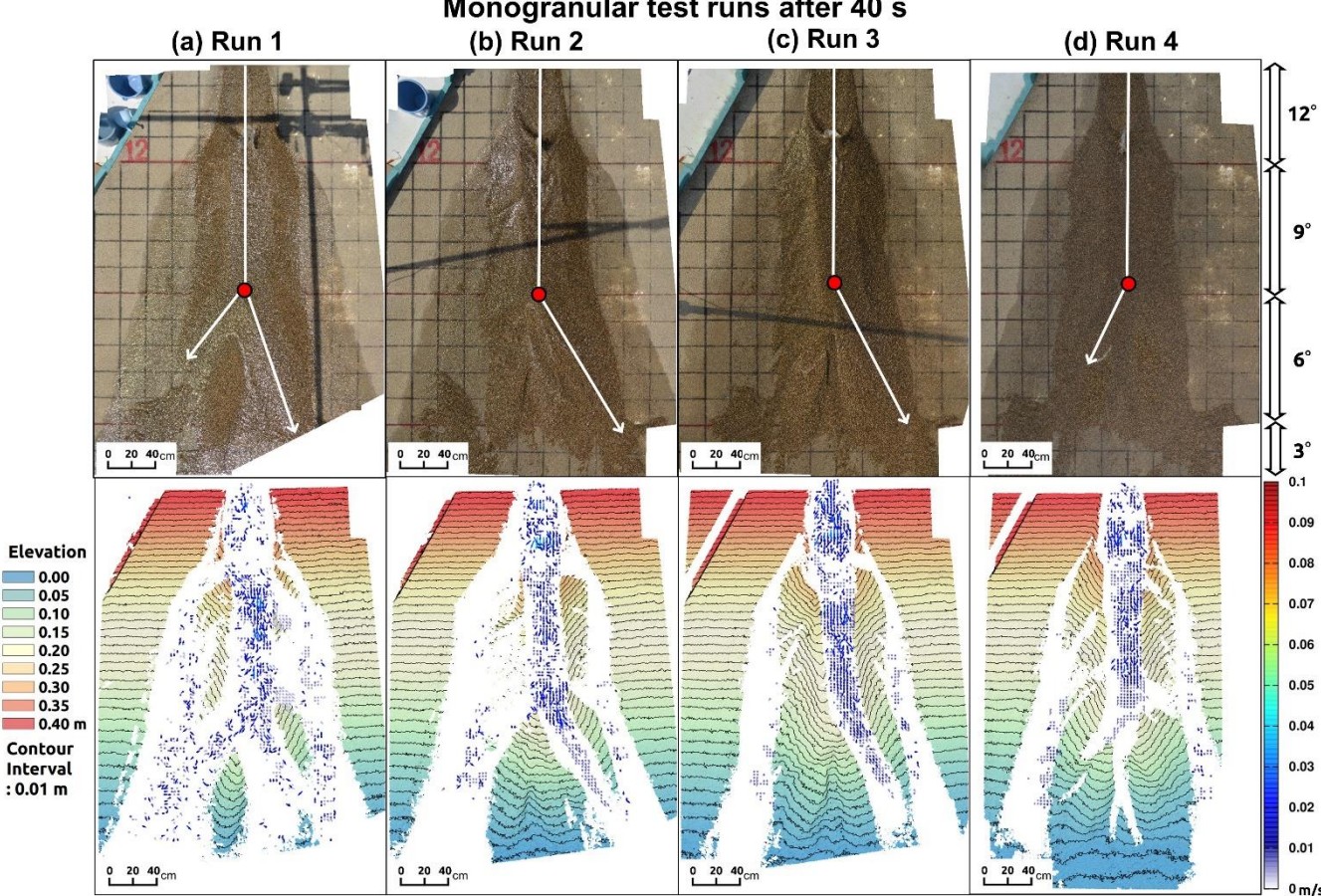

**Figure 12**: Fan morphology 40 s after the start of runout of the monogranular flows. The upper and lower panels show orthophotos and digital elevation models (DEMs) with flow vectors, respectively. Respective sets of the upper orthophoto and lower DEM represent corresponding results of each experimental test run. The white arrows on the orthophotos indicate the assumed principal direction of flow descent. The red points in the orthophotos indicate the assumed occurrence point of the avulsion. The elevation of the DEMs is depicted assuming that the area with a 3° slope (i.e., the area furthest downstream from where the slope angle changed from a 6° to 3° slope) has elevation of zero.

**Multigranular test runs after 40 s**

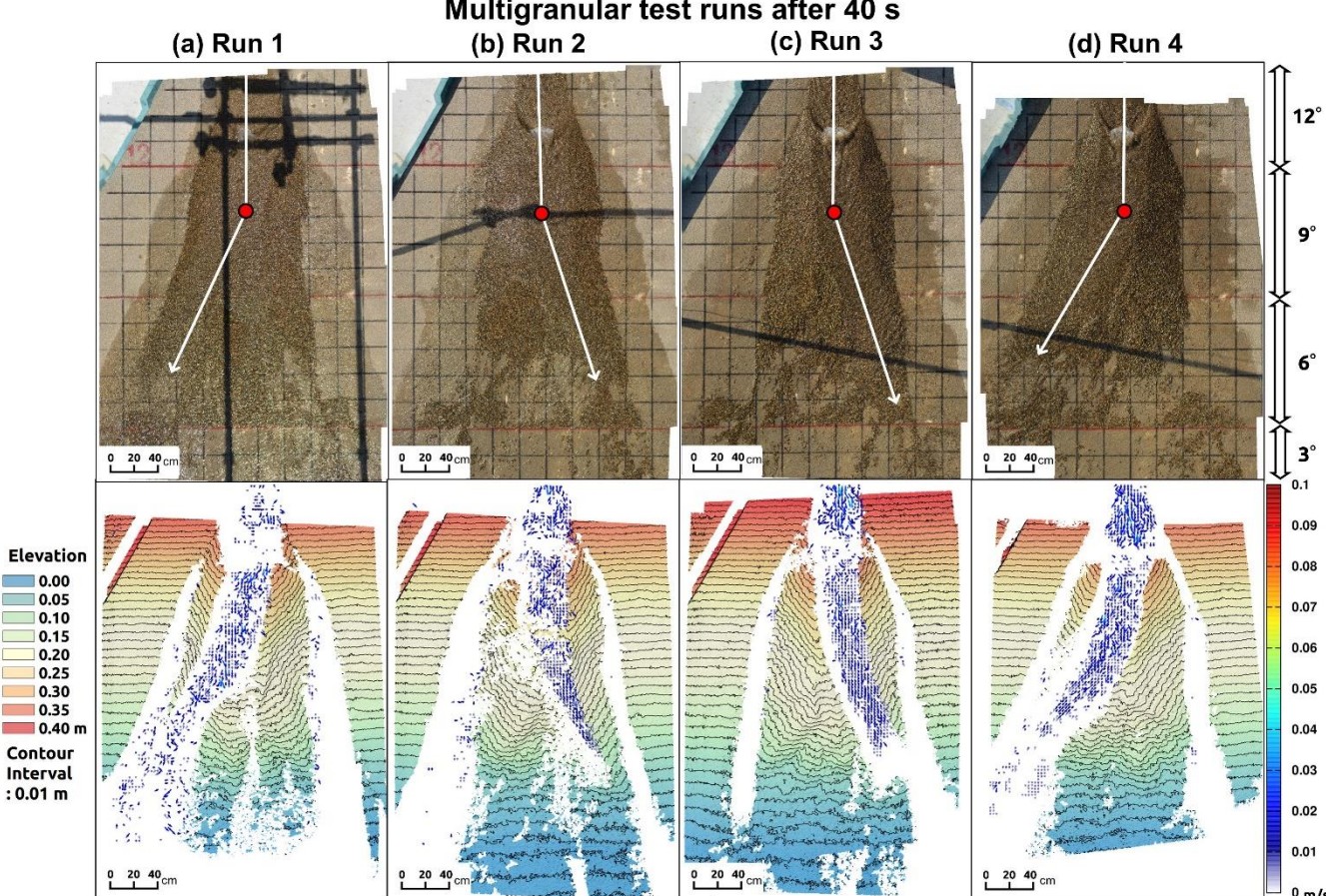

**Figure 13**: Fan morphology 40 s after the start of runout of the multigranular flows. The upper and lower panels show orthophotos and digital elevation models (DEMs) with flow vectors, respectively. Respective sets of the upper orthophoto and lower DEM represent corresponding results of each experimental test run. The white arrows on the orthophotos indicate the assumed principal direction of flow descent. The red points in the orthophotos indicate the assumed occurrence point of the avulsion. The elevation of the DEMs is depicted 840 assuming that the area with a 3° slope (i.e., the area furthest downstream from where the slope angle changed from a 6° to 3° slope) has elevation of zero.

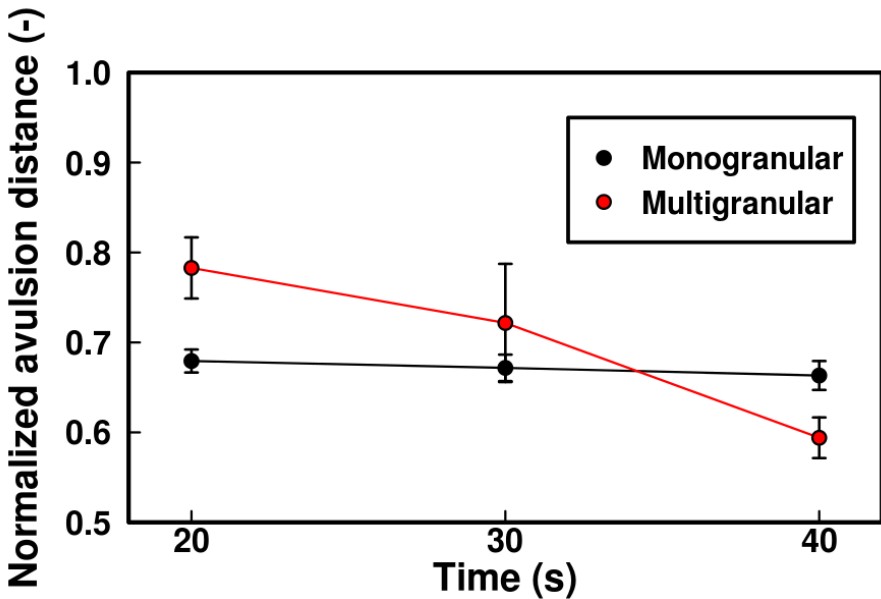

**Figure 14:** Change in normalized avulsion distance with time. The error bar indicates the standard deviation between the four experimental runs.

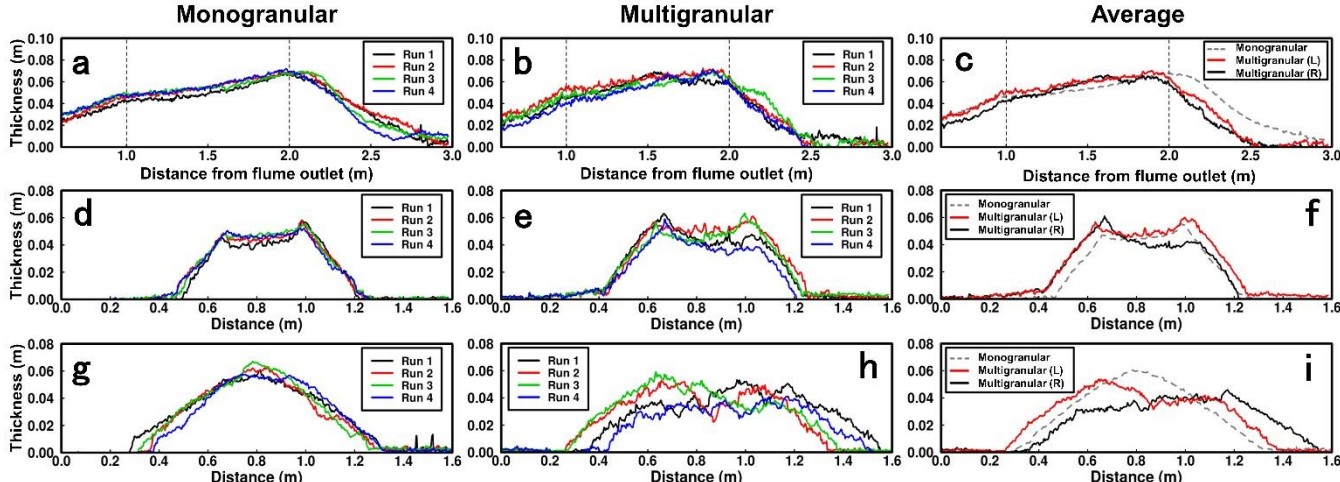

Figure 15: Profiles of the final fans. The left, center, and right panels indicate monogranular flows, multigranular flows, and their averages, respectively. (a)–(c) Longitudinal profiles at the center of the fan. Vertical broken lines indicate the boundaries of bed slope (i.e., the change points from 12° to 9° and from 9° to 6°). (d)–(f) Cross section (transverse profile) located 1 m downstream from the flume outlet. (g)–(i) Cross section (transverse profile) located 2.2 m downstream from the flume outlet. In panels d–i, the x-axis indicates the distance from the left-bank side of the cross sections (the fan was centered at approximately 0.8 m). In panels c, f, and i, the broken gray line indicates the average value of all monogranular flows. The red and black lines indicate the average values of the flows that produced fans that were elongated on the left-bank side (i.e., test runs 2 and 3) and on the right-bank side (test runs 1 and 4), respectively.

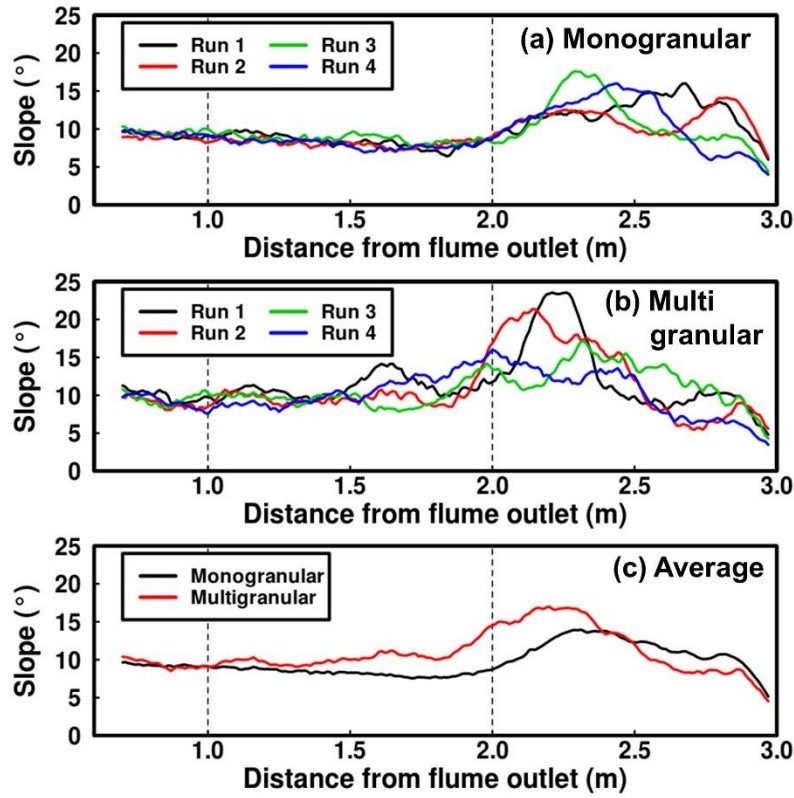

**Figure 16:** Slopes along the center of the final fans. The slope values were averaged over 0.2 m intervals. The upper, middle, and lower panels indicate monogranular flows, multigranular flows, and their averages, respectively. Vertical broken lines indicate the boundaries of bed slope (i.e., the change points from 12° to 9° and from 9° to 6°).

860

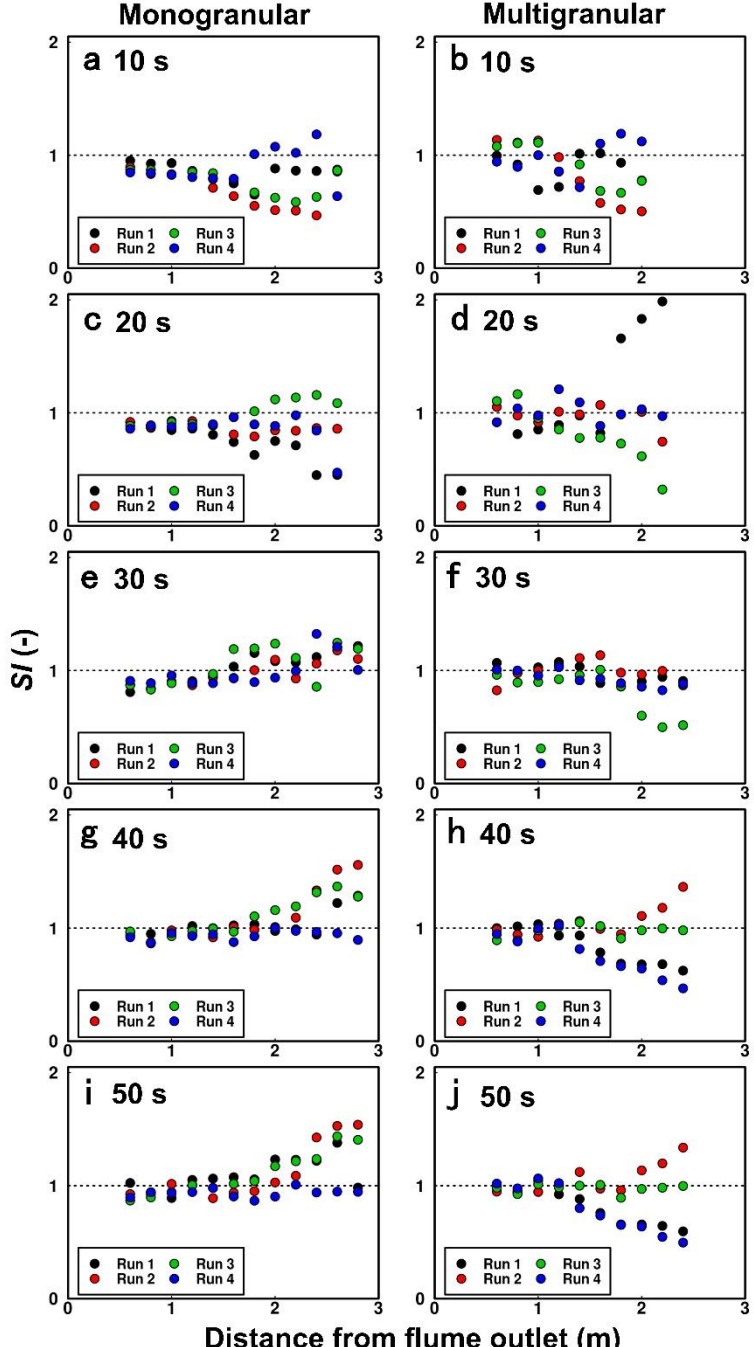

**Figure 17:** Changes in symmetric index (*SI*) values in response to fan-morphology formation. The left and right panels show results for monogranular flows and multigranular flows, respectively: (a) and (b) after 10 s from the start of flow runout, (c) and (d) after 20 s from the start of flow runout, (e) and (f) after 30 s from the start of flow runout, (g) and (h) after 40 s from the start of flow runout, and (i) and (j) after 50 s from the start of flow runout.

865

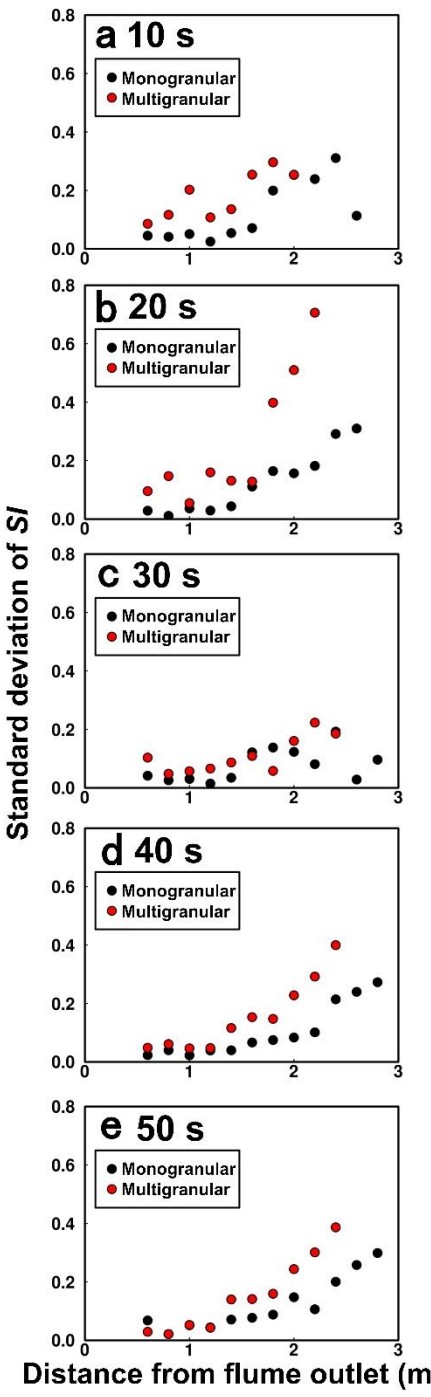

**Figure 18:** Standard deviation of symmetric index (*SI*) values: (a) after 10 s from the start of flow runout, (b) after 20 s from the start of flow runout, (c) after 30 s from the start of flow runout, (d) after 40 s from the start of flow runout, and (e) after 50 s from the start of flow runout.

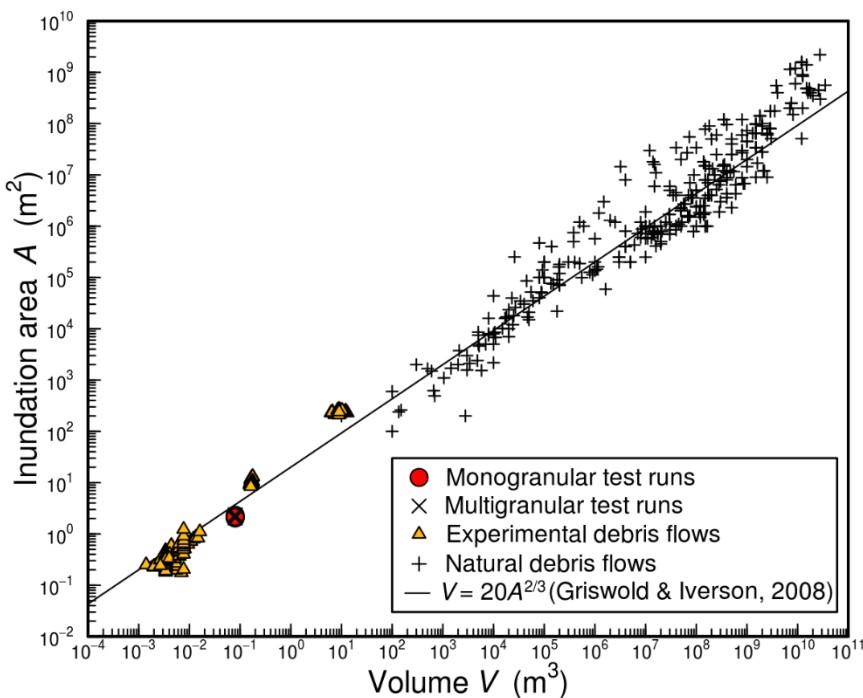

**Figure 19:** Debris-flow volume versus inundation area. Data concerning experiments (n = 454) are from Liu (1996), Major (1997), D'Agostino et al. (2010), De Haas et al. (2015b), Hürlimann et al. (2015), and Baselt et al. (2022). Data concerning natural debris flows (n = 323) are from Abele (1974), Li (1983), Crandell et al. (1984), Siebert (1984), Francis et al. (1985), Siebert et al. (1987), Hazlett et al. (1991), Hayashi and Self (1992), Siebe et al. (1992), Stoopes and Sheridan (1992), Iverson et al. (1998, 2015), Capra et al. (2002), Berti and Simoni (2007), Griswold and Iverson (2008), D'Agostino et al. (2010), Dufresne et al. (2019), Fan et al. (2019), and Friele et al. (2020). Note that the monogranular and multigranular test runs of this study are overlain in the log-log plane, and that the flow volume was approximated as 0.08 m$^3$ on the basis of the supplied sediment volume. The solid black line is the best-fit regression carve ($V = 20A^{2/3}$) derived by Griswold and Iverson (2008).