# Peer review of "Effect of debris-flow sediment grain-size distribution on fan morphology"

_Earth Surface Dynamics, 2021_

## Author Comment (AC1)

We are grateful for the constructive and detailed comments of both reviewers. In accordance with their helpful and insightful recommendations, we will substantially revise the original manuscript to clarify the results and significance of this research.

Below, we provide our preliminary responses to each of the reviewers' comments. The comments of the reviewers are shown in italics and different colors (i.e., those of Reviewer 1 and 2 are brown and blue, respectively), while our responses are provided in black text.

**Reviewer 1 COMMENTS AND AUTHOR RESPONSE**

**R1 General comment 1:**
*The authors investigate how the grain-size distribution of debris flows affects the fan-forming processes. For this, flume tests have been conducted to compare the debris-flow fan morphology under varying sediment source grain-size distributions. The obtained results associated with the debris-flow runout and space-time variations in the fan morphology provide important insights into how the grain-size distribution affects the fan formative processes. The topic is interesting, investigation is novel, and is within the scope of esurf. However, the ms requires a thorough revision in concept, mechanics and dynamics. Detailed suggestions and comments are provided below hopping that they will help to improve the quality, consistency and clarity of the revised ms.*

**Reply to R1 General comment 1:** We sincerely appreciate your thorough and helpful review. In accordance with your comments, we will carefully revise the manuscript to clarify our findings and assertions in terms of the mechanics and dynamics.

**R1 General comment 2:**
*Abstract can be improved. E.g., "Grain-size distribution was closely related to spatial diversity in fan morphology and stratigraphy." Should be other way round, "Spatial diversity in fan morphology and stratigraphy were found to be closely related to grain-size distribution."*

**Reply to R1 General comment 2:** We will revise the abstract to improve the quality and clarity in accordance with your helpful guidance.

**R1 General comment 3:** *Introduction is not that much concerned about the main topic on how the grain-size distribution influences the debris flow fan morphology. So, it needs to be substantially expanded focusing on how the grainsize of the source material affects the deposition and fan formation process including the important and often dominant dynamical processes - the material separation, erosion and run-out dynamics.*

**Reply to R1 General comment 3:** To highlight our main topic on how the grain-size distribution

influences the fan morphology, we will revise the introduction in light of important debris-flow dynamics.

**R1 General comment 4:** *Often the writing is not explicit, not smooth and difficult to follow. It seems if the ms was made for very short paper, less for a professional journal, requiring clearer and smoother presentation. Particular attention should be given on these issues.*

**Reply to R1 General comment 4:** We apologize for the confusion. To meet the required clearer and smoother presentation, we will thoroughly revise the manuscript in terms of clarity, overall language, and terminology.

**R1 Specific comment 1:** *L36-37: "Changes in the physical parameters (e.g., flow rate, duration, and sediment concentration)": The writing must be significantly improved, conceptually: flow rate, duration, and sediment concentration are not the physical parameters, rather they are the dynamical quantities. Physical parameters include densities, viscosities, frictions, slope geometry, curvature, etc.*

**Reply to R1 Specific comment 1:** We agree with this comment and will revise the sentences regarding the physical parameters.

**R1 Specific comment 2:** *L38-39: "and sediment entrainment rate (Egashira et al., 2001; De Haas and Van Woerkom, 2016)." better change to "and sediment entrainment rate (Egashira et al., 2001; De Haas and Van Woerkom, 2016; Pudasaini and Fischer, 2020: https://doi.org/10.1016/j.ijmultiphaseflow.2020.103416) and separation between the particles and fluid in the mixture (https://doi.org/10.1016/j.ijmultiphaseflow.2020.103292)."*

**Reply to R1 Specific comment 2:** We will revise the sentence in accordance with your comment. Moreover, we will add new results regarding the phase separation between the particles (solid) and fluid (please see **Reply to R1 Specific comment 14**).

**R1 Specific comment 3:** *L39-40: "and depending on the topographic complexity, could produce varying functional changes in subsequent debris flows.": not clear how topography produces functional changes and which? "functional and structural changes" what are the functional and structural changes? General readers may not be able to follow the text.*

**Reply to R1 Specific comment 3:** By avoiding the use of vague terms, such as functional and structural changes, we will revise the introduction section thoroughly.

**R1 Specific comment 4:** *L50: "A straight flume (8 m long and 0.1 m wide, with a uniform 15° bed slope": This is not true, and the text around must be improved appropriately, consistent with the*

*corresponding figure.*

**Reply to R1 Specific comment 4:** We will revise the related sentences and figure to explain them correctly.

**R1 Specific comment 5:** *L50-67: "erodible bed conditions": Here comes the great thing! I see two major aspects in this ms. First, as the authors say, the effect of particle size in the erodible bed and how it will affect the deposition fan. I understand this differently than the author, it is not the particle size in initial debris mass (initially water is released) that will influence the deposition fan, but it is how the particle size of the erodible bed that affects the deposition fan when that bed substrate is eroded and entrained by the water flood released from upstream, consequently forming a debris mixture that ultimately flows down and deposits in the gentle open flat slope forming debris fan. The text does not mention this.   Second, probably even more important, is the fact that the flood erodes and entrains the granular bed converting it in to the debris flow. So, the physical process of erosion, entrainment and the associated mobility must be described. This could however be done with respect to the mechanical erosion rate models and the mechanical model for the mass flow mobility with erosion (Pudasaini and Krautblatter, 2021: https://www.nature.com/articles/s41467-021-26959-5). I would focus on these governing aspects of experiments.*
*Writing style needs to be made more appropriate with better physical understanding. "The supplied water generated a granular flow that imitated a single debris-flow surge and then entrained the erodible bed to the deposition area" This is difficult to follow, probably not representing reality. Does the water flow first generate the granular front by entraining the granular bed? I guess, as the water front impacts the granular bed it will erode and entrain the grain, mixing will take place resulting in the subsequent debris flow. Not that the way the authors explained. Moreover, you mixed up erosion and entrainment, which are clearly two different mechanical processes as proven by the reference mentioned above. So, the process of erosion and entrainment should be carefully investigated/discussed.*

**Reply to R1 Specific comment 4:** We sincerely appreciate this important comment. We have read the new paper (Pudasaini and Krautblatter, 2021) and interpreted your concern as explained below:
1. Pudasaini and Krautblatter (2021) derived a new theoretical framework called the three-E (erosion-entrainment-energy) mechanical concepts.
2. They defined two velocities: Erosion velocity is the velocity that bed substrate is fed by the debris flow (i.e., the velocity vector intersects the flow line) and Entrainment velocity is the velocity that debris flow transports (entrains) the eroded bed materials. These velocities are basically different.
3. Erosion velocity contributes to the momentum transfer in the debris flow, whereas Entrainment velocity contributes to changes in the inertia of the debris flow. Thus, depending on the relationship between Erosion velocity and Entrainment velocity, the mobility of the debris flow can change.

4. Because in our experiments there is the possibility that these velocities were different between mono-granular material runs and multi-granular material runs, you have wondered that the debris-flow mobility was also different between mono-granular and multi-granular flows.

First, we agree that this new framework can be the key to theoretically unraveling the sediment erosion and deposition processes of debris flows. There is the possibility that our experiments are not adequate for direct assessing debris-flow dynamics based on the proposed 3E concepts. Both beds in the flume and the deposition area (connected plane) gradually change their morphology (slope), meaning that we could not trace the boundary between the flow bottom and bed surface easily, and preventing the measurement of erosion rate. Because of this limitation, most previous studies relied on the measurement of changes in the sediment concentration of debris flows rather than the erosion rate (e.g., Lanzoni et al., 2017). Indeed, even Pudasaini and Krautblatter (2021) have recognized the difficulty of the demonstration of their 3E model using existing flume test and field data. Anyway, while considering the limitations of our experimental setup, we will improve the introduction and discussion sections of the revised manuscript in light of this important framework.

The use of "entrainment" was intended to avoid confusion of the mixing between the erosion of fluvial sediment and erosion of bedrock (like incision). The erosion velocity is carefully defined by Pudasaini and Krautblatter (2021), but we were afraid that most readers recall "bedrock erosion" rather than debris-flow erosion processes. In the revised manuscript, we will adequate terms properly. For example, we will revise the sentences regarding the generation of debris flow as follows.

**Revision:** *By suddenly supplying water from the upper end of the flume, we generated a granular-water mixture flow that imitated debris flow similar to Lanzoni et al. (2017). We could not control erodible bed saturation completely because bed materials included voids. Saturated bed conditions were approximated by carefully supplying clear water across the entire erodible bed using watering cans just before we started the water supply from the upper end of the flume. Following this operation, a steady flow of clear water (the fluid density is ~1,000 kg/m³) was supplied at a rate of 0.003 m³/s for 60 s from the upper end of the flume. The supplied water plunged over the erodible bed and flowed downstream, generating a runoff front over the loose sediment particles. The runoff front scoured the sediment particles of the erodible bed and entrained the eroded particles, dispersing the entrained particles throughout the flow depth, and eventually transforming to a granular-water mixture flow that imitated a single debris-flow surge in the flume (Lanzoni et al., 2017).*

**R1 Specific comment 5:** *"the fan morphology gradually formed in accordance with the runout and inundation of the released granular flow": This is rather the fan of the granular-water mixture debris flow. For a granular fan, you must only have the dry material without water, for which the fan will be substantially different than what it is now. Two types of granular flow, namely mono-granular and multi-granular, were used to determine the impact of grain-size distribution within a debris flow on the fan-forming processes.": This is not right. You have two types of granular materials in the erodible bed, resulting in two types of debris flows, one consisting of water and mono-granular material, another composed of water and multi-granular [this term needs to be defined carefully, as mono-granular and multi-granular are not the usual terms, usual terms are the mono-dispersed and poly-dispersed, with respect to grain size, etc.].*

**Reply to R1 Specific comment 5:** We sincerely appreciate this comment and will revise the related sentences to avoid this confusion as follows.

**Revision:** *The generated granular-water mixture flow reached the deposition area and caused its runout and inundation. The slope of the deposition area decreased from 12° to 3° at a rate of 3° per meter (Fig. 1a, b), and so the fan morphology gradually formed in accordance with the runout and inundation of the granular-water mixture flow.*

Additionally, in the earlier part of the revised manuscript, we will define the mono-granular and multi-granular flows, respectively.

**Revision:** *In this study, we intended to generate and compare the granular-water mixture flows that are of similar flow states but with different grain-size distributions, to focus on the effects of debris-flow grain-size distribution on the debris-flow fan during the debris-flow runout and inundation processes. To accomplish this, two types of sediment particles were used to generate two types of the granular-water mixture flows: mono-granular particles that sediment particles are quasi-mono-dispersed sediment particles consisting of 2.02–3.24 mm (the average grain size, $D_{50}$, is 2.6 mm) and multi-granular particles that are poly-dispersed sediment particles consisting of 0.6–7.5 mm (Table 1). The density and the internal friction angle of both particles are 2,640 kg/m$^3$ and 34.0°, respectively. Hereafter, the granular-water mixture flows that are generated by the mono-granular particles and the multi-granular particles are referred to as the mono-granular flow and the multi-granular flow, respectively. We carried out four experimental test runs for the mono-granular flow and multi-granular flows, respectively.*

**R1 Specific comment 6:** *Not enough information on the material and channel are provided, e.g., the basal and internal friction angles of the granular material, viscosity of water, their densities, and so on. This information is crucial in understanding erosion-entrainment and mobility, the mixing and separation between particles and fluids, and the transport/deposition of debris mixture.*

**Reply to R1 Specific comment 6:** We agree with this comment and will add this information in the revised manuscript (please see **Reply to R1 Specific comment 4** and **Reply to R1 Specific comment 5**).

**R1 Specific comment 7:** *Another principle concern is the representative grainsize, the two granular materials, mono-dispersed and poly-dispersed are represented by the same average grain size (D50). This does not help to physically clearly study the erosion-entrainment, transport and deposition fans, except that you can say – we observe this and that for the mono-dispersed and poly-dispersed erodible bed. But, we don't know how small and big particles in the mixture influence the erosion, mixing or separation, dynamics and deposition processes. Moreover, different grains might need to be represented by different rheological equations. These are crucial aspects the authors should discuss. Otherwise the results can not be understood mechanically and dynamically clearly, and these data cannot that easily be used in model validation and parameter calibration.*

**Reply to R1 Specific comment 7:** We agree with this comment. By referring to new results how differences in the grain-size distribution lead to variation in the phase separation and symmetry of the fan morphology, we will improve the discussion section in light of known dynamics with respect to sediment erosion/deposition, flow runout, and the phase separation (**Reply to R1 Specific comment 14** and **Reply to R2 Comment 25**).

**R1 Specific comment 8:** *L69-70: "sediment was released to the deposition area". A bit strange writing. First, it is not sediment, it is the debris material. Second, it is not released to the deposition area, it is the transported material in the deposition area. So, the dynamical perspectives are weak.*

**Reply to R1 Specific comment 8:** We will revise the sentence in accordance with your comment (please see **Reply to R1 Specific comment 5**).

**R1 Specific comment 9:** *L70-73: "The flow depth of a generated granular flow cannot be measured in the flume because the thickness of the erodible bed decreases sequentially in response to the sediment entrainment. Therefore, the displacement of the flow surface at three positions in the flume (upper, middle, and lower, Fig. 1a) was measured to account for this shortcoming, using ultrasonic displacement meters":*
*I agree with the first sentence, it is a really complex process, however, there are some literature in*

*this direction with some success (Lanzoni et al., 2017: https://doi.org/10.1002/2016JF004046). The authors should put some efforts to review relevant literature. The bed erosion process is an under-investigated process, and I respect any attempt in this direction. The second sentence is not the solution to the first, because, the measured flow depth cannot be split into the material from the flow and from the bed. Thus, it cannot be straightforward connected to the erosion depth. Furthermore, the involved energy associated with erosion is the dominant factor to decisively defining the dynamics, runout and the associated impact forces of the erosive mass flows. This needs to be discussed with respect to the references mentioned above.*

**Reply to R1 Specific comment 9:** We agree with your statement about the erosion/entrainment processes and the necessity of the descriptions based on relevant literature. As Pudasaini and Krautblatter (2021) proposed, erosion and entrainment potentially influence the debris-flow dynamics and runout through the different involved energy (different contributions of momentum conservation). Because we could not directly measure erosion and entrainment velocity from experimental data, the explicit application of the 3E concepts proposed by Pudasaini and Krautblatter (2021) is difficult. Thus, while referring to the literature effectively, we will consider such aspects in the discussion section of the revised manuscript. Additionally, we will revise the sentences regarding the flow depth measurements to improve clarity as follows.

**Revision:** *We measured the flow depths of the generated debris flows in the flume and investigated the runout and fan-forming processes at the deposition area. By comparing the changes in the flow depth between the mono-granular flows and the multi-granular flows, we assessed whether it has been deemed that the multi-granular flows exhibited the analogous hydrograph and velocity, compared with those of the mono-granular flows. Note that, we could not directly measure the flow depths of the generated debris flows because the thickness of the erodible bed decreases sequentially in response to the sediment erosion and entrainment. The continuous sediment erosion and entrainment hampered to estimate of the boundary between the flowing bottom and the bed surface (e.g., Lanzoni et al., 2017), which hampers a quantitative measurement of the flow depth of the debris flow. Instead of the flow depth, we measured the changes in the displacement of the flow surface by three ultrasonic displacement meters (Omron, E4PA) at a sampling rate of 50 Hz. Each ultrasonic displacement meter was installed at the three positions in the flume, respectively (Fig. 1a). Hereafter, the positions are referred to as upper, middle, and lower measurement positions, respectively, from the upstream to downstream of the flume. Because the initial depth of the erodible bed was adjusted as 0.2 m, the flow depths of the debris flow were interpreted based on the measured displacement subtracted from this initial bed depth. Based on differences in the timing that the debris-flow front reached the lower position, we compared the flow rate in the flume among test runs.*

**R1 Specific comment 10:** *I stop suggesting and commenting on the mechanical and dynamical*

*aspects of the ms, and hope that the authors will improve the text while revising it.*

**Reply to R1 Specific comment 10:** We sincerely appreciate your constructive suggestions again. In accordance with your comments, we will carefully revise the manuscript to strengthen aspects in relation to debris-flow mechanics.

**R1 Specific comment 11:** *Fig. 2: Figures are difficult to follow. It should be self-explanatory. For example, Run 1-4, are they repeated exps.?*

**Reply to R1 Specific comment 11:** "Run" indicates each experimental test run. We will revise the sentences in the method section to convey the results smoothly (please see **Reply to R1 Specific comment 5**).

**R1 Specific comment 12:** *L98-99: "The thickness of the erodible bed decreased monotonically with time, probably because the entrainment rate was the same in all the test runs, irrespective of the grain-size distribution of the granular flows": This cannot be true, could only be a speculation. Because, as proved in the above-mentioned references, erosion rate is a complex phenomenon, and changes with the dynamic load applied by the flow and resisted by the bed. This needs to be discussed.*

**Reply to R1 Specific comment 12:** As we responded above, we will carefully describe and discuss the aspects regarding erosion/entrainment rate (please see **Reply to R1 Specific comment 4** and **Reply to R1 Specific comment 9**).

**R1 Specific comment 13:** *L100-102: "Overall, the results from the flume experiment showed that the difference in the grain-size distribution did not lead to substantial changes in the hydrograph and arrival time of the granular flows.": I can't fully agree with this. E.g., if you take the mean of four runs in C and F and plot them in one figure, you will see discernible difference.*

**Reply to R1 Specific comment 13:** We agree that the previous statement was a strong tone and will carefully explain the differences and similarities between mono-granular and multi-granular flows.

**R1 Specific comment 14:** *L113: Grain size separation is one aspect, but separation between particles and fluid (as seen in the experimental results) is another, even more complex mechanical phenomenon in debris flow. However, the authors did not discuss anything on it.*

**Reply to R1 Specific comment 14:** We sincerely appreciate this important suggestion and agree that the separation between particles (sediment) and fluid is certainly one of the important aspects. As the preliminary analysis, we measured time-series changes in runout distances of the fronts of

generated debris flows with a temporal resolution of 0.1 s, using the captured video and the grid lines at the deposition area (**Figure R1**). We found the different trend with respect to the separation between the solid (sediment particles) and fluid phases: the phase separation of the multi-granular flows occurred in the earlier stage of runout processes compared with those of the mono-granular flows. We will assess and discuss these results in the revised manuscript.

[Figure]

**Figure R1:** Time series changes in the runout distances of the flow fronts. (a) mono-granular flows. (b) multi-granular flows.

**R1 Specific comment 15:** *L123: "avulsed obviously": it is better also to put orthophoto to clearly see avulsion. The quality of Fig. 5-8 should be improved, with filters, or whatever means such that we can clearly see avulsion. The problem I have seen is that avulsion cannot be predicted, or was not possible with the present setup. We should understand why this is happening. This needs to be discussed, because, one of the main aims of experiments should be to generate reproducible results.*

**Reply to R1 Specific comment 15:** We apologize for the confusion. While effectively using metrics regarding the fan shape, we will improve the explanation how the avulsion occurred in response to the fan development (please see **Reply to R2 Comment 25**).

**R1 Specific comment 16:** *L142-144: "Some equations that describe debris flows assume that multi-granular debris flows can be approximated to mono-granular debris flows with the same average grain-size (e.g., Egashira et al., 1997; Takahashi, 2007).": This is not the state-of-the art. The multi-mechanical, multi-phase mass flow model by Pudasaini and Mergili (2019: https://doi.org/10.1029/2019JF005204) has proven the necessity of simulating debris flows as mixture of different materials, that has been used in accurately simulating complex multi-phase natural events (Mergili et al., 2020: https://doi.org/10.5194/nhess-20-505-2020; Shugar et al., 2021: DOI: 10.1126/science.abh4455). The ms should be up dated with relevant, recent literature.*

**Reply to R1 Specific comment 16:** We apologize for the confusion. In the revised manuscript, while referring to the suggested multi-phase model, we will revise the related sentences to improve clarity.

**R1 Specific comment 17:** *L144-145: "However, the mono-granular and multi-granular flows with the same average grain-size produced fans with different morphologies": This is probably the most important aspect of this ms, and I like it. However, it has not yet been clearly discussed for why this is so. The authors should put some energy to explore why it is happening, that will lift the importance of this paper to a higher level.*

**Reply to R1 Specific comment 17:** We agree with these comments and will emphasize the explanation of why the differences in the fan morphology arose in the revised discussion section, by referring to new results (please see **Reply to R1 Specific comment 14** and **Reply to R2 Comment 25**).

**R1 Specific comment 18:** *L146-147: "which indicates that existing models that assume a mono-granular approximation may provide ambiguous simulations of the debris-flow deposition and inundation ranges.": This proves the need of multi-phase mass flow models (mentioned above) in properly simulating debris flows. This should be discussed.*

**Reply to R1 Specific comment 18:** We sincerely appreciate this important comment and agree with the importance of how relatively small particles behave like fluid or solid and change the stress structure of debris flows. As you mentioned, natural debris flows obviously contain such small particles, which may be responsible for further variations in the fan morphology. We will revise the discussion section considering this aspect.

**R1 Specific comment 19:** *Discussion and Conclusion, References:*
*Needs re-working, including the above suggestions. E.g., multi-phase flow simulations, erosion-entrainment and mobility, separating particles and fluid, and so on. Important point why the flow with the poly-dispersed erodible bed has shorter travel distance and run-out reveals that more energy has been consumed for this than the bed with mono-dispersed particles. This exclusively depends on the erosion velocity controlling the mobility of the mass flow, this fact has been proven by the mechanical model for the mobility of erosive mass flows by Pudasaini and Krautblatter (2021). The discussion and conclusion should give proper space for these important mechanical and dynamical aspects also observed in this ms.*

**Reply to R1 Specific comment 19:** We sincerely appreciate this important comment and will discuss this aspect referring to new results (please see **Reply to R1 Specific comment 14**).

**R1 Technical comment 1:** *L22: "sinks", the meaning was not that clear, better would be "deposits"?*

**Reply to R1 Technical comment 1:** By avoiding the use of vague terms, we will improve the clarity of the introduction section thoroughly.

**R1 Technical comment 2:** *L29: Please check English.*

**Reply to R1 Technical comment 2:** We will revise it adequately.

**R1 Technical comment 3:** *L85: "SfM-MVS": is its meaning clear?*

**Reply to R1 Technical comment 3:** By referring adequate literature, we will revise this sentence.

**R1 Technical comment 4:** *L97-98: "while, apart from run 1, those of the multi-granular flows were around ~0.03 m": Not true. Please check all the technical details carefully.*

**Reply to R1 Technical comment 4:** We will carefully revise it to improve clarity.

**Reviewer 2 COMMENTS AND AUTHOR RESPONSE**

**R2 Overall comment:** *In this manuscript the authors used a flume to analyse how the grain size distribution of a debris flow may impact the morphology of the resulting fan. The authors ran 2 sets of experiment runs in their flume, all parameters inside the experiments were kept constant except the grainsize distribution. One set of experiment runs used monogranular sediment while the other used multigranular. Both sets had the same average grain size so the authors could identify the impact of variations in grain size on the debris flow fan. They quantified these changes by measuring the surge height in the flume, the speed of the surge within the deposition area, the runout distance and a DEM of the final debris flow. The authors discovered that multigranular debris flows were more likely to produce alluviations in the debris flow resulting in asymmetrical fans. They postulate that these alluviations are the result of grain size segregation occurring within the flow where coarse sediment is forced to the front of the flow where it can produce an obstacle for any following surges.*

*I think this study is an interesting addition to the literature on debris flows. The experiments seem well thought-out and the results aim to fill a clear knowledge gap. However, there are several areas where I feel the manuscript needs to be improved before it is ready for publication. The manuscript is very short and as a result I feel that significant detail is missing, particularly from the description of the experiment design, results and discussion. I also found the figures poorly cited and discussed throughout. I have provided more specific comments below for the authors to read through. I hope the authors find my comments useful and I look forward to seeing the revised manuscript.*

**Reply to R2 Overall comment:** We sincerely appreciate your insightful review and constructive suggestions. In accordance with your comments, we will thoroughly revise the manuscript to improve clarity and the level of detail in all sections. My response to each of your specific comments is provided below.

**R2 Comment 1:** *Introduction: The introduction is too short and vague to be of use to the reader. Despite the research statement at the end of the section it is not completely clear how the authors see the study contributing to the literature. It is also not obvious from these paragraphs why the authors have chosen to focus on grain size distribution for this study rather than many of the other controlling factors highlighted here. Finally, this section would be greatly improved if there was better separation between discussing debris flow physics from debris flow fans. Currently it is very confusing whether the authors are referring to how a certain parameter might affect a debris flow or how it may affect the autogenic fan forming processes or the links between the two.*

**Reply to R2 Comment 1:** We agree with this comment and will revise the introduction section by separating descriptions with respect to debris-flow physics and debris-flow fans.

**R2 Comment 2:** *Lines 25 – 26: What specific climate and sediment dynamics information can be identified from debris flow fans? And how is it derived? How is the form (which is what is investigated here) important?*

**Reply to R2 Comment 2:** By avoiding the use of vague terms, we will improve the clarity of the introduction section thoroughly.

**R2 Comment 3:** *Lines 26 – 29: If debris flow fans are primarily formed by autogenic processes how can information on any external forcing be derived from them?*

**Reply to R2 Comment 3:** By avoiding the use of "autogenic", we will explain how fan-forming processes are driven in the revised introduction section.

**R2 Comment 4:** *Lines 31-33: Shifting the focus of the paragraph from a geological perspective of fans to one about hazard is confusing to the reader particularly as neither focus is well covered.*

**Reply to R2 Comment 4:** We agree with this comment. We will explain those aspects with regard to debris-flow hazards and records of sediment regimes as general importance to study debris-flow fans in the first paragraph of the introduction, rather than the explanation in detail.

**R2 Comment 5:** *Lines 34-36: It would be useful to the reader if the authors would elaborate on how*

*these physical factors affect morphology and stratigraphy of the fans.*

**Reply to R2 Comment 5:** We agree and will carefully explain on linkages between debris-flow physical factors and influences on morphology and stratigraphy of the fans.

**R2 Comment 6:** *Lines 36 – 38: From this section the reader cannot tell how these changes will affect the debris flows. The authors do not define the property that is changing carefully nor do they describe the impact of these changes on debris flow behaviour. Without this information the readers can not make the link between debris flows and the resulting fan.*

**Reply to R2 Comment 6:** To taking account this comment and **R2 Comment 4**, while avoiding the use of vague terms, such as functional and structural changes, we will revise the introduction section thoroughly.

**R2 Comment 7:** *Lines 44 – 48: This is a good succinct research statement however it is completely disconnected from the preceding 2 paragraphs. It does not mention why the authors have chosen to focus on grain size distributions nor how they expect them to change the debris flow fan.*

**Reply to R2 Comment 7:** To convey our intention and why we designed the experiments that control only differences in grain-size distribution, we will thoroughly revise the introduction section.

**R2 Comment 8:** *Methods: The experiment design is reasonably well explained, however I struggled to understand what exactly was being measured. The authors have gone to great lengths to capture the vast amounts of data generated by the experiments; however, they do not discuss why they collected these particular datasets or what they plan to do with them. A better motivating statement within the introduction will help to improve this section.*

**Reply to R2 Comment 8:** We agree and will provide adequate context at the end of the introduction section to smoothly connect to the methods section. Also, we will carefully describe why we measured both flow depth in the channel and changes in the fan morphology to convey our intention that examines whether debris-flow surges with different grain-size distributions can provide different fan morphology even their flow states are similar.

**Revision in the end of the introduction section:** *The primary objective of this study was to assess how the grain-size distribution influences the fan morphology, especially during the debris-flow runout and inundation. To archive this, we carried out reduced-scale flume tests to compare the fan morphology that was formed by the single debris-flow surge with the different grain-size distribution but with similar flow characteristics. Using the photogrammetry and video-image analysis, we*

investigate how the difference in the grain-size distribution serves to variations in the runout characteristics and fan morphology. The intention underlying this comparison was to interpret the differences in the fan morphology in terms of known debris-flow mechanics. The final goal was to elucidate whether the difference in the grain-size distribution of the debris flow can change the fan morphology by solely influencing during the runout process without the difference of the debris-flow magnitude in the channel.

**Revision in the method section:** *We measured the flow depths of the generated debris flows in the flume and investigated the runout and fan-forming processes at the deposition area. By comparing the changes in the flow depth between the mono-granular flows and the multi-granular flows, we assessed whether it has been deemed that the multi-granular flows exhibited the analogous hydrograph and velocity, compared with those of the mono-granular flows. Note that, we could not directly measure the flow depths of the generated debris flows because the thickness of the erodible bed decreases sequentially in response to the sediment erosion and entrainment. The continuous sediment erosion and entrainment hampered to estimate of the boundary between the flowing bottom and the bed surface (e.g., Lanzoni et al., 2017), which hampers a quantitative measurement of the flow depth of the debris flow. Instead of the flow depth, we measured the changes in the displacement of the flow surface by three ultrasonic displacement meters (Omron, E4PA) at a sampling rate of 50 Hz. Each ultrasonic displacement meter was installed at the three positions in the flume, respectively (Figure 1a). Hereafter, the positions are referred to as upper, middle, and lower measurement positions, respectively, from the upstream to downstream of the flume. Because the initial depth of the erodible bed was adjusted as 0.2 m, the flow depths of the debris flow were interpreted based on the measured displacement subtracted from this initial bed depth. Based on differences in the timing that the debris-flow front reached the lower position, we compared the flow rate in the flume among test runs.*

*The measurements of the runout and fan-forming processes relied on image analyses. To observe the runout and fan-forming processes, four digital cameras were installed above the deposition area (Fig. 1a) similar to Tsunetaka et al. (2019). One of these cameras (PENTAX, K-3ii) recorded a video of the fan-forming processes at a frame rate of 60 fps. The images extracted from the recorded video were used to analyze the debris-flow runout distance and velocity as well as the changes in the flow direction during the inundation at the deposition area. The other three of these cameras (Nikon, D5100) were automatically synchronized using the external shutter (Canon, TC-80N3) and captured images at 1-s intervals. These sets of three synchronized images were processed to generate topographic data of the formed fan morphology using a photogrammetry method. Below, we explain the detail of each image-based analysis.*

**R2 Comment 9:** *Lines 50-60: The authors have not explained how the debris flow surge is generated. The paragraph could be separated to first describe how and where the surge is generated before*

*discussing the depositional area.*

**Reply to R2 Comment 9:** While referring to the adequate literature and considering given comments, we will carefully explain how we generated debris-flow surges (please see **Reply to R1 Specific comment 4**).

**R2 Comment 10:** *Lines 61-67: How fans are produced in this study? Is the fan a result of a single surge triggered by the outlet of water? Or the result of multiple surges? Is the erodible layer rebuilt between surges? What is going to be measured as a result of these experiments?*

**Reply to R2 Comment 10:** We focused on the debris-flow fan that is formed by a single debris-flow surge. Each experimental result (each run) indicates the data of the fan formed by a single surge rather than multiple surges. We will explain why we designed such setup and condition by descriptions regarding our intention (please see **Reply to R1 Specific comment 5**).

**R2 Comment 11:** *Lines 69-74: Why is flow displacement being measured? Why is the flow height of the surge important to the authors?*

**Reply to R2 Comment 11:** We intended to clarify differences and similarities between the mono-granular and multi-granular flows in the flume using measurements of flow heights. We will carefully explain our intention and how the measured depth helps our interpretations (please see **Reply to R1 Specific comment 9** and **Reply to R2 Comment 8**).

**R2 Comment 12:** *Lines 79-80: What is being measured and how?*

**Reply to R2 Comment 12:** To check the accuracy of the SfM photogrammetry, we compared the deposition depths of the experimental debris-flow fan that were directly measured after each experiment run using a ruler. We will explain it in detail in the revised manuscript as follows.

**Revision:** *To assess the accuracy of DEMs, deposit depths of the debris-flow fan were directly measured at the intersections of the grid lines using a ruler after each experimental run, and compared the measurements with the deposition depths extracted from the generated DEM. The measured elevations corresponded to the DEM-extracted elevations, thereby indicating that the DEMs approximated well to the fan morphology (Fig. S1).*

**R2 Comment 13:** *Line 86: It is not clear what is meant by "the SfM-MVS photogrammetry could not measure locations where granular flows descended". Does this mean that the photogrammetry cannot measure the flow when it is moving?*

**Reply to R2 Comment 13:** Yes, the moving flow provided complex surface undulations and different brightness in captured images, which resulted in unmeasurable zones of the photogrammetry results. We will clarify this point in the revised manuscript as follows.

**Revision:** *During the inundation of sediment at the deposition area, the SfM-MVS photogrammetry could not measure locations where granular flows descended (i.e., moving zone), which resulted in holes of DEMs due to lacking topographic data.*

**R2 Comment 14:** *Results: The results section also suffers from the same problems as the previous sections. What is being measured and compared between the different runs is not specifically stated and as a result it is hard to understand some of the findings of the manuscript. Many of the result figures are poorly explained and some are cited out of order or not cited at all.*

**Reply to R2 Comment 14:** We agree with this comment and will revise the results section by referring to figures adequately.

**R2 Comment 15:** *Line 95: What is the lower portion of the flume? How is this defined?*

**Reply to R2 Comment 15:** The lower position should be stated as the lower measurement position of the ultrasonic sensor. We will revise the related sentences to define it (please see **Reply to R1 Specific comment 9** and **Reply to R2 Comment 8**).

**R2 Comment 16:** *Line 96: Same with arrival point and upper position.*

**Reply to R2 Comment 16:** Similarly, we should define the upper position indicates as the upper measurement position of the ultrasonic sensor. We will revise the related sentences to define it (please see **Reply to R1 Specific comment 9** and **Reply to R2 Comment 8**).

**R2 Comment 17:** *Line 97: What is a run? This refers back to the earlier point that it is not clear whether the experiment is single or multiple surges.*

**Reply to R2 Comment 17:** A run indicates an independent experimental test run. We will define it in the revised manuscript (please see **Reply to R1 Specific comment 5** and **Reply to R1 Specific comment 11**).

**R2 Comment 18:** *Line 101: Unclear how the arrival time is measured.*

**Reply to R2 Comment 18:** We measured the time from the timing that the flow front started its runout at the deposition area to the timing that the flow front stopped as the arrival time. We will add new results with respect to the arrival time in the revised manuscript with an explanation of how we measured them (please see **Reply to R1 Specific comment 14**).

**R2 Comment 19:** *Lines 103-106: This should be in the discussion or introduction rather than in the results*

**Reply to R2 Comment 19:** We agree and will discuss this aspect regarding flow state and similarity low in the discussions section.

**R2 Comment 20:** *Lines 110 – 113: It is unclear which panel is figure 3 is being referred to. Panel 3c is also not cited at all in this section.*

**Reply to R2 Comment 20:** We apologize for the confusion and will revise the related sentences to clarify the referred figure.

**R2 Comment 21:** *Figures S2 and S3 seem important to the overall narrative of the manuscript and therefore the authors should consider including them as part of the main text.*

**Reply to R2 Comment 21:** We appreciate this suggestion and will include Figures S2 and S3 in the revised main text.

**R2 Comment 22:** *Lines 119-121: State how the locations of the lobes differ, are they closer to the flume exit? Does the slope differ between the 2 locations?*

**Reply to R2 Comment 22:** We agree that a detailed explanation is necessary here. Because the generated DEMs 10-seconds after the start of the runout could not measure the lobe surface due to the lack of data, we cannot mention differences in the slope accurately. Given this, as you mentioned, we will explain the difference based on the distance between the lobes to the flume outlet.

**R2 Comment 23:** *Line 124: Why is the series of events being described in terms of time? Time is not likely to be a controlling factor in how the debris flow behaves. The slope over which it is traveling is much more likely to be the control (along with the grain size distribution).*

**Reply to R2 Comment 23:** We sincerely appreciate this important comment and agree that the slope is one of the factors that directly controls how the debris flow behaves. In this study, we focused on how a single surge (i.e., a continuous flow) forms the debris-flow fan in accordance with the sediment

deposition and inundation, which is the difference of our study compared with previous related studies that focus on whether the differences of characteristics among surges influence on the fan morphology. These previous studies allowed to measure the slope by measuring the topography of lobes before and after supplying each surge. In contrast, because in this study the surge continuously moves until the end of an experimental run, we could not measure the changes in the surface slope of the fan during the fan-forming phase. This is an inevitable limitation arising from the difference in the study target. Given this, while highlighting the difference in the target of our study, we will explain time-series changes in the fan-forming processes by referring to new results with respect to the symmetry of the fan (please see **Reply to R2 Comment 25**).

**R2 Comment 24:** *Figures 5 – 9 are very poorly explained and hard to interpret. This is not helped by there being no explanation of what is meant by "Run" in the experiment.*

**Reply to R2 Comment 24:** We apologize for the confusion and will revise the related sentences to clarify what run indicates here (a run indicates an independent experimental case, also please see **Reply to R2 Comment 10 and 17**).

**R2 Comment 25:** *Lines 217-129: A numerical metric would help to compare the shape of the mono vs multigranular flows. Perhaps the angle of deviation from directly straight or a ratio of the left vs right side length?*

**Reply to R2 Comment 25:** We sincerely appreciate this important comment and agree that a numerical metric can help the comparison of topography between the mono- and multi-granular flows. Considering the simplicity of the definition, we will add the latter metric of the ratio of the length.

To investigate differences in the fan shape between the mono-granular and multi-granular flows, we proposed an index that focuses on the symmetry of the fan shape. The proposed symmetric index (SI) is defined as

$$SI = LR/LL -(1)$$

where LR and RR are the length of the fan from the midline to the edge of the left and right bank side of the fan, respectively. When the fan width is close to symmetry, a SI value is approximately one. We calculated values of SI indices from the width of the fan at cross-sections per 0.2 m from the outlet of the flume, using orthophotos and DEMs 10, 20, 30, 40, 50 seconds after the start of the flow runout (**Figure R2**).

Time-series changes in the SR indices demonstrate the linkage between the sifting of the direction of the flow runout and fan development processes. The differences in the SI values among test runs became notable with the increase in the distance from the flume outlet in the measurements after 20 s, 40 s, and 50 s from the start of the flow runout (**Figure R2 c–d and g–j**). This increase in the asymmetry of the fan morphology corresponded to the timing of shifting the flow direction (i.e.,

avulsion process). Moreover, the range in the SI values among test runs widened in the multi-granular flows compared with that of the mono-granular flows (e.g., **Figure R2 g–h**). Therefore, the wide-ranged grain-size distribution within the debris flows could result in diverse deposition and inundation patterns when the avulsion occurred. We will improve the clarity of the results and discussion sections in the revised manuscript while explaining these new results in detail.

[Figure]

**Figure R2:** Time series changes in the SI values of the debris-flow fans. (a) mono-granular flows. (b) multi-granular flows. The left and right panels indicate mono-granular flows and multi-granular flows, respectively.

**R2 Comment 26:** *Line 135: The difference in shape at the 2.2m line could be due to the difference in runout. While in the monogranular flow the authors are measuring the apex of the flow height in the multigranular it after the apex. As the debris flows are producing a fan like shape you would expect the fan to be wider after the apex regardless of the granular structure.*

**Reply to R2 Comment 26:** The difference in the runout distance is closely linked to the difference in the fan morphology between the mono-granular and multi-granular flows, which is responsible for the difference in the changes in the flow directions during the fan-forming processes. As you mentioned, we also should note the location of the fan apex and Figure 10g–i measured the difference in the shape lower than the fan apex carefully. Considering this, we will revise the third paragraph to clarify how the difference in the runout between mono-granular and multi-granular flows propagates to the differentiation in the fan shape.

**R2 Comment 27:** *Discussion: The discussion, similarly to the introduction, is lacking in detail and is too vague in some of its points to make an impact on the reader. Currently the discussion spends too much time focusing on areas the authors did not investigate (pore fluid seepage) and not enough time putting their results back into the context of the literature. To help the reader the authors should put their results, which are interesting and novel, front and centre and discuss the processes that they actually recorded before moving on to areas they do not have direct evidence for.*

**Reply to R2 Comment 27:** We agree that the discussion section too much focuses on the impact of the pore fluid seepage, rather than discussing the obtained results in detail. In the revised manuscript, we will mainly focus on the description of how flows provide the fan with different shapes in accordance with the difference in the grain-size distribution. Then, we will discuss the mechanisms that potentially form different fan shapes (please see **Reply to R1 Specific comment 14** and **Reply to R2 Comment 25**).

**R2 Comment 28:** *Lines 141 – 142: The term "processes" is too vague and the results do not mention stratigraphy at all so this seems like a strange sentence to start the discussion with.*

**Reply to R2 Comment 28:** We will revise this sentence to focus on the difference in the fan morphology.

**R2 Comment 29:** *Lines 142 – 147: This section is poorly linked to the previous opening statement of the discussions.*

**Reply to R2 Comment 29:** Again, according to this comment and **R2 Comment 28**, we will revise this sentence to focus on the difference in the fan morphology.

**R2 Comment 30:** *Lines 150 – 151: This would also apply if the flow was monogranular with coarse grains.*

**Reply to R2 Comment 30:** We agree with this comment. The coarse monogranular grains can

increase the flow resistance but also may decrease the entrainment rate in the channel. To control the flow state in the channel between the monogranular and multigranular debris flows, we carefully designed the grain-size distribution of the multi-granular flow. Specifically, we intended to generate the flows with a similar entrainment rate (i.e., flow depth) without the phase shift of relatively small sediment particles. While highlighting our intention of the experimental setup, we will revise the main text.

**R2 Comment 31:** *Lines 152 – 155: The authors previously mentioned that there was minimal difference between the thicknesses of the mono and multi-granular flows. This idea of the coarser grains forming an obstacle which diverts the tail of the flow should be expanded upon further with more descriptions if the authors believe it to be significant.*

**Reply to R2 Comment 31:** We agree with this comment. By referring to new results adequately, we will expand the explanation regarding this idea (please see **Reply to R1 Specific comment 14**).

**R2 Comment 32:** *Lines 158 – 164: It is unclear what the authors are suggesting here. How can there be moisture differences in the bed of the depositional area? The depositional area is the same for all of the test runs? Unless they are discussing deposition on a previously deposited fan? This is very underdeveloped.*

**Reply to R2 Comment 32:** We agree with this comment. Again, by referring to new results adequately, we will expand the explanation regarding this idea (please see **Reply to R1 Specific comment 14**).

**R2 Comment 33:** *Lines 165 – 169: This section is a strange ending to the manuscript. It focuses on two areas which the authors did not measure in their study; stratigraphy and moisture content of the fan.*

**Reply to R2 Comment 33:** We agree with this comment. Based on new results with respect to the phase separation and fan morphology, we will revise the discussion section thoroughly (please see **Reply to R1 Specific comment 14** and **Reply to R2 Comment 25**).

**R2 Comment 34:** *Conclusions: The conclusions does not contain any references to the discussion and therefore it feels disconnected from the rest of the manuscript.*

**Reply to R2 Comment 34:** We agree with this comment. Taking account into how the phase separation and fan asymmetry influence fan-forming processes, we will revise the discussion section. By referring to our new findings and assertion in the revised discussion section, we will improve the clarity of the revised conclusion section.

---

## Author Response (AR1)

We are grateful for the constructive and detailed comments of both reviewers. In accordance with their helpful and insightful recommendations, we have thoroughly revised the original manuscript to clarify the results and to emphasize the significance of this research.

Below, we provide our responses to the specific comments and questions received from the reviewers. The comments of the reviewers are shown in italics and different colors (i.e., those of Reviewers 1 and 2 are presented in brown and blue, respectively), while our responses are provided in normal black text.

**Reviewer 1 COMMENTS AND AUTHOR RESPONSE**

**R1 General comment 1:**

*The authors investigate how the grain-size distribution of debris flows affects the fan-forming processes. For this, flume tests have been conducted to compare the debris-flow fan morphology under varying sediment source grain-size distributions. The obtained results associated with the debris-flow runout and space-time variations in the fan morphology provide important insights into how the grain-size distribution affects the fan formative processes. The topic is interesting, investigation is novel, and is within the scope of esurf. However, the ms requires a thorough revision in concept, mechanics and dynamics. Detailed suggestions and comments are provided below hopping that they will help to improve the quality, consistency and clarity of the revised ms.*

**Reply to R1 General comment 1:** We sincerely appreciate your thorough and helpful review. In accordance with your comments, we have carefully revised the manuscript to clarify our findings and the assertions made throughout the text in terms of the mechanics and dynamics.

**R1 General comment 2:**

*Abstract can be improved. E.g., "Grain-size distribution was closely related to spatial diversity in fan morphology and stratigraphy." Should be other way round, "Spatial diversity in fan morphology and stratigraphy were found to be closely related to grain-size distribution."*

**Reply to R1 General comment 2:** We appreciate your helpful and constructive guidance. We have thoroughly revised the abstract (**P1 L9-25**) as shown below:

**P1 L9-25:** Knowledge of how debris flows result in the fan-shaped morphology around a channel outlet is crucial for mitigation of debris-flow-related disasters and investigation of previous sediment transport from the upper channel. Therefore, using flume tests, this study conducted fan-morphology experiments to assess the effects of differences in grain-size distribution within debris flows on changes in fan morphology. Two types of debris-flow material, i.e., monogranular particles comprising monodispersed sediment particles and multigranular particles comprising polydispersed sediment

particles, were used to generate monogranular and multigranular experimental debris flows, respectively. By adjusting the average grain size coincident between the monogranular and multigranular flows, we generated two types of debris flow with similar flow properties but different grain-size distributions. Although the flow depths were mostly similar between the monogranular and multigranular flows before the start of the debris-flow runout, the runout distances of the front of the multigranular flows were shorter than those of the monogranular flows. The difference in runout distance was responsible for the variations in the extent to and location in which the debris flows changed their direction of descent, resulting in the different shapes and morphologies of the fans in response to grain-size distribution. Although the direction of descent of the flows changed repeatedly, the extent of morphological symmetry of the debris-flow fans increased at a similar time during fan formation irrespective of the grain-size distribution. In contrast to this similarity in the rate of change in fan symmetry, the shift of the multigranular flow directions eventually increased the extent of asymmetry in fan morphology and expanded the scale of deviations in fan morphology between experimental test runs. Therefore, wide-ranging grain-size distributions within debris flows likely result in complex fan morphology with a high degree of asymmetry.

**R1 General comment 3:** *Introduction is not that much concerned about the main topic on how the grain-size distribution influences the debris flow fan morphology. So, it needs to be substantially expanded focusing on how the grainsize of the source material affects the deposition and fan formation process including the important and often dominant dynamical processes - the material separation, erosion and run-out dynamics.*

**Reply to R1 General comment 3:** To highlight our main topic of how grain-size distribution influences fan morphology, we have revised the introduction in light of the debris-flow dynamic processes as shown below (**P1 L27-88**):

[revised manuscript text omitted]

**R1 General comment 4:** *Often the writing is not explicit, not smooth and difficult to follow. It seems if the ms was made for very short paper, less for a professional journal, requiring clearer and smoother presentation. Particular attention should be given on these issues.*

**Reply to R1 General comment 4:** We apologize for the confusion. To meet the requirement for clearer and smoother presentation, we have thoroughly revised the manuscript in terms of clarity, overall language, and terminology and we believe that the standard of writing is now of the required level and quality throughout the text.

**R1 Specific comment 1:** *L36-37: "Changes in the physical parameters (e.g., flow rate, duration, and*

*sediment concentration)": The writing must be significantly improved, conceptually: flow rate, duration, and sediment concentration are not the physical parameters, rather they are the dynamical quantities. Physical parameters include densities, viscosities, frictions, slope geometry, curvature, etc.*

**Reply to R1 Specific comment 1:** We agree with this comment. As part of the changes made to this section, such vague terms have been deleted.

**R1 Specific comment 2:** *L38-39: "and sediment entrainment rate (Egashira et al., 2001; De Haas and Van Woerkom, 2016)." better change to "and sediment entrainment rate (Egashira et al., 2001; De Haas and Van Woerkom, 2016; Pudasaini and Fischer, 2020: https://doi.org/10.1016/j.ijmultiphaseflow.2020.103416) and separation between the particles and fluid in the mixture (https://doi.org/10.1016/j.ijmultiphaseflow.2020.103292)."*

**Reply to R1 Specific comment 2:** We agree with this comment. As part of the changes made to this section, this wording has been deleted, but such aspects have been considered in the revised introduction (please see **Reply to R1 General comment 3**).

**R1 Specific comment 3:** *L39-40: "and depending on the topographic complexity, could produce varying functional changes in subsequent debris flows.": not clear how topography produces functional changes and which? "functional and structural changes" what are the functional and structural changes? General readers may not be able to follow the text.*

**Reply to R1 Specific comment 3:** We agree with this comment. As part of the changes made to this section, such vague terms have been deleted.

**R1 Specific comment 4:** *L50: "A straight flume (8 m long and 0.1 m wide, with a uniform 15° bed slope": This is not true, and the text around must be improved appropriately, consistent with the corresponding figure.*

**Reply to R1 Specific comment 4:** We apologize for the confusion. The text was correct. For consistency between the text and the figure, we have revised the dimensions of the flume shown in Fig. 1 as shown below:

[Figure]

**Figure 1:** Test flume setup. (a) Dimensions of the test flume and equipment. (b) View of the flume and the deposition area. (c) Grain-size distribution of the sediment materials used in the experiments. Figure modified from Tsunetaka et al. (2019).

**R1 Specific comment 5:** *L50-67: "erodible bed conditions": Here comes the great thing! I see two major aspects in this ms. First, as the authors say, the effect of particle size in the erodible bed and how it will affect the deposition fan. I understand this differently than the author, it is not the particle size in initial debris mass (initially water is released) that will influence the deposition fan, but it is how the particle size of the erodible bed that affects the deposition fan when that bed substrate is eroded and entrained by the water flood released from upstream, consequently forming a debris mixture that ultimately flows down and deposits in the gentle open flat slope forming debris fan. The text does not mention this. Second, probably even more important, is the fact that the flood erodes and entrains the granular bed converting it in to the debris flow. So, the physical process of erosion, entrainment and the associated mobility must be described. This could however be done with respect to the mechanical erosion rate models and the mechanical model for the mass flow mobility with erosion (Pudasaini and Krautblatter, 2021: https://www.nature.com/articles/s41467-021-26959-5). I would focus on these governing aspects of experiments.*

*Writing style needs to be made more appropriate with better physical understanding. "The supplied water generated a granular flow that imitated a single debris-flow surge and then entrained the*

*erodible bed to the deposition area" This is difficult to follow, probably not representing reality. Does the water flow first generate the granular front by entraining the granular bed? I guess, as the water front impacts the granular bed it will erode and entrain the grain, mixing will take place resulting in the subsequent debris flow. Not that the way the authors explained. Moreover, you mixed up erosion and entrainment, which are clearly two different mechanical processes as proven by the reference mentioned above. So, the process of erosion and entrainment should be carefully investigated/discussed.*

**Reply to R1 Specific comment 4:** We sincerely appreciate this important comment. We have read the paper by Pudasaini and Krautblatter (2021) and we interpret your concern as explained below:

1. Pudasaini and Krautblatter (2021) derived a new theoretical framework called the three-E (erosion–entrainment–energy) mechanical concepts.

2. They defined two velocities: erosion velocity is the velocity that bed substrate is fed by the debris flow (i.e., the velocity vector intersects the flow line) and entrainment velocity is the velocity that debris flow transports (entrains) the eroded bed materials. These velocities are different.

3. Erosion velocity contributes to the momentum transfer in the debris flow, whereas entrainment velocity contributes to changes in the inertia of the debris flow. Thus, depending on the relationship between erosion velocity and entrainment velocity, the mobility of the debris flow can change.

4. Because in our experiments there was the possibility that these velocities were different between the monogranular material runs and the multigranular material runs, you have wondered whether the debris-flow mobility was also different between the monogranular and multigranular flows.

We agree that this new framework could be the key to the theoretical unraveling of the sediment erosion and deposition processes of debris flows. There is the possibility that our experiments were not adequate for direct assessment of debris-flow dynamics based on the proposed 3E concepts. The beds in both the flume and the deposition area (connected plane) gradually changed morphology (slope), meaning that we could not trace the boundary between the flow bottom and bed surface easily, which prevented measurement of erosion rate. Because of this limitation, most previous studies relied on measurement of changes in the sediment concentration of debris flows rather than the erosion rate (e.g., Lanzoni et al., 2017). Indeed, even Pudasaini and Krautblatter (2021) recognized the difficulty of demonstrating their 3E model using existing flume test and field data.

In light of the above, the importance of the difference between the erosion and entrainment velocities has been considered in the revised introduction (please see **Reply to R1 General comment 3** for details of the revision of the introduction section) and discussion sections (**P13 L387-390**).

Additionally, we have revised related sentences to use the terms "erosion" and "entrainment" properly (e.g., **P2 L53-64**, please see **Reply to R1 General comment 3**).

**P13 L387-390:** It is noteworthy that the fronts of the solid phase of the multigranular flows continued their runout after the start of phase separation (Fig. 3b), which is different from the coincidental start of phase separation and halting of the front of the solid phase in the monogranular flows (Figs. 3a and 4). The solid-phase runout after the start of phase separation in the multigranular flows implies that the solid phase retained sufficient momentum to entrain and transport sediment particles. In this sense, deposition of the solid phase of the monogranular flow fronts was caused by the complete and sudden stop of the solid phase (i.e., sediment particles), which might be physically different from that of the multigranular flow fronts. Theoretical analysis of debris-flow mechanics that carefully divides the effects between the erosion velocity (i.e., the velocity of sediment erosion from the bed by the flow) and the entrainment velocity (i.e., the velocity of the transportation of eroded sediment by the flow) demonstrated that the contribution to flow momentum is different between the erosion and the entrainment velocities; consequently, their differences are responsible for the variation in the fluidity of debris flows (Pudasaini and Krautblatter, 2021). Detailed analysis of the difference between the erosion (deposition) and entrainment velocities is difficult owing to limitations of the experimental setup. However, the different trends in runout between the monogranular and multigranular flows highlight that further understanding of the erosion/deposition mechanisms is crucial for accurate estimation of debris-flow deposition range.

**R1 Specific comment 5:** *"the fan morphology gradually formed in accordance with the runout and inundation of the released granular flow": This is rather the fan of the granular-water mixture debris flow. For a granular fan, you must only have the dry material without water, for which the fan will be substantially different than what it is now. Two types of granular flow, namely mono-granular and multi-granular, were used to determine the impact of grain-size distribution within a debris flow on the fan-forming processes.": This is not right. You have two types of granular materials in the erodible bed, resulting in two types of debris flows, one consisting of water and mono-granular material, another composed of water and multi-granular [this term needs to be defined carefully, as mono-granular and multi-granular are not the usual terms, usual terms are the mono-dispersed and poly-dispersed, with respect to grain size, etc.].*

**Reply to R1 Specific comment 5:** We sincerely appreciate this comment and have revised the related sentences to avoid this confusion as shown below (**P4 L106-116**):

**P4 L106-116:** By suddenly supplying water from the upper end of the flume, we generated a granular–water mixture flow that imitated a debris flow, similar to Lanzoni et al. (2017). We could not control the erodible bed saturation completely because the bed materials included voids. Fully saturated bed conditions were approximated by carefully supplying clear water across the entire erodible bed using watering cans just before we initiated the water supply from the upper end of the flume. Following this operation, a steady flow of clear water (fluid density: ~1000 kg m$^{-3}$) was supplied at a rate of 0.003

$m^3 s^{-1}$ for 60 s from the upper end of the flume. The supplied water plunged over the erodible bed and flowed downstream, generating a runoff front over the bed sediment particles. The runoff front scoured the sediment particles of the erodible bed and entrained the eroded material, dispersing the entrained particles throughout the flow depth, and eventually transforming the flow into a granular–water mixture that imitated a single debris-flow surge (Lanzoni et al., 2017). The generated granular–water mixture flow descended to the deposition area, causing runout and inundation, which formed the fan morphology. The slope of the deposition area decreased from 12° to 3° at a rate of 3° per meter (Fig. 1a, b).

Additionally, in an earlier part of the revised manuscript, we provide definitions of the monogranular and multigranular flows as shown below (**P4 L118-126**):

**P4 L118-126:** In this study, we generated granular–water mixture flows with similar flow properties but different grain-size distributions to compare the effects of debris-flow grain-size distribution on fan morphology during the debris-flow runout and inundation processes. To accomplish this, two types of sediment particles were used to generate two types of granular–water mixture flows: monogranular particles comprising quasi-monodispersed sediment particles with size of 2.02–3.24 mm (average grain size, D50: 2.6 mm) and multigranular particles comprising polydispersed sediment particles with size of 0.6–7.5 mm (Fig. 1c, Table 1). The density and the internal friction angle of both particles were 2640 kg $m^{-3}$ and 34.0°, respectively (Hotta et al., 2021). Hereafter, the granular–water mixture flows generated by the monogranular particles and by the multigranular particles are referred to as monogranular flow and multigranular flow, respectively. We conducted four separate experimental test runs for both the monogranular flow and the multigranular flow.

**R1 Specific comment 6:** *Not enough information on the material and channel are provided, e.g., the basal and internal friction angles of the granular material, viscosity of water, their densities, and so on. This information is crucial in understanding erosion-entrainment and mobility, the mixing and separation between particles and fluids, and the transport/deposition of debris mixture.*

**Reply to R1 Specific comment 6:** We agree with this comment and have added such information in the revised manuscript (please see **Reply to R1 Specific comment 5**).

**R1 Specific comment 7:** *Another principle concern is the representative grainsize, the two granular materials, mono-dispersed and poly-dispersed are represented by the same average grain size (D50). This does not help to physically clearly study the erosion-entrainment, transport and deposition fans, except that you can say – we observe this and that for the mono-dispersed and poly-dispersed erodible bed. But, we don't know how small and big particles in the mixture influence the erosion, mixing or separation, dynamics and deposition processes. Moreover, different grains might need to*

*be represented by different rheological equations. These are crucial aspects the authors should discuss. Otherwise the results can not be understood mechanically and dynamically clearly, and these data cannot that easily be used in model validation and parameter calibration.*

**Reply to R1 Specific comment 7:** We agree with this comment. By referring to new results on how differences in grain-size distribution lead to variation in phase separation and symmetry of fan morphology (Fig. 3 in the revised manuscript), we have improved the discussion section in light of known dynamics with respect to sediment erosion/deposition, flow runout, and phase separation (please see **Reply to R1 Specific comment 14** and **Reply to R2 Comment 25**).

**R1 Specific comment 8:** *L69-70: "sediment was released to the deposition area". A bit strange writing. First, it is not sediment, it is the debris material. Second, it is not released to the deposition area, it is the transported material in the deposition area. So, the dynamical perspectives are weak.*

**Reply to R1 Specific comment 8:** We have revised the sentence in accordance with your comment (please see **Reply to R1 Specific comment 5**).

**R1 Specific comment 9:** *L70-73: "The flow depth of a generated granular flow cannot be measured in the flume because the thickness of the erodible bed decreases sequentially in response to the sediment entrainment. Therefore, the displacement of the flow surface at three positions in the flume (upper, middle, and lower, Fig. 1a) was measured to account for this shortcoming, using ultrasonic displacement meters":*
*I agree with the first sentence, it is a really complex process, however, there are some literature in this direction with some success (Lanzoni et al., 2017: https://doi.org/10.1002/2016JF004046). The authors should put some efforts to review relevant literature. The bed erosion process is an under-investigated process, and I respect any attempt in this direction. The second sentence is not the solution to the first, because, the measured flow depth cannot be split into the material from the flow and from the bed. Thus, it cannot be straightforward connected to the erosion depth. Furthermore, the involved energy associated with erosion is the dominant factor to decisively defining the dynamics, runout and the associated impact forces of the erosive mass flows. This needs to be discussed with respect to the references mentioned above.*

**Reply to R1 Specific comment 9:** We agree with your statement about the erosion/entrainment processes and the necessity for descriptions based on relevant literature. Through considered reference to the literature, we have revised the sentences regarding flow-depth measurements to improve clarity as shown below (**P5 L146-159**):

**P5 L146-159:** We measured the flow depths of the generated debris flows in the flume and

investigated the properties of runout and fan morphology at the deposition area. By comparing the changes in the flow depths between the monogranular and multigranular flows, we assessed whether the multigranular flows exhibited hydrograph and velocity characteristics analogous to those of the monogranular flows. Note that we could not directly measure the flow depths of the generated debris flows because the thickness of the erodible bed decreased sequentially in response to sediment erosion by the debris flow. The continuous sediment erosion in response to debris-flow descent made it impossible to define the boundary between the debris-flow bottom and the bed surface (e.g., Lanzoni et al., 2017), which hampered quantitative measurement of debris-flow depth. Instead of flow depth, we measured changes in the displacement of the flow surface using three ultrasonic displacement meters (Omron, E4PA) at a sampling rate of 50 Hz. The ultrasonic displacement meters were installed at three positions separated by a distance of 1 m above the sediment bed from upstream to downstream in the flume; hereafter, referred to as the upper, middle, and lower measurement positions, respectively (Fig. 1a). Because the initial depth of the erodible bed was adjusted to 0.2 m, the flow depths of the debris flows were calculated by subtracting this initial depth from the measured displacement. We compared the flow rate in the flume among the test runs on the basis of differences in the timing at which the debris-flow front reached the lower position.

**R1 Specific comment 10:** *I stop suggesting and commenting on the mechanical and dynamical aspects of the ms, and hope that the authors will improve the text while revising it.*

**Reply to R1 Specific comment 10:** We again express our sincere appreciation of your constructive suggestions. In accordance with your comments, we have carefully revised the manuscript to strengthen aspects in relation to debris-flow mechanics and dynamics.

**R1 Specific comment 11:** *Fig. 2: Figures are difficult to follow. It should be self-explanatory. For example, Run 1-4, are they repeated exps.?*

**Reply to R1 Specific comment 11:** "Run" indicates each experimental test run. We have revised the related sentences in the method section to convey the results more clearly as shown below (**P5 L125-126**):

**P5 L125-126:** We conducted four separate experimental test runs for both the monogranular flow and the multigranular flow.

**R1 Specific comment 12:** *L98-99: "The thickness of the erodible bed decreased monotonically with time, probably because the entrainment rate was the same in all the test runs, irrespective of the grain-size distribution of the granular flows": This cannot be true, could only be a speculation. Because, as proved in the above-mentioned references, erosion rate is a complex phenomenon, and changes*

*with the dynamic load applied by the flow and resisted by the bed. This needs to be discussed.*

**Reply to R1 Specific comment 12:** We appreciate this comment. We have revised the related sentences to avoid such speculation (**P7 L209-218**) and to discuss the importance of the difference between erosion and entrainment velocities (please see **Reply to R1 Specific comment 4**).

**P7 L209-218:** Changes in surface height at the upper measurement position indicate that the erodible bed was gradually eroded after the arrival of the flow front irrespective of the grain-size distribution (Fig. 2a, b). After ~22–23 s, the surface heights of the multigranular test runs decreased to below 0.15 m (Fig. 2b), whereas those of the monogranular test runs were >0.15 m at the same time (Fig. 2a), indicating that bed material was eroded at a slightly faster rate by the multigranular flows than by the monogranular flows. In relation to this difference, the fronts of the multigranular flows reached the middle measurement position somewhat faster than those of the monogranular flows, although the differences in arrival time were <1 s between the monogranular and multigranular test runs (Fig. 2c, d). Focusing on the flow fronts, both the monogranular flows and the multigranular flows descended the flume from the upper to the lower measurement positions in ~6–7 s (Fig. 2e, f). Given an initial erodible bed thickness of approximately 0.2 m, the peaks of the monogranular and multigranular flows developed from ~0.03 to 0.07 m before reaching the flume outlet.

**R1 Specific comment 13:** *L100-102: "Overall, the results from the flume experiment showed that the difference in the grain-size distribution did not lead to substantial changes in the hydrograph and arrival time of the granular flows.": I can't fully agree with this. E.g., if you take the mean of four runs in C and F and plot them in one figure, you will see discernible difference.*

**Reply to R1 Specific comment 13:** We agree that the tone of the statement in the original text was too strong, and we have carefully explained the differences and similarities between the monogranular and multigranular flows as shown below (**P8 L229-234**, please see **Reply to R1 Specific comment 12**):

**P8 L229-234:** Following the descent of the flow front, the rate of decrease in surface height was found increasingly similar between the test runs irrespective of the grain-size distribution within the debris flows (Fig. 2). It indicates that the thickness of the erodible bed decreased monotonically with time in accordance with the erosion and entrainment of sediment by the flow body and tail. Overall, the results derived from the flume experiments revealed that differences in grain-size distribution did not lead to substantial changes in the hydrograph and arrival time of the generated granular flows in the flume, with the exception of the peak of flow depth.

**R1 Specific comment 14:** *L113: Grain size separation is one aspect, but separation between particles and fluid (as seen in the experimental results) is another, even more complex mechanical phenomenon in debris flow. However, the authors did not discuss anything on it.*

**Reply to R1 Specific comment 14:** We sincerely appreciate this important suggestion and agree that separation between particles (sediment) and fluid is one of the important aspects. We measured the time series changes in runout distance of the fronts of the generated debris flows with temporal resolution of 0.1 s using captured video and the grid lines drawn in the deposition area (**Figure 3**). We found different trends with respect to separation between the solid (sediment particles) and fluid phases. In comparison with monogranular flows, the phase separation of the multigranular flows occurred in an earlier stage of the runout process. We have explained and discussed these results in the revised manuscript as shown below (**P8 L236-255**):

[Figure]

**Figure 3**: Change in runout distances of the flow fronts with time: (a) monogranular flows and (b) multigranular flows. Continuous and broken lines indicate runout distances for the fluid and solid phases, respectively. Black dotted lines are assumed graphs for velocities of 0.5 and 1 m s$^{-1}$.

**P8 L236-255:** Characteristics of debris-flow runout were clearly different between the monogranular and multigranular flows. Before the runout distance exceeded 1 m, flow velocities (i.e., the slope of the graphs) differed somewhat between the test runs irrespective of the grain-size distribution within the debris flows (Fig. 3), which was likely attributable to variation in the peak of flow depth (Fig 2e, f). At this stage, the solid and fluid phases of both types of flow descended together as a single complete mixture flow, and their velocities were synchronized with each other.

After the runout distance of the flow fronts exceeded ~1.0 m, the velocities of the monogranular flows decreased gradually with increase in runout distance, but the velocities of the solid and fluid phases remained analogous (Fig. 3a). However, the trend of the multigranular flows differed. The velocity of the solid phase of the multigranular flows decreased rapidly, which increased the relative difference in the velocities of the flow fronts between the solid and fluid phases of the multigranular flows (Fig. 3b). The separation between the solid and fluid phases of the multigranular flows thus occurred at an earlier stage of the runout process in comparison with that of the monogranular flows (Fig. 3).

Following the start of phase separation of the multigranular flows, the solid phase continued its runout with further increase in the relative difference in the velocities between both phases, especially after the runout distance of the fluid phase exceeded 2 m (Fig. 3a). Before the runout distance exceeded 2 m, the velocities of the monogranular flows were similar to those of the fluid phase of the multigranular flows (Fig. 3). Subsequently, the monogranular flows decelerated, whereas the fluid phase of the multigranular flows maintained its velocity and descended at ~0.5 m s$^{-1}$. Consequently, the fluid phase of the multigranular flows traveled slightly faster and progressed further downstream. Phase separation of the monogranular flows occurred after the runout distance of the flow fronts exceeded ~2.7–2.8 m. Therefore, the runout distance and velocity differed not only between the monogranular and multigranular flows but also between the respective solid and fluid phases of these flows.

**R1 Specific comment 15:** *L123: "avulsed obviously": it is better also to put orthophoto to clearly see avulsion. The quality of Fig. 5-8 should be improved, with filters, or whatever means such that we can clearly see avulsion. The problem I have seen is that avulsion cannot be predicted, or was not possible with the present setup. We should understand why this is happening. This needs to be discussed, because, one of the main aims of experiments should be to generate reproducible results.*

**Reply to R1 Specific comment 15:** We apologize for the confusion, and we have revised Figs. 5–8 using the orthophotos (Figures 7–12 in the revised manuscript; as an example, Fig. 12 is shown below). Additionally, using metrics of the symmetry of fan shapes, we have revised the sentences regarding changes in the flow direction and the formation of fan morphology (please see **Reply to R2 Comment 25**).

[Figure]

**Figure 12:** Fan morphology 40 s after the start of runout of the multigranular flows. The upper and lower panels show orthophotos and digital elevation models (DEMs) with flow vectors, respectively. Respective sets of the upper orthophoto and lower DEM represent corresponding results of each experimental test run. The white arrows on the orthophotos indicate the assumed principal direction of flow descent. The elevation of the DEMs is depicted assuming that the area with a 3° slope (i.e., the area furthest downstream from where the slope angle changed from a 6° to 3° slope) has elevation of zero.

**R1 Specific comment 16:** *L142-144: "Some equations that describe debris flows assume that multi-granular debris flows can be approximated to mono-granular debris flows with the same average grain-size (e.g., Egashira et al., 1997; Takahashi, 2007).": This is not the state-of-the art. The multi-mechanical, multi-phase mass flow model by Pudasaini and Mergili (2019: https://doi.org/10.1029/2019JF005204) has proven the necessity of simulating debris flows as mixture of different materials, that has been used in accurately simulating complex multi-phase natural events (Mergili et al., 2020: https://doi.org/10.5194/nhess-20-505-2020; Shugar et al., 2021: DOI: 10.1126/science.abh4455). The ms should be up dated with relevant, recent literature.*

**Reply to R1 Specific comment 16:** We apologize for the confusion. In the revised manuscript, we

have discussed such problems of previous models and considered the perspective for accurate simulation using the recent multiphase model as shown below (**P13 L399-423**):

**P13 L399-423:** Importantly, in comparison with monogranular fans, the extent of asymmetry of the multigranular fans differed substantially between test runs (Figs. 14 and 15). The variations in the asymmetry of the multigranular fans suggest that debris flows can randomly shift their descent direction. Some models assume that multigranular debris flows can be approximated to monogranular debris flows with the same average grain size, which allows estimation of debris-flow properties such as flow velocity and depth (e.g., Egashira et al., 1997; Takahashi, 2007). Indeed, our flume-based experimental results exhibited similar flow velocity and depth as debris flows with the same average grain size but with different grain-size distributions (Fig. 2). However, given that the differences in runout characteristics resulted in different fan morphologies between the monogranular and multigranular flows, the use of debris-flow models that involve grain-size approximation could be responsible for inevitable uncertainty in the estimation of fan morphology formed by debris-flow runout. Even in the early stage of flow runout, i.e., after 20 s from the start of runout, the shape of the multigranular fans exhibited asymmetry in comparison with that of the monogranular fans (Fig. 14c, d), which was likely responsible for greater final asymmetry in multigranular fan morphology (Fig. 14i, j). It is likely that the short runout distance of the multigranular flows resulted in thick sediment deposition close to the flume outlet, and the swift phase separation accelerated the inundation of the fluid phase to the distal downstream area. In this sense, phase separation facilitated the increase in the extent of unsaturation of the fan deposits. A bed consisting of unsaturated sediment particles potentially decreases the pore-fluid pressure at the bottom of a debris flow and increases the resistance of the flow body (Major and Iverson, 1999; Staley et al., 2011), resulting in complex patterns of flow direction and sediment deposition (Tsunetaka et al., 2019). Thus, the variations in the extent of the saturation of the fan sediment materials facilitated by phase separation might have triggered the differences in the fan morphology between the multigranular test runs.

In this context, the extent of phase separation broadly constrains fan morphology. The advance in the multiphase model describing a granular–fluid mixture flow allows us to reflect on the effects of separation between the granular (solid) and fluid phases in numerical simulations (Pudasaini and Mergili, 2019), and to progress the theoretical interpretation of debris-flow mechanics. Our results demonstrate that further investigation of the relationships between the grain-size distribution within debris flows and the extent of phase separation and related changes in runout distance could lead to accurate forecasting of the range of debris flow deposition and inundation.

**R1 Specific comment 17:** *L144-145: "However, the mono-granular and multi-granular flows with the same average grain-size produced fans with different morphologies": This is probably the most important aspect of this ms, and I like it. However, it has not yet been clearly discussed for why this is so. The authors should put some energy to explore why it is happening, that will lift the importance*

*of this paper to a higher level.*

**Reply to R1 Specific comment 17:** We agree with this comment. We have discussed the relevance of the differences in fan morphology with regard to differences both in runout distance and in the timing of phase separation in the revised manuscript as shown below (**P12 L354-376**):

**P12 L354-376:** Relatively small and large particles within a debris flow can both influence changes in the runout distance of multigranular flow fronts (De Haas et al., 2015b; Hürlimann et al., 2015). In this study, the decrease in flow resistance due to small sediment particles was intentionally avoided by adjusting the composition of the multigranular flows. Indeed, the arrival time of the flow fronts in the flume was similar between the monogranular and multigranular flows (Fig. 2), suggesting that the effects of small particles on flow resistance were negligible. Unlike the unrelated small sediment particles, large sediment particles were accumulated in the multigranular flow fronts, at least during their runout (Figs. 5 and 6), and potentially caused the decrease in the runout distance (Fig. 4). Large sediment particles increase flow resistance and decrease flow velocity as bed slope decreases (e.g., Egashira et al., 1997; Takahashi, 2007). The velocity of the fronts of the solid phase of the multigranular flows decreased substantially when runout distance exceeded 1 m (i.e., when the front reached the point at which the bed slope decreased from 12° to 9°) in comparison with that of the monogranular flows (Fig. 3), suggesting that large particles caused a decrease in flow velocity. Thus, even when debris flows have hydrographs that are similar at the outlet of the channel, differences in the extent of accumulation of large particles in the flow front can lead to changes in runout distance and consequently form fans with different morphology.

Separation between the solid and fluid phases might be one of the principal mechanisms that alter runout distance. The fluid phase consisting of pore fluid in a multiphase-mixture flow generally acts to reduce flow resistance and drive flow descent (Takahashi, 2007; von Boetticher et al., 2016; Pudasaini and Mergili, 2019). The substantial decrease in the velocity of the front of the solid phase of the multigranular flows progressed phase separation during flow runout in the early stage (Fig. 3b), which increasingly can reduce the velocity of the solid phase owing to the absence of pore fluid. Numerical simulations that considered phase separation demonstrated that a strong front structure attributable to accumulation of solids in the flow front can lead to rapid phase separation (Pudasaini and Fischer, 2020). Thus, the large sediment particles that accumulated at the flow front of the multigranular flows potentially advanced phase separation during flow runout. Therefore, the increase in flow resistance of the multigranular flow fronts could have arisen owing to synergistic effects between the increase in the representative grain size of the solid phase and the decrease in the pore fluid by phase separation.

**R1 Specific comment 18:** *L146-147: "which indicates that existing models that assume a mono-granular approximation may provide ambiguous simulations of the debris-flow deposition and inundation ranges.": This proves the need of multi-phase mass flow models (mentioned above) in*

*properly simulating debris flows. This should be discussed.*

**Reply to R1 Specific comment 18:** We appreciate this comment. We have revised the discussion section with consideration of this aspect (please see **Reply to R1 Specific comment 16**).

**R1 Specific comment 19:** *Discussion and Conclusion, References:*
*Needs re-working, including the above suggestions. E.g., multi-phase flow simulations, erosion-entrainment and mobility, separating particles and fluid, and so on. Important point why the flow with the poly-dispersed erodible bed has shorter travel distance and run-out reveals that more energy has been consumed for this than the bed with mono-dispersed particles. This exclusively depends on the erosion velocity controlling the mobility of the mass flow, this fact has been proven by the mechanical model for the mobility of erosive mass flows by Pudasaini and Krautblatter (2021). The discussion and conclusion should give proper space for these important mechanical and dynamical aspects also observed in this ms.*

**Reply to R1 Specific comment 19:** We appreciate this comment and we have discussed this aspect by referring to new results (please see **Reply to R1 Specific comment 4**).

**R1 Technical comment 1:** *L22: "sinks", the meaning was not that clear, better would be "deposits"?*

**Reply to R1 Technical comment 1:** We appreciate this comment. As part of the changes made to this section, such vague terms have been deleted.

**R1 Technical comment 2:** *L29: Please check English.*

**Reply to R1 Technical comment 2:** We appreciate this comment. As part of the changes made to this section, this sentence has been deleted.

**R1 Technical comment 3:** *L85: "SfM-MVS": is its meaning clear?*

**Reply to R1 Technical comment 3:** We appreciate this comment. We have added adequate references as shown below (**P7 L185-186**):

**P7 L185-186:** We measured the process of fan-morphology formation in response to debris-flow runout and inundation using structure motion multi-view stereo (SfM-MVS) photogrammetry (Westoby et al., 2012; Fonstad et al., 2013).

**R1 Technical comment 4:** *L97-98: "while, apart from run 1, those of the multi-granular flows were*

*around ~0.03 m": Not true. Please check all the technical details carefully.*

**Reply to R1 Technical comment 4:** We appreciate this comment. We have revised the related sentences appropriately (please see **Reply to R1 Specific comment 12**):

**R2 Overall comment:** *In this manuscript the authors used a flume to analyse how the grain size distribution of a debris flow may impact the morphology of the resulting fan. The authors ran 2 sets of experiment runs in their flume, all parameters inside the experiments were kept constant except the grainsize distribution. One set of experiment runs used monogranular sediment while the other used multigranular. Both sets had the same average grain size so the authors could identify the impact of variations in grain size on the debris flow fan. They quantified these changes by measuring the surge height in the flume, the speed of the surge within the deposition area, the runout distance and a DEM of the final debris flow. The authors discovered that multigranular debris flows were more likely to produce alluviations in the debris flow resulting in asymmetrical fans. They postulate that these alluviations are the result of grain size segregation occurring within the flow where coarse sediment is forced to the front of the flow where it can produce an obstacle for any following surges.*

*I think this study is an interesting addition to the literature on debris flows. The experiments seem well thought-out and the results aim to fill a clear knowledge gap. However, there are several areas where I feel the manuscript needs to be improved before it is ready for publication. The manuscript is very short and as a result I feel that significant detail is missing, particularly from the description of the experiment design, results and discussion. I also found the figures poorly cited and discussed throughout. I have provided more specific comments below for the authors to read through. I hope the authors find my comments useful and I look forward to seeing the revised manuscript.*

**Reply to R2 Overall comment:** We sincerely appreciate your insightful review and constructive suggestions. In accordance with your comments, we have thoroughly revised the manuscript to improve clarity and the level of detail in all sections. Our response to each of your specific comments is provided below.

**R2 Comment 1:** *Introduction: The introduction is too short and vague to be of use to the reader. Despite the research statement at the end of the section it is not completely clear how the authors see the study contributing to the literature. It is also not obvious from these paragraphs why the authors have chosen to focus on grain size distribution for this study rather than many of the other controlling factors highlighted here. Finally, this section would be greatly improved if there was better separation between discussing debris flow physics from debris flow fans. Currently it is very confusing whether the authors are referring to how a certain parameter might affect a debris flow or how it may affect the autogenic fan forming processes or the links between the two.*

**Reply to R2 Comment 1:** We agree with this comment and have thoroughly revised the introduction section (please see **Reply to R1 General comment 3**).

**R2 Comment 2:** *Lines 25 – 26: What specific climate and sediment dynamics information can be identified from debris flow fans? And how is it derived? How is the form (which is what is investigated here) important?*

**Reply to R2 Comment 2:** We appreciate this comment. As part of the changes made to this section, this sentence was revised as shown below (**P1 L27-37**):

**P1 L27-37:** Debris flows often cause damage to downstream communities and infrastructure through their runout and associated sediment deposition (Dowling and Santi, 2014). Understanding how debris flows manifest around the channel outlet is important for mitigation of their impact on downstream areas and for prevention of related hazards. Debris flows often occur with various recurrence intervals and different magnitude in the same watershed (Jakob et al., 2005; Brayshaw and Hassan, 2009; Frank et al., 2019). Sediment deposition attributable to such debris flows leads to the formation of the fan-shaped morphology around the channel outlet, i.e., the so-called debris-flow fan, which is recognized as a geomorphological record of sedimentary sequences driven by past climatic and environmental conditions (Dühnforth et al., 2007; De Haas et al., 2015a, 2019; Schürch et al., 2016; D'Arcy et al., 2017; Kiefer et al., 2021). In this sense, studies on the morphology and the stratigraphy of debris-flow fans are fundamental to interpretation of previous sediment dynamics and their drivers. Assessing how debris flows result in fan morphology around a channel outlet is therefore crucial both for investigation of sediment transport episodes and for mitigation of debris-flow-related disasters.

**R2 Comment 3:** *Lines 26 – 29: If debris flow fans are primarily formed by autogenic processes how can information on any external forcing be derived from them?*

**Reply to R2 Comment 3:** By avoiding the use of the term "autogenic," we have revised the sentences with respect to the formation of fan morphology as shown below (**P2 L38-52**):

**P2 L38-52:** The morphology of a debris-flow fan is governed mainly by three processes that are driven by the runout and deposition of debris flows. Debris-flow surges are stacked stepwise around the outlet of the channel while backfilling the existing channel on the fan (De Haas et al., 2016, 2018a). The backfilling process reduces the flow capacity of the existing channel by decreasing the surface gradient, which consequently results in avulsion that shifts the flow direction of subsequent debris-flow surges (De Haas et al., 2016, 2018a). The avulsion of a debris-flow surge involves erosion of the sediment of the fan that leads to channelization on the fan (De Haas et al., 2016, 2018a). The newly formed channel will then be backfilled when further debris-flow surges occur, thereby repeating the fan-forming cycle of the backfilling, avulsion, and channelization processes (De Haas et al., 2016,

2018a). Monitoring in situ debris-flow runout around a channel outlet is difficult because of the low frequency of occurrence of debris flows (e.g., De Haas et al., 2018a; Imaizumi et al., 2019). However, earlier experiments using a reduced-scale flume demonstrated that the composition (grain-size distribution) and sequence of debris-flow surges govern the formation of fan morphology and the tempo of the fan-forming cycle (De Haas et al., 2016, 2018b; Adams et al., 2019; Tsunetaka et al., 2019). Moreover, relationships derived between the sequence of natural debris-flow lobes and fan morphology indicate that the fan-forming cycle is driven by the backfilling, avulsion, and channelization processes, similar to the results obtained from flume tests (De Haas et al., 2018a, 2019).

**R2 Comment 4:** *Lines 31-33: Shifting the focus of the paragraph from a geological perspective of fans to one about hazard is confusing to the reader particularly as neither focus is well covered.*

**Reply to R2 Comment 4:** We agree with this comment. We have described those aspects with regard to debris-flow hazards and records of sediment regimes as a general indication of the importance of studies on debris-flow fans in the first paragraph of the introduction (please see **Reply to R2 Comment 2**).

**R2 Comment 5:** *Lines 34-36: It would be useful to the reader if the authors would elaborate on how these physical factors affect morphology and stratigraphy of the fans.*

**Reply to R2 Comment 5:** We have revised the introduction section in light of the known dynamics of debris flows (please see **Reply to R1 General comment 3**).

**R2 Comment 6:** *Lines 36 – 38: From this section the reader cannot tell how these changes will affect the debris flows. The authors do not define the property that is changing carefully nor do they describe the impact of these changes on debris flow behaviour. Without this information the readers can not make the link between debris flows and the resulting fan.*

**Reply to R2 Comment 6:** To take account of this comment and **R2 Comment 4**, vague terms such as functional and structural changes have been avoided in our thorough revision of the introduction section (please see **Reply to R1 General comment 3**).

**R2 Comment 7:** *Lines 44 – 48: This is a good succinct research statement however it is completely disconnected from the preceding 2 paragraphs. It does not mention why the authors have chosen to focus on grain size distributions nor how they expect them to change the debris flow fan.*

**Reply to R2 Comment 7:** To convey our intention and to explain why we designed the experiments

to control only differences in grain-size distribution, we have thoroughly revised the introduction section (please see **Reply to R1 General comment 3**).

**R2 Comment 8:** *Methods: The experiment design is reasonably well explained, however I struggled to understand what exactly was being measured. The authors have gone to great lengths to capture the vast amounts of data generated by the experiments; however, they do not discuss why they collected these particular datasets or what they plan to do with them. A better motivating statement within the introduction will help to improve this section.*

**Reply to R2 Comment 8:** We agree and have provided adequate context at the end of the introduction section for smooth connection to the methods section (**P3 L81-88**). Additionally, we have carefully described why we measured both flow depth in the channel and changes in fan morphology to convey our intention to examine whether debris-flow surges with different grain-size distributions could provide different fan morphologies even if their flow properties were similar.

**P3 L81-88:** The primary objective of this study was to assess how the grain-size distribution within a debris flow influences fan morphology, especially during debris-flow runout and inundation. We conducted reduced-scale flume tests to compare fan morphologies that resulted from single debris-flow surges with different grain-size distributions but with similar flow properties. Using photogrammetry and video-image analysis, we investigated how differences in grain-size distribution within debris flows influence variations in runout characteristics and fan morphologies. The intention underlying this comparison was to interpret the differences in fan morphology in terms of known debris-flow mechanics. The final objective was to elucidate whether differences in grain-size distribution within debris flows could change fan morphology solely by influencing the runout process without variation of the dynamic properties of the debris flow in the channel.

**R2 Comment 9:** *Lines 50-60: The authors have not explained how the debris flow surge is generated. The paragraph could be separated to first describe how and where the surge is generated before discussing the depositional area.*

**Reply to R2 Comment 9:** With reference to appropriate literature and following consideration of earlier reviewer comments, we have revised the manuscript to carefully explain the process through which we generated the debris-flow surges (please see **Reply to R1 Specific comment 4**).

**R2 Comment 10:** *Lines 61-67: How fans are produced in this study? Is the fan a result of a single surge triggered by the outlet of water? Or the result of multiple surges? Is the erodible layer rebuilt between surges? What is going to be measured as a result of these experiments?*

**Reply to R2 Comment 10:** We focused on the debris-flow fan formed by a single debris-flow surge. Each experimental result (each test run) provided data on the fan formed by a single surge rather than by multiple surges. We have explained the logic behind the design of the setup and conditions in descriptions of our research intentions (please see **Reply to R1 Specific comment 5**).

**R2 Comment 11:** *Lines 69-74: Why is flow displacement being measured? Why is the flow height of the surge important to the authors?*

**Reply to R2 Comment 11:** We intended to clarify the differences and similarities between the monogranular and multigranular flows in the flume using measurements of flow height. We have explained our intention and how the measured depth helps our interpretations (please see **Reply to R1 Specific comment 9** and **Reply to R2 Comment 8**).

**R2 Comment 12:** *Lines 79-80: What is being measured and how?*

**Reply to R2 Comment 12:** To check the accuracy of the SfM photogrammetry, we compared the deposition depths of the experimental debris-flow fan that were measured directly after each experimental run using a ruler. We have explained this procedure in detail in the revised manuscript as shown below (**P7 L190-194**):

**P7 L190-194:** To assess the DEM accuracy, we used a ruler to directly measure the deposition depths of the fan morphology at the intersections of the grid lines after each respective experimental run, and we compared the measurements with the deposition depths extracted from the generated DEM. The measured elevations corresponded to the DEM-extracted elevations, indicating that the DEMs represented reasonable approximations of the fan morphology (Fig. S1).

**R2 Comment 13:** *Line 86: It is not clear what is meant by "the SfM-MVS photogrammetry could not measure locations where granular flows descended". Does this mean that the photogrammetry cannot measure the flow when it is moving?*

**Reply to R2 Comment 13:** Yes, the moving flow produced a complex surface with undulations and varying brightness in the captured images, which resulted in unmeasurable zones of the photogrammetry results. We have clarified this point in the revised manuscript as shown below (**P7 L194-198**):

**P7 L194-198:** During debris-flow inundation in the deposition area, the SfM-MVS photogrammetry could not perform measurements for locations in which flows descended (i.e., moving zones), which resulted in holes in the DEMs due to missing topographic data. Conversely, the vectors of flow velocity

projected by PIV analysis could only be observed in moving zones. Consideration of both the DEMs and the vectors projected by PIV analysis allowed assessment of the relationships between changes in flow direction and fan development.

**R2 Comment 14:** *Results: The results section also suffers from the same problems as the previous sections. What is being measured and compared between the different runs is not specifically stated and as a result it is hard to understand some of the findings of the manuscript. Many of the result figures are poorly explained and some are cited out of order or not cited at all.*

**Reply to R2 Comment 14:** We agree with this comment and have revised the results section by making suitable reference to the figures. For example, the subsection with respect to the formation of fan morphology was revised as shown below (**P9 L275-297**):

**P9 L275-297:** The extent of the changes in flow direction and deposition range of sediment particles differed between the monogranular and multigranular flows. In the first 10 s of flow runout, both types of granular flow descended in an approximately straight direction (Fig. S2). After 20 s from the start of flow runout, the monogranular flows descended in a straight line through the zone with a 9°–12° slope (i.e., from the flume outlet to 2 m downstream), but the flow direction shifted somewhat toward the left-bank side owing to avulsion in the zone with a 6° slope (i.e., over 2 m downstream from the flume outlet) (Fig. 7). The multigranular flows, after 20 s from the start of flow runout, changed their flow direction further in the upper zone (i.e., at approximately 1.8 m downstream from the outlet of the flume) in comparison with the monogranular flows (Fig. 8). In multigranular test runs 1 and 4, the flow direction shifted to the left- and right-bank sides, respectively, whereas in multigranular test runs 2 and 3, the flow bifurcated (Fig. 8).

After 30 s from the start of flow runout, the monogranular flows descended continuously further toward the left-bank side, but in test run 4, the flow became slightly bifurcated (Fig. 9). At this stage, in multigranular test run 1, the flow descent direction shifted somewhat from toward the left-bank side to toward the right-bank side (Fig. 10a). In test runs 2–4, the flow direction was mostly maintained but the location at which the flow direction changed moved ~0.2 m upstream (i.e., to approximately 1.6 m downstream from the outlet of the flume) (Fig. 10b–d). After 40 s from the start of flow runout, at ~2 m lower from the outlet of the flume, the flow bifurcated in monogranular test run 1 (Fig. 11a), continuously descended toward the left-bank side in monogranular test runs 2 and 3 (Fig. 11b, c), and mainly descended toward the right-bank side in monogranular test run 4 (Fig. 11d). In the test runs of the multigranular flows, the point of drifting of flow direction occurred further upstream of the deposition area, i.e., ~1.4 m downstream of the outlet of the flume (Fig. 12). The descent direction of the multigranular flow inclined toward the right-bank side in test runs 1 and 4 (Fig. 12a, d), but inclined toward the left-bank side in test runs 2 and 3 (Fig. 12b, c). Subsequently, there was no substantial change in descent direction of any of the flows (Figs. S3 and S4). The eventual range of sediment

deposition differed in response to grain-size distribution (Figs. S5 and S6), and also varied substantially between the multigranular test runs (Fig. S6).

**R2 Comment 15:** *Line 95: What is the lower portion of the flume? How is this defined?*

**Reply to R2 Comment 15:** The lower position should be stated as the lower measurement position of the ultrasonic sensor. We have revised the related sentences to define the positions used for measurement of the flow height as shown below (**P6 L154-156**):

**P6 L154-156:** The ultrasonic displacement meters were installed at three positions separated by a distance of 1 m above the sediment bed from upstream to downstream in the flume; hereafter, referred to as the upper, middle, and lower measurement positions, respectively (Fig. 1a).

**R2 Comment 16:** *Line 96: Same with arrival point and upper position.*

**Reply to R2 Comment 16:** Similarly, we should define the upper position as the upper measurement position of the ultrasonic sensor. We have revised the related sentences accordingly (please see **Reply to R2 Comment 15**).

**R2 Comment 17:** *Line 97: What is a run? This refers back to the earlier point that it is not clear whether the experiment is single or multiple surges.*

**Reply to R2 Comment 17:** A run indicates an individual experimental test run. We have defined "run" in the revised manuscript accordingly (please see **Reply to R1 Specific comment 11**).

**R2 Comment 18:** *Line 101: Unclear how the arrival time is measured.*

**Reply to R2 Comment 18:** We measured the time at which the flow front started its runout at the deposition area to the time when the flow front stopped as the arrival time. We have added new results with respect to the arrival time in the revised manuscript with an explanation of how we obtained the measurements (please see **Reply to R1 Specific comment 14**).

**R2 Comment 19:** *Lines 103-106: This should be in the discussion or introduction rather than in the results*

**Reply to R2 Comment 19:** We agree with this comment. As part of the changes made to this section, these sentences have been deleted.

**R2 Comment 20:** *Lines 110 – 113: It is unclear which panel is figure 3 is being referred to. Panel 3c is also not cited at all in this section.*

**Reply to R2 Comment 20:** We agree with this comment. As part of the changes made to this section, Figure 3c has been deleted.

**R2 Comment 21:** *Figures S2 and S3 seem important to the overall narrative of the manuscript and therefore the authors should consider including them as part of the main text.*

**Reply to R2 Comment 21:** We appreciate this suggestion and have moved Figs. S2 and S3 to the revised main text. The related sentences have also been revised as shown below (**P9 L266-273**):

**P9 L266-273:** Following multigranular test runs 2 and 3, the grain size of the sediment particles around the midline of the fans was observed (Figs. 5 and 6). At all observation points, relatively large particles were deposited from the surface of the deposition area (i.e., zero on the ruler) to depth of ~1–2 cm (Figs. 5b–f and 6b–f). More small particles were deposited above the relatively large particles at observation points b–e (Figs. 5b–e and 6b–e), indicating that transported sediment particles were stacked above the lobe-like morphology following the halting of the front of the solid phase. Around the front of the solid phase (i.e., the downstream edge of the fans), only relatively large particles were observed (Figs. 5f and 6f). The sediment particles were thus segregated by grain size, and consequently relatively large particles accumulated at the fronts of the multigranular flows.

**R2 Comment 22:** *Lines 119-121: State how the locations of the lobes differ, are they closer to the flume exit? Does the slope differ between the 2 locations?*

**Reply to R2 Comment 22:** We agree that detailed explanation is necessary here. Because the generated DEMs 10 s after the start of runout could not be used to measure the lobe surface owing to lack of data, we could not investigate the differences in slope accurately. Given this, we have explained the difference based on the distance between the lobes to the flume outlet as shown below (**P9 L256-258**):

**P9 L256-258:** In this context, the locations at which the front of the solid phase stopped (i.e., deposited sediment particles) differed between the monogranular and multigranular flows. Thus, in the early stage of formation of fan morphology, in contrast to the monogranular flows, lobe-like morphologies were formed on the upstream side by the multigranular flows (Fig. S2).

**R2 Comment 23:** *Line 124: Why is the series of events being described in terms of time? Time is not likely to be a controlling factor in how the debris flow behaves. The slope over which it is traveling is*

*much more likely to be the control (along with the grain size distribution).*

**Reply to R2 Comment 23:** We sincerely appreciate this important comment and agree that slope is one of the factors that directly controls how a debris flow behaves. In this study, we focused on how a single surge (i.e., a continuous flow) forms a debris-flow fan in accordance with the sediment deposition and inundation, which is the main difference of our study in comparison with previous related studies that focus on whether differences in the characteristics among surges influence fan morphology. These previous studies measured the slope by measuring the topography of lobes before and after supplying each surge. In contrast, the surge moved continuously until the end of an experimental run in this study; therefore, we could not measure changes in the surface slope of the fan during the fan-forming phase. However, because the hydrographs were likely analogous between the monogranular and multigranular flows, the volumes of the transported flows were also similar at the same times. Given this, we compared time series changes in fan formation when the sediment volume coincided between the monogranular and multigranular flows.

**R2 Comment 24:** *Figures 5 – 9 are very poorly explained and hard to interpret. This is not helped by there being no explanation of what is meant by "Run" in the experiment.*

**Reply to R2 Comment 24:** We apologize for the confusion and have revised the related sentences to clarify what run indicates here (a run indicates an individual experimental case, also please see **Reply to R2 Comment 10 and Reply to R2 Comment 17**).

**R2 Comment 25:** *Lines 217-129: A numerical metric would help to compare the shape of the mono vs multigranular flows. Perhaps the angle of deviation from directly straight or a ratio of the left vs right side length?*

**Reply to R2 Comment 25:** We sincerely appreciate this important comment and agree that a numerical metric would help in the comparison of topography between the monogranular and multigranular flows. Considering the simplicity of the definition, we have added the latter metric of the ratio of the length.

To investigate differences in the shape of the fans derived from both the monogranular and multigranular flows, we proposed an index that focuses on fan shape symmetry. The proposed symmetric index (*SI*) is defined as follows:

$SI = LR/LL$ –(1)

where *LR* and *RR* represent the length of the fan from the midline to the edge of the left-bank side and to the edge of the right-bank side of the fan, respectively. When fan width is close to symmetry, the *SI* value is approximately one. We calculated the *SI* values for the width of the fan at cross sections at 0.2 m intervals from the outlet of the flume using orthophotos and DEMs acquired 10, 20, 30, 40,

and 50 s after the start of debris-flow runout (**Fig.14**). The content of Fig. 14 was explained in the revised manuscript as shown below (**P10 L311-333**):

[Figure]

**Figure 14**: Changes in symmetric index (*SI*) values in response to fan-morphology formation. The left and right panels show results for monogranular flows and multigranular flows, respectively. (a) and (b) after 10 s from the start of flow runout, (c) and (d) after 20 s from the start of flow runout, (e) and (f) after 30 s from the start of flow runout, (g) and (h) after 40 s from the start of flow runout, and (i) and (j) after 50 s from the start of flow runout.

**P10 L311-333:** Changes in the SI values revealed the relevance of the shifting flow direction with regard to the formed fan morphology. After 10 s from the start of flow runout, the SI values differed between the test runs, especially at over 1.6 m downstream from the flume outlet, irrespective of the grain-size distribution (Fig. 14a, b). After 20 s from the start of runout of the monogranular flows, SI values of >1 were observed in test run 3 (Fig. 14c), whereas such values were observed in another test run (run 4) after 10 s (Fig. 14a). This reflects the avulsion of the monogranular flows that somewhat shifted the flow direction in the zone with a 6° bed slope (Figs. 7 and 14c). After 20 s from the start of runout of the multigranular flows, the range of the SI values differed substantially between the various test runs (Fig. 14d). At this stage, depending on differences in the extent of avulsion between the multigranular test runs (Fig. 8), the SI values of the monogranular flows were close to 1.0 in test runs 2 and 4, but differed substantially from ~0.3 to 2.0 at 2.2 m downstream from the flume outlet between test runs 1 and 3 (Fig. 14d). Therefore, the cross-sectional asymmetry of the fans became increasingly conspicuous owing to the avulsion process of the multigranular flows.

Although the asymmetry of the fan shape became increased by avulsion in the early stage of the formation of fan morphology, after 30 s from the start of flow runout, the range of SI values became narrow and close to 1.0 irrespective of the measurement location and grain-size distribution (Fig. 14e, f). With the exception of multigranular test run 3 (Fig. 14e), the SI values were in the range of ~0.8–1.3, indicating that both the monogranular and the multigranular flows produced symmetric fan shapes when the flows descended for 30 s (Fig. 14e, f). Because of the variation in flow direction after 40 s from the start of flow runout (Figs. 11 and 12), the range of SI values widened among both the monogranular and the multigranular test runs (Fig. 14g, h). The SI values of the monogranular flows were approximately 1.0 at all measurement locations in test run 4, but were greater than 1.0 in test runs 1–3, especially at distal locations from the flume outlet (Fig. 14g). At this stage, the SI values of the multigranular flows differed substantially between test runs, ranging between ~0.5 and 1.4 at the maxima, indicating notable avulsion (Fig. 14h). Because of the absence of notable changes in flow direction during the period 40–50 s from the start of flow runout (Figs. S3 and S4), the SI values after 50 s were analogous to those after 40 s irrespective of measurement location and grain-size distribution (Fig. 14g–j).

**R2 Comment 26:** *Line 135: The difference in shape at the 2.2m line could be due to the difference in runout. While in the monogranular flow the authors are measuring the apex of the flow height in the multigranular it after the apex. As the debris flows are producing a fan like shape you would expect the fan to be wider after the apex regardless of the granular structure.*

**Reply to R2 Comment 26:** We appreciate this comment and we have carefully described that even monogranular fans exhibited asymmetry in fan shape by referring to the results of the new metric (please see **Reply to R2 Comment 25**). The difference in runout distance was closely linked to the difference in fan morphology between the monogranular and multigranular flows, which is responsible

for the difference in the changes in flow direction during fan-morphology formation. This aspect has been discussed in the discussion section as shown below (**P12 L347-366**):

**P12 L347-366:** Avulsion occurred in both the monogranular and the multigranular flows but its extent and occurrence location differed owing to differences in grain-size distribution (Figs. 7–12). The runout distance of the fronts of the multigranular flows was shorter than that of the monogranular flows (Fig. 4), which led to avulsion of the multigranular flows at locations closer to the outlet of the flume (Figs. 11 and 12). The differences in the extent and location of debris-flow avulsion resulted in different fan morphologies between the monogranular and multigranular flows (Figs. 13 and 14). Thus, changes in the runout distance attributable to differences in the grain-size distribution of the debris flows were responsible for the variation in fan morphology.

Relatively small and large particles within a debris flow can both influence changes in the runout distance of multigranular flow fronts (De Haas et al., 2015b; Hürlimann et al., 2015). In this study, the decrease in flow resistance due to small sediment particles was intentionally avoided by adjusting the composition of the multigranular flows. Indeed, the arrival time of the flow fronts in the flume was similar between the monogranular and multigranular flows (Fig. 2), suggesting that the effects of small particles on flow resistance were negligible. Unlike the unrelated small sediment particles, large sediment particles were accumulated in the multigranular flow fronts, at least during their runout (Figs. 5 and 6), and potentially caused the decrease in the runout distance (Fig. 4). Large sediment particles increase flow resistance and decrease flow velocity as bed slope decreases (e.g., Egashira et al., 1997; Takahashi, 2007). The velocity of the fronts of the solid phase of the multigranular flows decreased substantially when runout distance exceeded 1 m (i.e., when the front reached the point at which the bed slope decreased from 12° to 9°) in comparison with that of the monogranular flows (Fig. 3), suggesting that large particles caused a decrease in flow velocity. Thus, even when debris flows have hydrographs that are similar at the outlet of the channel, differences in the extent of accumulation of large particles in the flow front can lead to changes in runout distance and consequently form fans with different morphology.

**R2 Comment 27:** *Discussion: The discussion, similarly to the introduction, is lacking in detail and is too vague in some of its points to make an impact on the reader. Currently the discussion spends too much time focusing on areas the authors did not investigate (pore fluid seepage) and not enough time putting their results back into the context of the literature. To help the reader the authors should put their results, which are interesting and novel, front and centre and discuss the processes that they actually recorded before moving on to areas they do not have direct evidence for.*

**Reply to R2 Comment 27:** We agree that the discussion section focused too much on the impact of pore fluid seepage, rather than discussing the obtained results in detail. In the revised manuscript, we have concentrated on how differences both in runout distance and in the timing of phase

separation influence differences in fan morphology in response to the grain-size distribution (please see **Reply to R1 Specific comment 17** and **Reply to R2 Comment 26**):

**R2 Comment 28:** *Lines 141 – 142: The term "processes" is too vague and the results do not mention stratigraphy at all so this seems like a strange sentence to start the discussion with.*

**Reply to R2 Comment 28:** We agree with this comment. As part of the changes made to this section, the related sentences have bene revised as shown below (**P13 L392-398**):

**P13 L392-398:** In comparison with the monogranular flows, the multigranular flows formed fans with reasonably asymmetric morphology (Fig. 14), which resulted from the avulsion process that caused marked shifts in flow direction (Fig. 12). Despite differences in the extent of avulsion between the monogranular and multigranular flows, the extent of symmetry in fan morphology increased at the same timings (Figs. 14 and 15), suggesting that the pace at which avulsion occurred was similar irrespective of the grain-size distribution of the debris flow. The wide-ranging grain-size distribution within debris flows thus leads to marked shifts of flow direction by avulsion rather than to changes in the pace of avulsion, and likely expands the horizontal deposition range of the sediment.

**R2 Comment 29:** *Lines 142 – 147: This section is poorly linked to the previous opening statement of the discussions.*

**Reply to R2 Comment 29:** We agree with this comment. As part of the changes made to this section, the related sentences have been revised (please see **Reply to R2 Comment 28**).

**R2 Comment 30:** *Lines 150 – 151: This would also apply if the flow was monogranular with coarse grains.*

**Reply to R2 Comment 30:** We agree with this comment. To avoid vague explanations, we have revised the related sentences by referring to the results with respect to phase separation as shown below (**P13 L408-417**):

**P13 L408-417:** Even in the early stage of flow runout, i.e., after 20 s from the start of runout, the shape of the multigranular fans exhibited asymmetry in comparison with that of the monogranular fans (Fig. 14c, d), which was likely responsible for greater final asymmetry in multigranular fan morphology (Fig. 14i, j). It is likely that the short runout distance of the multigranular flows resulted in thick sediment deposition close to the flume outlet, and the swift phase separation accelerated the inundation of the fluid phase to the distal downstream area. In this sense, phase separation facilitated the increase in the extent of unsaturation of the fan deposits. A bed consisting of unsaturated sediment

particles potentially decreases the pore-fluid pressure at the bottom of a debris flow and increases the resistance of the flow body (Major and Iverson, 1999; Staley et al., 2011), resulting in complex patterns of flow direction and sediment deposition (Tsunetaka et al., 2019). Thus, the variations in the extent of the saturation of the fan sediment materials facilitated by phase separation might have triggered the differences in the fan morphology between the multigranular test runs.

**R2 Comment 31:** *Lines 152 – 155: The authors previously mentioned that there was minimal difference between the thicknesses of the mono and multi-granular flows. This idea of the coarser grains forming an obstacle which diverts the tail of the flow should be expanded upon further with more descriptions if the authors believe it to be significant.*

**Reply to R2 Comment 31:** We agree with this comment. This aspect was likely facilitated by the phase separation. Thus, we have revised the related sentences by referring to the results with respect to phase separation (please see **Reply to R2 Comment 30**).

**R2 Comment 32:** *Lines 158 – 164: It is unclear what the authors are suggesting here. How can there be moisture differences in the bed of the depositional area? The depositional area is the same for all of the test runs? Unless they are discussing deposition on a previously deposited fan? This is very underdeveloped.*

**Reply to R2 Comment 32:** We agree with this comment. Again, with appropriate reference to new results, we have revised the explanation regarding this idea (please see **Reply to R2 Comment 30**).

**R2 Comment 33:** *Lines 165 – 169: This section is a strange ending to the manuscript. It focuses on two areas which the authors did not measure in their study; stratigraphy and moisture content of the fan.*

**Reply to R2 Comment 33:** We agree with this comment. As part of the changes made to this section, the related sentences have been deleted and the end of this section has been revised as shown below (**P14 L418-423**):

**P14 L418-423:** In this context, the extent of phase separation broadly constrains fan morphology. The advance in the multiphase model describing a granular–fluid mixture flow allows us to reflect on the effects of separation between the granular (solid) and fluid phases in numerical simulations (Pudasaini and Mergili, 2019), and to progress the theoretical interpretation of debris-flow mechanics. Our results demonstrate that further investigation of the relationships between the grain-size distribution within debris flows and the extent of phase separation and related changes in runout distance could lead to accurate forecasting of the range of debris flow deposition and inundation.

**R2 Comment 34:** *Conclusions: The conclusions does not contain any references to the discussion and therefore it feels disconnected from the rest of the manuscript.*

**Reply to R2 Comment 34:** We agree with this comment. The conclusion section has been thoroughly revised as shown below (**P14 L426-447**):

**P14 L426-447:** In this study, we conducted flume-based experiments to investigate how differences in the grain-size distribution within debris flows change the morphology of debris-flow fans. Two types of sediment particles were used to generate two types of granular–water mixture flows that imitated a single debris-flow surge: monogranular particles comprising quasi-monodispersed sediment particles and multigranular particles comprising polydispersed sediment particles. The granular–water mixture flows generated using the monogranular particles and the multigranular particles were referred to as monogranular flows and multigranular flows, respectively. The average grain size was adjusted to coincide between the monogranular and multigranular flows, which allowed us to compare the fan morphologies formed by debris flows that had similar flow properties but different grain-size distributions.

Despite similarities in the flow properties before the start of debris-flow runout, the runout distance of the fronts of the multigranular flows was less than that of the monogranular flows, which was likely attributable both to accumulation of relatively large sediment particles, and to the swift separation between the solid and fluid phases of the multigranular flows during runout. The short runout distances of the multigranular flows were responsible for sediment deposition closer to the flume outlet, which led to avulsion that markedly shifted the flow direction during fan formation. Consequently, in comparison with the monogranular fans, the fans of the multigranular flows formed with horizontally asymmetric shapes, highlighting that fan morphology can vary in response to grain-size distribution within a debris flow.

The extent of the symmetry of debris-flow fan morphology increased at a similar time during debris-flow runout irrespective of grain-size distribution and test runs. However, avulsion that shifted the flow direction increased the extent of asymmetry of fan morphology, and also increased the morphological deviations between test runs, especially for the multigranular flows. Therefore, wide-ranging grain-size distribution within a debris flow rather than change in the rate of fan formation likely results in complex fan morphology with high asymmetry. Our results suggest that further understanding of the relationships between differences in grain-size distribution and runout of debris flows could reduce uncertainty in the estimation and interpretation of debris-flow fan morphology.

---

## Author Response (AR2)

We are grateful for the constructive and detailed comments of Prof. Shiva P. Pudasaini and Reviewer 2. In accordance with their helpful and insightful recommendations, we have modified the original manuscript to both improve the presentation and strengthen the discussion on the limitations and potential of the results derived from our reduced-scale flume tests.

Below, we provide our responses to the specific comments and questions received from the reviewers. The comments of the reviewers are shown in italics and different colors (i.e., those of the Associate Editor, Reviewer 1, and Reviewer 2 are presented in green, brown, and blue, respectively), while our responses are provided in normal black text.

**Associate editor COMMENTS AND AUTHOR RESPONSE**

**AE comment:**

*thank you for the revised paper. I have received two further reviews by the original reviewers. Reviewer #1 is largely satisfied with your changes and merely makes a few language suggestions. Reviewer #2 is more critical and makes a few important points. I'd like to highlight two (related) comments. First, there is little discussion about how your experimental results scale up to natural debris flows and their fans. Please add a section in the discussion on this topic, possibly also the mentioning the limited range of conditions that you tested and the small number of data points. Second, most of the analysis and discussion focusses on the scaled parameter values. Reviewer #2 rightly points out that normalized parameters might yield slightly different interpretations and may be more easily generalizable (relating to point 1).*

*I think you can address the comments at various levels, most easily by revising and expanding the discussion. It might strengthen the paper, however, to come up with a suitable non-dimensional framework to present the data.*

*Overall, I think the necessary revisions are moderate. I have decided on major revisions, as I would like reviewer #2 to have another look at the revised manuscript.*

*Please address all of the comments and submit a detailed rebuttal with your revisions. I am looking forward to seeing your revised paper and wish you a very nice weekend.*

*All the best, Jens Turowski*

**Reply to AE comment:** We greatly appreciate the time and effort of the editors and reviewers in providing constructive and insightful feedback, which has helped us improve our manuscript. We sincerely agree with the comment and indications received from Reviewer 2. We have added a new subsection in the discussion section to explain how the experimental results represent the properties

of natural debris-flow fans (**P15 L474-516**). In this subsection, we discuss the similarities between experimental and natural debris-flow fans in terms of qualitative observations and geometrical comparisons (**Figure 19**). On the basis of these similarities, we suggest that the four experimental runs likely exhibited representative results for our experimental setup with adequate reproducibility. At the end of this subsection, we also note the limitation of our flume tests in that they focus only the aspect of the effects of grain-size distribution within debris flows on fan-formation processes, and we emphasize the need for comprehensive assessment using further accumulated field data.

Additionally, to support our interpretation, we investigated changes in the normalized avulsion distance (i.e., the ratio between the distance from the flume outlet to the occurrence point of the avulsion and the distance of the runout of the solid phase) (**Figure 14**) and differences in the surface slope of the final fan morphology (**Figure 16**) between the monogranular and multigranular flows. The results, which do not alter our interpretation and discussion, further highlight that different runout distances can result in different fan morphology (**P11 L321-325** and **P11 L340-351**).

All related figures and text are mentioned in **Reply to R2 Overall comment**. We hope that these responses satisfactorily address all issues and concerns regarding our original submission and that our manuscript might now be considered suitable for publication in Earth Surface Dynamics.

**Reviewer 1 COMMENTS AND AUTHOR RESPONSE**

**R1 General comment 1:**
*The ms has been thoroughly revised, enhancing its quality and clarity. I appreciate the authors for their work. However, in parts, the writing is incomplete, or inconsistent. The description could still be made better, including the mechanisms of erosion and phase separation with respect to the available mechanical models.*

**Reply to R1 General comment 1:** We sincerely appreciate your thorough and helpful review of our manuscript. To improve the presentation, we have carefully revised the manuscript in accordance with your comments. Moreover, in accordance with the comments from the Associate Editor and Reviewer 2, we have added some new results and additional discussion (please see **Reply to AE Comment**) to further improve both the quality and the clarity of the manuscript.

**R1 Specific comment 1:** *L63: "fluctuation of inertia" --> "change of inertia"*

**Reply to R1 Specific comment 1:** We have made the suggested changes, thank you (**P2 L63**).

**R1 Specific comment 2:** *L67: It is better to call "interstitial water and colloidal sediment particles"*

**Reply to R1 Specific comment 2:** We have made the suggested changes, thank you (**P3 L68**).

**R1 Specific comment 3:** *L128: --> "This phenomenon has been explained by Pudasaini and Krautblatter (2021) with their mechanical erosion model for debris flows involving the state of excess energy during the erosion process."*

**Reply to R1 Specific comment 3:** We have added the suggested sentence, thank you (**P5 L137-139**).

**R1 Specific comment 4:** *L175-184: Not clear, which velocity you are talking about, solid, water, or both? Please make it clear.*

**Reply to R1 Specific comment 4:** We apologize for the confusion. Because we measured the velocity of either the solid or the water or both, the sentence has been revised as follows (**P7 L190-191**):

**P7 L190-191:** the paired image sets were processed to estimate the vectors of the flow velocity of either the solid or the water or both at the surface of the deposition area.

**R1 Specific comment 5:** *L201: This definition of the index is applicable only for the geometry, any physically less representative, but there are other asymmetries too! E.g., depth, density, particle-fluid distribution, etc. Please discuss.*

**Reply to R1 Specific comment 5:** We sincerely appreciate this important comment. As you correctly highlighted, asymmetries of the physical (dynamic) properties of debris flows in the deposition area are also crucial regarding changes in fan morphology. However, unfortunately, we could not quantitatively measure such properties after the flow runout. This is an obvious limitation of our flume tests, which we have discussed in relation to the general experimental limitations (**P16 L497-516**, please see **Reply to AE Comment**).

**R1 Specific comment 6:** *L266-273: You could improve descriptions in connection to the figure panels.*

**Reply to R1 Specific comment 6:** According to this comment and R2 Comment 12, we have revised the lead sentence of this paragraph (please see **Reply to R2 Comment 12**).

**R1 Specific comment 7:** *Fig. 5-6: Panels b-f: are these vertical sections?*

**Reply to R1 Specific comment 7:** We apologize for the confusion. To indicate the flow direction, we have revised Figs. 5 and 6 (please see **Reply to R2 Comment 13**).

**R1 Specific comment 8:** *Fig. 7-8 (etc.): The quality is poor.*

**Reply to R1 Specific comment 8:** To improve clarity and quality, we have indicated the location of the point of the avulsion in Figs. 8–13.

**Figure 8:**

[Figure]

**Figure 8:** Fan morphology 20 s after the start of runout of the monogranular flows. The upper and lower panels show orthophotos and digital elevation models (DEMs) with flow vectors, respectively. Respective sets of the upper orthophoto and lower DEM represent corresponding results of each experimental test run. The white arrows on the orthophotos indicate the assumed principal direction of flow descent. The red points in the orthophotos indicate the assumed occurrence point of the avulsion. The elevation of the DEMs is depicted assuming that the area with a 3° slope (i.e., the area furthest downstream from where the slope angle changed from a 6° to 3° slope) has elevation of zero.

**R1 Specific comment 9:** *L371: "Numerical simulations" --> "Numerical simulations based on the mechanical model"*

**Reply to R1 Specific comment 9:** We have made the suggested changes, thank you (**P13 L416-417**).

**R1 Specific comment 10:** *L386: "for the variation in the fluidity" --> "for the state of the erosion induced excess energy and the mobility"*

**Reply to R1 Specific comment 10:** We have made the suggested changes, thank you (**P14 L432-433**).

**R1 Specific comment 11:** *L390: "deposition range." --> "deposition range that can be achieved with the mechanical erosion and mobility model (Pudasaini and Krautblatter, 2021)."*

**Reply to R1 Specific comment 11:** We have made the suggested changes, thank you (**P14 L436-437**).

**R1 Specific comment 12:** *L434: "Despite similarities in the flow properties before the start of debris-flow runout": You can't say this in general. You only have similarity in flow depth. But, you don't know the internal structure, mechanics and dynamics. So, I would re-phrase it and say it 'the debris mixture hydrograph'. Also, in the main text.*

**Reply to R1 Specific comment 12:** We appreciate this comment. We have revised the related sentences to avoid such speculation (**P1 L16, P3 L86, P5 L127,** and **P17 L526-527**).

**R1 Specific comment 13:** *L437-438: "The short runout distances of the multigranular flows were responsible for sediment deposition closer to the flume outlet": two parts of this sentence are the same. This is not the reason. Please explain it physically.*

**Reply to R1 Specific comment 13:** We appreciate this comment. We have revised the related sentences to explain the reason on the basis of the newly added results (**P17 L530-531**).

**P17 L528-529:** The short runout distances of the multigranular flows were responsible for changes in the location at which the avulsion occurred, which led to avulsion that markedly shifted the flow direction during fan formation.

**R1 Specific comment 14:** *References: Please check thoroughly with all necessary information as required by the journal, including doi, paper nr., etc.*

**Reply to R1 Specific comment 14:** We have carefully checked and revised the entries in the reference list to ensure that they provide all the required information in the correct format (**P18 L555-730**).

**Reviewer 2 COMMENTS AND AUTHOR RESPONSE**

**R2 Overall comment:** *Tsunetaka et al present the results of a set of flume experiments where they test the impact of debris flow grain size distributions on the morphology of debris flow fans. By comparing the results of 8 experiment runs, 4 monogranular and 4 multigranular, they discover that debris flows with coarser grains have shorter runouts due to early solid and fluid phase separation. Due to the shorter runout multigranular flows the fans are more complex as changes in direction can occur more frequently during the runout of the flow. While I believe this is an interesting result, in its current form it is difficult to apply it to real debris flow fans. Without this link it is not clear whether this manuscript represents a significant step forward in our understanding of debris flow fan formation.*

*I believe the results of the manuscript are hard to apply to real debris flow fans due to three main reasons. I will describe these reasons briefly here and provide detailed section specific feedback below.*
*In the experiments described, the fans are produced by a single debris flow like surge resulting from a continual flow entraining sediment from the bed of the flume. However, in reality debris flow fans are produced by multiple flows depositing sequentially over the course of potentially thousands of years. Therefore, it is not currently clear how the deposits from the experiments can be compared to actual debris flow fans. Instead, it is likely these results can help us to decipher single events from fan stratigraphy.*

*Most of the analysis of the experiment is in terms of time and distance, both of which are highly specific to the experiment run and cannot easily be scaled up to a real debris flow or easily compared between model runs. Throughout the results and discussion, the authors describe how avulsions in multigranular flows occur earlier (both in terms of distance from the flume outlet and in time) than monogranular flows. However, as this result is likely due to the shorter runout distance of the multigranular flows it is not clear whether this result represents a significant difference in behaviour between the 2 types of flow. By normalising the runout length of the flow, it is possible to compare whether the avulsions in multigranular flows truly occur earlier in the evolution of the fan. If there is no difference in the normalised runs then grainsize is not a controlling factor on avulsion frequency or location. Normalising the runout length will also allow for comparison between the model results and actual debris flows and debris flow deposits.*

*Finally, there is surprisingly little reference to the changing slope of the runout area despite slope being a well-known control on the deposition of sediment. By analysing the slope at which the flows stop at the differences in friction between the flows can be quantified. Multigranular flows may stop in steeper slopes than monogranular flows which could allow for some calculation of the friction angle in the deposited sediment. This result would again help to link the experimental results to real debris*

**Reply to R2 Overall comment:** We sincerely appreciate your insightful review and constructive suggestions. We agree that our experiment focused on the fan morphology formed by a single successive debris-flow surge rather than that formed by an accumulation of multiple debris-flow surges. This arose because of differences in the study objectives and intentions. We intended to investigate whether differences in the grain-size distribution within debris flows could lead to changes in fan morphology via differences in runout characteristics without considering differences in hydrographs and without considering the effects of geomorphology formed by previous debris-flow surges. To convey our intention and to clarify the study targets, we have revised the related sentences (**P3 L83-96**).

We also acknowledge that we performed only four experimental test runs for the respective grain sizes, and that our flume tests focused on a limited aspect of the effects of the grain-size distribution within the debris flows. On the basis of the similarities between experimental and natural debris-flow fans in terms of qualitative and geometrical comparisons, we suggest that despite the limited number of experimental runs, the experiments were well controlled and could have produced representative results under our experimental setup and operation (**Figure 19** and **P15 L475-496**). Additionally, we have emphasized and carefully discussed the limitations of our flume tests (**P16 L497-516**).

Moreover, we compared the normalized avulsion distance (i.e., the ratio between the distance from the flume outlet to the occurrence point of the avulsion and the distance of the runout of the solid phase) (**Figure 14**), which indicated that the avulsion of the multigranular flows occurred closer to the flume outlet in comparison with that of the monogranular flows (**P11 L321-325**). We also compared the surface slope of the final fan morphology between the monogranular and multigranular flows (**Figure 16**). The multigranular flows formed steeper slopes of the fan surface further upstream in comparison with the monogranular flows (**P11 L340-351**). These results support our interpretation and discussion that the short runout distance of the multigranular flows can lead to variations in fan morphology.

We believe that these revisions address all doubts and clarify the implications derived from our flume tests regarding the understanding of natural debris-flow fans.

**P3 L83-96:** The primary objective of this study was to assess how the grain-size distribution within a debris flow influences fan morphology, especially during debris-flow runout and inundation. We conducted reduced-scale flume tests to compare fan morphologies that resulted from single debris-flow surges with different grain-size distributions but with similar sediment mixture hydrographs. To investigate whether differences in the runout properties of both solid and fluid phases cause different sediment deposition patterns, we intended to avoid the effects of morphology resulting from previous debris-flow surges on the debris flow that runs out at the flume outlet. To achieve this, we focused on how a single successive debris-flow surge forms the fan-like morphology

around the flume outlet without geomorphological effects arising from previous debris flows. Thus, in this study, debris-flow fans are defined as the sediment deposition formed by a single successive debris-flow surge rather than the accumulation of multiple debris-flow surges. Using photogrammetry and video-image analysis, we investigated how differences in grain-size distribution within debris flows influence variations in runout characteristics and fan morphologies. The intention underlying this comparison was to interpret the differences in fan morphology in terms of known debris-flow mechanics. The final objective was to elucidate whether differences in grain-size distribution within debris flows could change fan morphology solely by influencing the runout process without variation of the dynamic properties of the debris flow in the channel.

[Figure]

**Figure 19:** Debris-flow volume versus inundation area. Data concerning experiments (n = 454) are from Liu (1996), Major (1997), D'Agostino et al. (2010), De Haas et al. (2015b), Hürlimann et al. (2015), and Baselt et al. (2022). Data concerning natural debris flows (n = 323) are from Abele (1974), Li (1983), Crandell et al. (1984), Siebert (1984), Francis et al. (1985), Siebert et al. (1987), Hazlett et al. (1991), Hayashi and Self (1992), Siebe et al. (1992), Stoopes and Sheridan (1992), Iverson et al. (1998, 2015), Capra et al. (2002), Berti and Simoni (2007), Griswold and Iverson (2008), D'Agostino et al. (2010), Dufresne et al. (2019), Fan et al. (2019), and Friele et al. (2020). Note that the monogranular and multigranular test runs of this study are overlain in the log-log plane, and that the flow volume was approximated as 0.08 m3 on the basis of the supplied sediment volume. The solid black line is the best-fit regression carve ($V = 20A^{2/3}$) derived by Griswold and Iverson (2008).

**P15 L475-496:** It should be noted that the flume tests conducted in this study were operated under limited conditions that considered only two types of grain-size distribution. Therefore, the extent to which the obtained experimental results represent the properties of natural debris-flow fans should be assessed with caution. The observations of the grain-size profiles of the multigranular flows (Figs. 6 and 7) indicate that the grain-size segregation of the sediment particles was similar to that of natural debris flows (e.g., Iverson, 1997; Zanuttigh and Lamberti, 2007; Johnson et al., 2012). Additionally, the wide-ranging grain-size distribution of the debris flows caused horizontal widening of the deposition range (Fig. 15). This relationship between horizontal deposition characteristics and grain-size distribution is also observed in stratigraphic records of natural debris-flow fans (Pederson et al., 2015). Thus, in terms of qualitative observations, our flume tests can be considered representative to a certain extent of the properties of natural debris-flow fans.

In terms of a geometrical scaling relationship, we compared the relationships between the volumes of the debris flows (V) and the inundation areas of the sediment deposits (A) similar to De Haas et al. (2015b) and Baselt et al. (2022) (Fig. 19). The inundation areas of the monogranular and multigranular test runs were ~2.224 $m^2$ and ~2.159 $m^2$, respectively (Table S1), which highlights that when the hydrograph and velocity of debris flows are similar before the start of runout, the effects of grain-size distribution within the debris flows on fan formation are reflected in change in the horizontal shape of the sediment inundation range but without substantial variation in the gross area. Owing to this similarity in the inundation area regardless of the grain-size distribution, all experimental runs were plotted in almost identical locations on the log-log *V-A* plane, and just below the best-fit regression curve for natural debris flows derived by Griswold and Iverson (2008) (Fig. 19). The *V-A* scaling relationships indicate that our experimental results are geometrically within the range of natural debris flows, and that our flume tests were well-controlled experiments across all experimental runs, especially regarding the resultant inundation areas. Given this reproducibility of the inundation area, although we performed only four test runs for both the monogranular flows and the multigranular flows, we believe that the obtained observations adequately reflect the representative behavior of experimental debris flows under the operation and setup of our flume tests.

**P16 L497-516:** In addition to these qualitative and geometric similarities between the experimental and natural debris flows, the similar flow depths suggest that the experimental debris flows were well-controlled in terms of their hydrographs, at least in the flume (Fig. 3). However, some dynamic properties, such as flow resistance (Egashira et al., 2001), sediment erosion and entrainment rate (McCoy et al., 2012; De Haas et al., 2022), and flow friction (Pudasaini and Miller, 2013; Lucas et al., 2014), are strongly governed by the scales of grain size and flow volume. Thus, especially for the experimental debris flows after their runout, our flume tests might not have completely met the dynamic similarity law of debris flows, similar to many other reduced-scale flume tests (e.g., De Haas et al., 2015b; Iverson, 2015). In this regard, our flume tests focus on a limited aspect of the effects of

the grain-size distribution within debris flows. Although the effects of fine sediment (e.g., silt and clay) were intentionally excluded in our experiments to control the hydrograph and the velocity of the debris flows in the flume, fine sediment might alter the resistance and stress structure of natural debris flows (Kaitna et al., 2016; Sakai et al., 2019; Nishiguchi and Uchida, 2022). Because these changes in the resistance and stress of debris flows might affect the rate of separation between the solid and fluid phases (Pudasaini and Fischer, 2020; Baselt et al., 2022), our flume tests could not identify the extent of phase separation on the scale of natural debris flows that comprise wide-ranging sediment particle size from silt to large boulders. In nature, various factors (e.g., phase separation) associated with particle size and grain-size distribution interact, and therefore the behavior of debris flows becomes increasingly complex (e.g., De Haas et al., 2018b). This is reflected in the wide-ranging variation in the *V-A* relationship of natural debris flows (Fig. 19). Our flume tests demonstrate that differences in the grain-size distribution within debris flows can change fan morphology, and likely support interpretation of the formation processes of fan morphology resulting from a single successive debris-flow surge. However, comprehensive assessment of the extent of the respective effects in relation to grain-size distribution within natural debris flows will require further accumulated field data.

**Figure 14:**

[Figure]

**Figure 14:** Change in normalized avulsion distance with time. The error bar indicates the standard deviation between the four experimental runs.

**P11 L321-325:** The normalized avulsion distances of the monogranular test runs were almost constant at approximately 0.67 from after 20 s to 40 s from the start of flow runout (Fig. 14). Unlike this fixed position of the avulsion of the monogranular flows, the normalized avulsion distance of the

multigranular test runs gradually decreased from ~0.78 to ~0.59 from after 20 s to 40 s from the start of flow runout (Fig. 14). This highlights the difference in the trend of the inundation processes between monogranular and multigranular flows.

**Figure 16:**

[Figure]

**Figure 16:** Slopes along the center of the final fans. The slope values were averaged over 0.2 m intervals. The upper, middle, and lower panels indicate monogranular flows, multigranular flows, and their averages, respectively. Vertical broken lines indicate the boundaries of bed slope (i.e., the change points from 12° to 9° and from 9° to 6°).

**P11 L340-351:** Corresponding to the difference in the runout distance of the solid phase (Fig. 5), the surface slopes along the center of the final fan morphology were different between the monogranular and multigranular flows (Fig. 16). The slopes of the monogranular test runs were similar at ~10° from the flume outlet to ~2 m downstream, but they increased to a maximum of ~15° and became somewhat varied further downstream between experimental runs (Fig. 16a). Similarly, the slopes of the multigranular test runs were similar at ~10° from the flume outlet to ~1.5 m downstream (Fig. 16b). However, the slopes of the multigranular test runs started to increase further upstream in comparison with the monogranular test runs; the slopes increased to a maximum of ~23° from ~1.5 to ~2.2 m downstream (Fig. 16b). These differences, reflected in the averaged slopes, indicate steeper surface slopes of the multigranular-fan morphology (Fig. 16c). Note that beyond 2.5 m downstream, the deposition thickness of the multigranular test runs was close to zero (Fig. 15b), indicating that the slope values do not represent the surface slopes of the final fan morphology. Indeed, in the section from 2.5 to 3.0 m downstream from the flume outlet, the slopes of the multigranular test runs were gentler (i.e., ~6°–8°) and closer to the surface slope of the deposition area (6°) in comparison with the monogranular test runs (Fig. 16c).

**R2 Comment 1:** *Line 16: This line is confusing to readers as it is not currently clear how a debris flow can have a flow depth before it starts to runout.*

**Reply to R2 Comment 1:** We agree with this comment and we have added some additional information to improve clarity (**P1 L11-19**):

**P1 L11-19:** Therefore, using a flume connected to a deposition area (inundation plane), this study conducted fan-morphology experiments to assess the effects of differences in grain-size distribution within debris flows on changes in fan morphology. Two types of debris-flow material, i.e., monogranular particles comprising monodispersed sediment particles and multigranular particles comprising polydispersed sediment particles, were used to generate monogranular and multigranular experimental debris flows, respectively. By adjusting the average grain size coincident between the monogranular and multigranular flows, we generated two types of debris flow with similar debris mixture hydrographs but different grain-size distributions in the flume. Although the flow depths were mostly similar between the monogranular and multigranular flows before the start of the debris-flow runout at the deposition area, the runout distances of the front of the multigranular flows were shorter than those of the monogranular flows.

**R2 Comment 2:** *Lines 69&61: Which processes are being referred to here?*

**Reply to R2 Comment 2:** We appreciate this question. We have revised the related sentences to improve clarity (**P2 L60-62** and **P3 L69-71**):

**P2 L60-62:** Moreover, the velocity that erodes channel deposits is susceptible to the influence of both grain-size distribution and slope of the channel bed (Egashira et al., 2001; Takahashi, 2007), and this erosion velocity differs fundamentally from the velocity that entrains the eroded sediment (Pudasaini and Krautblatter, 2021).

**P3 L69-71:** When debris flows leave the channel outlet, the relative difference in velocity between the solid and fluid phases increases and leads to phase separation (Pudasaini and Fischer, 2020; Baselt et al., 2022).

**R2 Comment 3:** *Line 68: It is not clear what is meant by "Discharge around". Perhaps "when debris flows leave the channel outlet…" would work better*

**Reply to R2 Comment 3:** We have made the suggested changes, thank you (**P3 L69-71**, please see **Reply to R2 Comment 2**).

**R2 Comment 4:** *Line 71: Unclear what is meant by "The progress of phase separation continues"*

**Reply to R2 Comment 4:** We agree with this comment. We have revised the related sentences to improve clarity (**P3 L71-72**):

**P3 L71-72:** Around the channel outlet, the solid phase eventually translates into sediment deposition, but the fluid phase continuously descends with the progress of phase separation.

**R2 Comment 5:** *Line 73: How is runout distance defined in this circumstance? Is it defined by the runout of the solid phase?*

**Reply to R2 Comment 5:** Your assumption intimated in the second question is correct in relation to the first. We intended to describe the runout distance of the solid phase and we have revised the related sentence accordingly (**P3 L72-75**):

**P3 L72-75:** The extent of the phase separation might vary in response to the grain-size distribution within a debris flow (Major and Iverson, 1999; Pudasaini and Fischer, 2020), potentially resulting in further difference in runout distance of the solid phase, in addition to the effects attributable to sediment erosion and entrainment processes in the channel.

**R2 Comment 6:** *Line 77: Unclear what "runout around the channel outlet means"*

**Reply to R2 Comment 6:** We appreciate this comment and we have revised the related sentence to improve clarity (**P3 L76-78**):

**P3 L76-78:** In other words, the grain-size distribution can influence the characteristics of both the debris-flow development in the channel and the runout distance after debris flows leave the channel outlet.

**R2 Comment 7:** *Line 84: There needs to a definition of what the authors consider a debris flow fan to be and how their experiment replicates this definition. Currently the experiments do not seem to result in a fan as defined by the references discussed in the references of the introduction.*

**Reply to R2 Comment 7:** To convey our intention and to explain why we designed the experiments to focus on fan morphology formed by a single continuous debris flow, we have included additional clarifying sentences (**P3 L83-96**, please see **Reply to R2 Overall comment**).

**R2 Comment 8:** *Line 118: What are these "similar flow properties"? Perhaps these need to be defined before discussing the differences between the flows.*

**Reply to R2 Comment 8:** We agree with this comment. To avoid such vague explanation, we have used the phrase "similar sediment mixture hydrographs" in the revised manuscript (please see **Reply to R1 Comment 12**).

**R2 Comment 9:** *Line 173: How are the solid and fluid phases identified and defined?*

**Reply to R2 Comment 9:** To improve clarity, we have added new figures (**Figures S1 and 2**) and referred to them as appropriate in the main text (**P6 L179-186**):

**Figure S1:**

[Figure]

**Figure S1:** Images extracted from the captured video with respect to flow runout of the monogranular test run4: (a) before the start of phase separation and (b) after the start of phase separation. Drawn grid lines indicate a square grid (0.2 × 0.2 m). The black line indicates the front of the solid phase, whereas the lower edge of the fluid phase is captured by the white line in the video. In (a), the solid and fluid phases are shown descending synchronously; in (b), the fluid phase has reached further downstream owing to phase separation.

**Figure 2:**

[Figure]

**Figure 2:** Sketch showing definitions of measurements associated with the flume experiments. The centerline is drawn as an extension of the central longitudinal axis of the flume.

**P6 L179-186:** We measured changes in the runout distance of the fronts of the generated debris flows with temporal resolution of 0.1 s using the captured video and the grid lines drawn in the deposition area. During the early stage of debris-flow runout, the solid phase (sediment particles) and fluid phase (clear water) descended synchronously as a complete granular–water mixture flow (Fig. S1a), but then they flowed separately with different velocities in accordance with the deceleration of the solid phase (Fig. S1b). Because the timing of the phase separation was clear in all experimental cases, we measured the fronts of both the solid and the fluid phases after separation to compare the extent of phase separation between the monogranular and multigranular flows. The runout distances of the solid and fluid phases were defined as the distance from the flume outlet to the front of the solid and fluid phases, respectively (Fig. 2).

**R2 Comment 10:** *Lines 200 – 205: The calculation of this metric could be better explained by diagram, currently it is not clear how the mid line is defined or what "the length of the fan from the midline to the edge…" describes.*

**Reply to R2 Comment 10:** We appreciate this comment. We have added a new figure to explain the definitions (**Figure 2**, please see **Reply to R2 Comment 9**) and we have revised the related sentences that describe these definitions (**P8 L224-228**):

**P8 L226-230:** Additionally, to investigate the differences in the shape of fans derived from both the

monogranular and the multigranular flows, we proposed an index that focuses on fan-shape symmetry. The proposed symmetric index (*SI*) is defined as follows:

$$SI = LL/LR \qquad (1)$$

where *LL* and *LR* represent the length of the fan from the centerline of the flume to the edge of the left-bank side and to the edge of the right-bank side of the fan shape, respectively (Fig. 2).

**R2 Comment 11:** *Line 237: How is the runoff distance measured and defined? Does the runout begin in the flume or once it enters the deposition zone?*

**Reply to R2 Comment 11:** We appreciate these questions. We have added a new figure (**Figure 2**, please see **Reply to R2 Comment 9**) and additional sentences (**P4 L108** and **P6 L185-186**) to clarify both the measurement method and the definition of runout distance:

**P4 L108:** In this study, debris-flow runout is defined as the descent of the debris flow downstream from the flume outlet.

**P6 L185-186:** The runout distances of the solid and fluid phases were defined as the distance from the flume outlet to the front of the solid and fluid phases, respectively (Fig. 2).

**R2 Comment 12:** *Line 266: It is unclear what is meant by "following multigranular test runs 2 and 3". Perhaps "the grain size of the deposits of test runs 2&3 were observed…" works better*

**Reply to R2 Comment 12:** We have made the suggested changes, thank you (**P10 L292**):

**P10 L292:** The grain sizes of the deposits of multigranular test runs 2 and 3 were observed (Figs. 6 and 7).

**R2 Comment 13:** *Lines 267-270: It is not clear what a depth of 1-2cm from the surface of the deposition area means. Are the coarser grains 1-2 cm above the base or 1-2 cm below the surface of the deposit? It is also not clear whether the deposit is fining upwards or has reverse grading.*

**Reply to R2 Comment 13:** We appreciate this comment. We have revised the related sentences to improve clarity, and we have modified Figs. 6 and 7 to indicate the direction of the measurement profiles (**P10 L292-294** and **Figures 6 and 7**).

**P10 L292-294:** At all observation points, relatively large particles were deposited above the base of the deposition area (i.e., zero on the ruler) to thickness of ~1–2 cm (Figs. 6b–f and 7b–f).

**Figure 6:**

[Figure]

**Figure 6:** (a) Orthophoto of the debris-flow fan formed by the multigranular flow in test run 3. The red circles indicate the points at which the images were taken. Images of the longitudinal profile from the right-bank side view: (b) 1 m downstream from the flume outlet (slope change point from 12° to 9°), (c) 1.4 m downstream from the flume outlet, (d) 1.8 m downstream from the flume outlet, (e) 2 m downstream from the flume outlet (slope change point from 9° to 6°), and (f) 2.4 m downstream from the flume outlet.

**R2 Comment 14:** *Line 307: Deposition depth is not intuitive, I would use deposit thickness.*

**Reply to R2 Comment 14:** We agree with this comment and we have changed the term "deposition depth" to "deposit thickness" (**P10 L293**, please see **Reply to R2 Comment 13**).

**R2 Comment 15:** *Line 347: The locations of the avulsions cannot be compared between runs due to*

*the difference in runout length. Normalising the runout length may indicate that the locations of the avulsions are fairly similar across all of the experiment runs.*

**Reply to R2 Comment 15:** We appreciate this comment. Comparison of the normalized avulsion distances indicated that the locations of the avulsions were different between the monogranular and multigranular flows (**Figure 14** and **P11 L321-325**, please see **Reply to R2 Overall comment**).

**R2 Comment 16:** *Line 363: The lack of consideration of the change in slope in the discussion of runout length is strange. An important confirmation of the coarse grains enhancement of friction would be if the multigranular flow stop on steeper slopes than the monogranular flows. This would also allow for easier comparison with real examples of debris flows.*

**Reply to R2 Comment 16:** We appreciate this comment. Comparison of the surface slope of the final fan morphology indicated that the multigranular flows resulted in a steeper surface slope in comparison with that of the monogranular flows. This aspect is reflected in the revised manuscript (**Figure 16** and **P11 L340-351**, please see **Reply to R2 Overall comment**).

**R2 Comment 17:** *Line 408: There has been no discussion on how coarser grains may cause avulsions other than by increasing the friction of the front of the debris flow. Therefore, it is not clear why a model which considers the friction of debris flows will not be able to replicate fan morphology.*

**Reply to R2 Comment 17:** We apologize for the confusion. We have revised the related sentences to improve the clarity of this discussion (**P14 L446-457**):

**P14 L446-457:** Importantly, in comparison with monogranular fans, the extent of asymmetry of the multigranular fans differed substantially between test runs (Figs. 17 and 18). The variations in the asymmetry of the multigranular fans suggest that debris flows with wide-ranging grain-size distribution can randomly shift their descent direction when the flows behave as unsteady flows that are freed from horizontal constraints owing to the channel-like topography. Some models assume that multigranular debris flows can be approximated to monogranular debris flows with the same average grain size (e.g., Egashira et al., 1997; Takahashi, 2007). Despite this assumption, such models allow estimation of debris-flow properties such as flow velocity and depth, especially under a steady flow state (Egashira et al., 1997; Takahashi, 2007). Indeed, in the flume, experimental results exhibited similar flow velocity and depth as debris flows with the same average grain size but with different grain-size distributions (Fig. 3). However, given that natural debris flows generally consist of wide-ranging grain-size sediment particles (e.g., Zanuttigh and Lamberti, 2007; Johnson et al., 2012), the use of debris-flow models that involve grain-size approximation could result in errors in the estimated runout distance of debris flows owing to unsteady behavior during flow runout. This likely

leads to inevitable uncertainty in the estimation of fan morphology formed by debris-flow runout.

**R2 Comment 18:** *Line 416: How much of this phase separation is resulting from the lack of very fine sediment in the flow? Clay particles significantly increase pore pressures in the flow and could prevent phase separation and possibly increase the runout length of the flow.*

**Reply to R2 Comment 18:** We appreciate this comment. As you correctly highlighted, our flume tests could not reveal the effects of fine sediment on phase separation and runout characteristics. We have explained these limitations of our flume tests in the revised manuscript (**P16 L497-516**, please see **Reply to R2 Overall comment**).